# SPLR: A Spiking Neural Network for Long-Range Temporal Dependency Learning

## Abstract

Spiking Neural Networks (SNNs) offer an efficient framework for processing event-driven data due to their sparse, spike-based communication, making them ideal for real-time tasks. However, their inability to capture long-range dependencies limits their effectiveness in complex temporal modeling. To address this challenge, we present a **SPLR (SPiking Network for Learning Long-range Relations)**, a novel architecture designed to overcome these limitations. The core contribution of SPLR is the **Spike-Aware HiPPO (SA-HiPPO)** mechanism, which adapts the HiPPO framework for discrete, spike-driven inputs, enabling efficient long-range memory retention in event-driven systems. Additionally, SPLR includes a convolutional layer that integrates state-space dynamics to enhance feature extraction while preserving the efficiency of sparse, asynchronous processing. Together, these innovations enable SPLR to model both short- and long-term dependencies effectively, outperforming prior methods on various event-based datasets. Experimental results demonstrate that SPLR achieves superior performance in tasks requiring fine-grained temporal dynamics and long-range memory, establishing it as a scalable and efficient solution for real-time applications such as event-based vision and sensor fusion in neuromorphic computing.

## 1 Introduction

Spiking Neural Networks (SNNs) are an emerging paradigm in neuromorphic computing, offering significant advantages in terms of energy efficiency, low latency, and event-driven processing. These characteristics make SNNs particularly suitable for real-time applications such as event-based vision, sensor fusion, and neuromorphic signal processing Roy et al. (2019); Davies et al. (2018); Rathi et al. (2023); Frenkel et al. (2021). Unlike conventional deep learning architectures, SNNs leverage sparse, spike-based communication to process data in an asynchronous and biologically inspired manner. Despite these advantages, SNNs face fundamental limitations in modeling long-range temporal dependencies. Specifically, the reliance on exponentially decaying membrane potentials in typical spiking neuron models like Leaky Integrate-and-Fire (LIF) neurons leads to rapid loss of historical information Bengio et al. (1994); Hochreiter & Schmidhuber (1997); Neftci et al. (2019); Bellec et al. (2018). This results in poor performance on tasks requiring retention of information over extended sequences, such as gesture recognition, activity classification, and spatio-temporal reasoning.

Addressing this limitation in SNNs has been a longstanding challenge. Traditional approaches have adapted techniques from recurrent neural networks (RNNs) and deep learning to the spike-based domain, such as incorporating surrogate gradient methods or dense recurrent connections Wu et al. (2018); Shrestha & Orchard (2018). However, these methods often incur significant computational costs, requiring frequent updates that are poorly aligned with the sparse, event-driven nature of SNNs. Furthermore, these approaches fail to fully exploit the inherent advantages of spike-driven dynamics, limiting their efficiency and scalability.

Recent advances in State-Space Models (SSMs) provide a promising framework for modeling long-range dependencies. SSMs, such as the Structured State Space (S4) and Mamba models, excel in encoding long-range state representations using continuous-time formulations Gu et al. (2020; 2022). However, these models are typically designed for dense and synchronous systems, and their direct integration into the discrete, asynchronous processing paradigm of SNNs has remained challenging.

Specifically, the mismatch between continuous memory updates in SSMs and the event-driven nature of SNNs leads to inefficiencies and incompatibilities Gu et al. (2021); Hasani et al. (2021).

In this work, we propose **SPLR (SPiking Network for Learning Long-Range Relations)**, a novel SNN architecture that integrates spike-adapted state-space dynamics to effectively model both short- and long-term temporal dependencies in event-driven systems. At the heart of SPLR is the **Spike-Aware HiPPO (SA-HiPPO)** mechanism, which dynamically adapts continuous memory retention frameworks for spike-based inputs, enabling efficient long-range memory retention while preserving the sparsity and low latency of SNNs. The SA-HiPPO mechanism is seamlessly integrated into the **SPLR convolutional layer**, which leverages state-space principles for efficient spatio-temporal feature extraction.

The SA-HiPPO mechanism within the SPLR convolutional layer adapts continuous-time memory retention for the sparse, asynchronous nature of SNNs by aligning memory updates with spike timings through a dynamic decay matrix. This ensures robust long-term dependency modeling while preserving computational efficiency. Building on this, the layer employs FFT-based convolutions and low-rank approximations for scalable spatio-temporal feature extraction. Unlike conventional frame-based methods, SPLR processes spikes event-by-event, maintaining temporal resolution and reducing latency and computational overhead, enabling accurate, efficient event-driven processing for tasks with complex temporal dependencies.

Our contributions are summarized as follows:

- **SPLR Architecture:** A novel SNN design that integrates spike-adapted state-space dynamics to overcome the limitations of traditional SNNs in modeling long-range dependencies.

- **Spike-Aware HiPPO (SA-HiPPO):** An adaptation of the HiPPO framework for spike-driven inputs, introducing dynamic memory retention mechanisms aligned with inter-spike intervals.

- **SPLR Convolutional Layer:** A state-space-inspired convolutional layer combining FFT-based operations and low-rank approximations for efficient spatio-temporal feature extraction in event-driven systems.

- **Scalability and Efficiency:** A unified architecture that achieves superior performance on event-based benchmarks while maintaining computational efficiency, scalability, and low latency for real-time applications.

## 2 RELATED WORKS

SSMs have emerged as a powerful framework for capturing long-range dependencies by encoding state information over extended sequences Gu et al. (2020; 2022). However, while continuous-time SSMs excel in dense and synchronous settings, adapting them to the sparse, asynchronous nature of SNNs poses significant challenges Gu et al. (2021); Hasani et al. (2021). Existing solutions, such as **SpikingLMU** Liu et al. (2024b) and **BinaryS4D** Stan & Rhodes (2024), attempt to address this gap but fall short in fully leveraging the unique characteristics of SNNs. SpikingLMU incorporates Legendre Memory Units (LMUs) into SNNs for long-range dependency modeling but relies on dense recurrent computations, which undermine the event-driven efficiency of SNNs. Similarly, BinaryS4D integrates state-space dynamics into spiking architectures but employs floating-point matrix multiplications, resulting in a hybrid model that deviates from the sparse, fully spiking paradigm and incurs computational overhead, limiting its suitability for real-time applications.

One of the critical limitations in existing approaches is the inability to effectively handle the *irregular and sparse timing of spikes* while maintaining efficient and robust long-range temporal modeling. Continuous-time memory mechanisms like **HiPPO** Gu et al. (2020) have demonstrated success in dense systems by optimizing memory retention over continuous sequences. However, their reliance on continuous updates and dense matrix computations makes them ill-suited for asynchronous, event-driven systems like SNNs.

To overcome these limitations, we propose the **Spike-Aware HiPPO (SA-HiPPO)** mechanism, a novel adaptation of HiPPO for spiking systems. SA-HiPPO introduces a *dynamic decay matrix* that adjusts memory retention based on inter-spike intervals, allowing it to align memory updates with the sparse and asynchronous nature of spike events. This innovation eliminates the need for dense updates,

preserving the computational efficiency and latency advantages of SNNs while enabling robust long-range temporal modeling. Unlike prior approaches, SA-HiPPO operates *entirely in an event-driven manner*, making it uniquely suited for real-time neuromorphic applications such as dynamic vision Gehrig & Scaramuzza (2024) and temporal reasoning Xiao et al. (2024), where *irregular timing* and low latency are critical. Event-driven systems in neuromorphic vision have explored hybrid strategies that combine frame- and event-based approaches to process high-speed temporal data Gehrig & Scaramuzza (2024); Schöne et al. (2024). While these methods are effective for specific tasks, they often fail to scale for long-range temporal dependencies in asynchronous data streams. Similarly, dendritic-inspired models like **DH-LIF** Zheng et al. (2024b) improve temporal processing through heterogeneous dynamics but introduce significant computational overhead, limiting their scalability for large datasets and real-time applications.

Our proposed **SA-HiPPO** addresses these limitations by introducing a dynamic decay matrix that adjusts memory retention based on inter-spike intervals. Unlike prior approaches, SA-HiPPO operates entirely in an event-driven manner, aligning memory updates with the sparse and asynchronous nature of spike events. This innovation preserves the computational efficiency and latency advantages of SNNs while enabling robust long-range temporal modeling, making it particularly suitable for real-time neuromorphic applications such as dynamic vision and temporal reasoning.

Table 1: Comparison of SPLR with prior methods, highlighting features like memory retention, event-driven processing, scalability, efficiency, asynchronous updates, and adaptability.

| Model | Type | Dynamic Memory Retention | Event-Driven Processing | Scalable Long-Range Modeling | Low Computational Overhead | Fully Asynchronous Updates | Adaptability to Sparse Data |
|---|---|---|---|---|---|---|---|
| **SpikingLMU** Liu et al. (2024b) | SSM | ✓ | ✗ | ✗ | ✗ | ✗ | ✓ |
| **BinaryS4D** Stan & Rhodes (2024) | SSM | ✗ | ✗ | ✓ | ✗ | ✗ | ✗ |
| **HiPPO** Gu et al. (2020) | SSM | ✗ | ✗ | ✓ | ✗ | ✗ | ✗ |
| **DH-LIF** Zheng et al. (2024b) | SNN | ✗ | ✓ | ✗ | ✗ | ✓ | ✗ |
| **EventMamba** Ren et al. (2024) | Hybrid CNN-SSM | ✗ | ✗ | ✓ | ✗ | ✗ | ✗ |
| **EventNet** Turrero et al. (2024) | Transformer | ✗ | ✗ | ✓ | ✗ | ✓ | ✗ |
| **SpikeRWKV** Yao et al. (2024) | Transformer | ✗ | ✓ | ✓ | ✗ | ✓ | ✓ |
| **S4** Gu et al. (2022) | SSM | ✗ | ✗ | ✓ | ✓ | ✗ | ✗ |
| **SPLR** (Ours) | SSM-SNN | ✓ | ✓ | ✓ | ✓ | ✓ | ✓ |

## 3 METHODS

The SPLR Model is designed to process asynchronous, sparse data in a biologically-inspired manner. This model combines several novel components, including dendritic attention mechanisms and SSMs, to efficiently handle event-based spiking inputs and capture long-range temporal dependencies. Figure 1(a) provides a high-level architecture of the model. The **Dendrite Attention Layer** first extracts spatio-temporal features from input spikes, which are then reduced spatially in the **Spatial Pooling Layer**. The **SPLR Convolution Layer** captures temporal dynamics and long-range dependencies, while the **Spike-Aware HiPPO (SA-HiPPO)** mechanism dynamically manages memory retention. Finally, the **Readout Layer** aggregates information for downstream tasks.

**1. Input Representation:** The input to the model is represented as a sequence of spike events, each defined by the tuple $(x, y, t, p)$, where $(x, y)$ are the spatial coordinates, $t$ is the timestamp, and $p$ is the magnitude or polarity of the spike. These events are streamed asynchronously, reflecting the sparsity of the data.

**2. Dendrite Attention Layer:** The model begins by passing the input through the *Dendrite Attention Layer*, constructed using DH-LIF neurons Zheng et al. (2024a), as shown in Figure 1(b). Each DH-LIF neuron has multiple dendritic branches, each characterized by a different timing factor $\tau_d$, enabling it to capture temporal dynamics across various scales. This is essential for accommodating the diverse timescales present in asynchronous spike inputs.

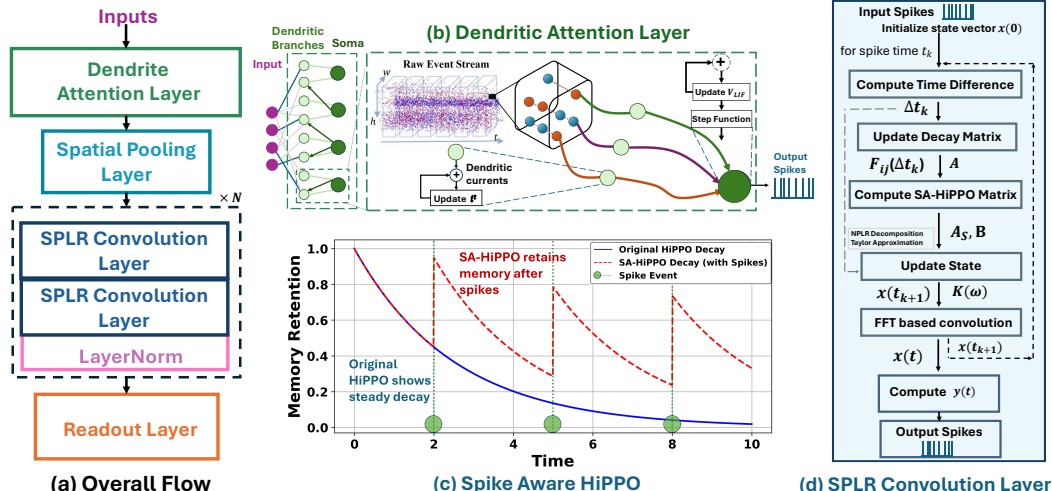

Figure 1: Block diagram of the proposed model architecture. Input spikes are processed event-by-event by the Dendrite Attention Layer, which extracts spatio-temporal features from local dendritic branches. The Spatial Pooling Layer aggregates spikes, followed by SPLR Convolution and Layer-Norm layers, which are repeated $N$ times to enable hierarchical feature extraction and long-range temporal dependency modeling.

The dynamics of the dendritic current $\mathbf{i}_d(t)$ are governed by:

$$\mathbf{i}_d(t+1) = \alpha_d \mathbf{i}_d(t) + \sum_{j \in \mathcal{N}_d} \mathbf{w}_j p_j,$$

where $\alpha_d = e^{-\frac{1}{\tau_d}}$ is the decay rate for branch $d$, and $\mathbf{w}_j$ represents the synaptic weight associated with presynaptic input $p_j$. The set $\mathcal{N}_d$ represents the presynaptic inputs connected to dendrite $d$, ensuring that each dendrite captures temporal features independently, acting as a temporal filter. Unlike a standard CUBA LIF neuron model, which integrates all inputs uniformly at the soma with a single timescale, the dendritic attention layer introduces multiple dendritic branches, each independently filtering inputs at different temporal scales. This design enables the neuron to selectively process asynchronous inputs and retain information across diverse temporal windows, providing greater flexibility and adaptability.

The dendritic currents from each branch are aggregated at the soma of the LIF neuron, resulting in the membrane potential:

$$V(t+1) = \beta V(t) + \sum_d \mathbf{g}_d \mathbf{i}_d(t),$$

where $\beta = e^{-\frac{1}{\tau_s}}$ represents the soma's decay rate, and $\mathbf{g}_d$ represents the coupling strength of dendrite $d$ to the soma. A spike is generated whenever the membrane potential exceeds a threshold $V_{\text{th}}$, allowing the neuron to selectively fire only when sufficiently excited.

**3. Spatial Pooling Layer:** Following the dendritic attention layer, a *Spatial Pooling Layer* is introduced to reduce the spatial dimensionality of the resulting output. Given the initial spike activity $I(x, y, t)$ at location $(x, y)$, the pooling operation reduces spatial dimensions while preserving temporal resolution:

$$I_{\text{pooled}}(x', y', t) = \max_{(x,y) \in P(x', y')} \left\{ I(x, y, t) \right\},$$

where $P(x', y')$ is a pooling window centered at $(x', y')$. Pooling reduces spatial complexity, simplifying subsequent processing in the network while retaining key features. This is especially useful for handling high-dimensional (HD) event streams, where the input contains large spatial areas.

**4. SPLR Convolution Layer:**

The **Spiking Network with Long-term Recurrent Dynamics (SPLR) Convolution Layer** is a pivotal component of the SPLR model, tailored to process event-based spiking inputs .

It captures long-range temporal dependencies and asynchronous dynamics by integrating the **Spike-Aware HiPPO (SA-HiPPO)** mechanism, **Normal Plus Low-Rank (NPLR) Decomposition**, and **Fast Fourier Transform (FFT) Convolution**. Together, these components enable efficient and scalable spatio-temporal feature extraction for event-driven systems (Figure 1(d)).

**Temporal Dynamics: Spiking State-Space Model:** Temporal dependencies are modeled using a continuous-time state-space representation:

Table 2: Table showing the SPLR components and their roles

| Component | Details |
| --- | --- |
| **Dendrite Attention Layer** | **Key Contribution:** Models multi-timescale dynamics inspired by biological dendrites. 
 **Role and Features:** Aggregates spiking inputs from multiple dendritic branches to enhance temporal robustness and filter noise dynamically. |
| **Spatial Pooling Layer** | **Key Contribution:** Reduces spatial complexity while preserving temporal features. 
 **Role and Features:** Prevents spatial bottlenecks and extracts critical spatio-temporal features efficiently. |
| **SPLR Convolution Layer** | **Key Contribution:** Combines SA-HiPPO, FFT, and NPLR for efficient spatio-temporal modeling. 
 **Role and Features:** Captures long-range dependencies with low latency and high scalability. |
| ⇒ **SA-HiPPO:** | *Dynamic memory retention* tailored for event-driven systems, enabling robust temporal modeling by *adapting memory retention* to inter-spike intervals. |
| ⇒ **FFT Convolution:** | Accelerated temporal modeling in the frequency domain, facilitating *fast and efficient processing* of high-dimensional temporal data. |

$$\dot{\mathbf{x}}(t) = \mathbf{A} \cdot \mathbf{x}(t) + \mathbf{B} \cdot \mathbf{S}(t),$$
$$\mathbf{y}(t) = \mathbf{C} \cdot \mathbf{x}(t),$$

where $\mathbf{x}(t) \in \mathbb{R}^N$ is the internal state, $\mathbf{S}(t) \in \mathbb{R}^M$ is the input spike train, and $\mathbf{A}, \mathbf{B}, \mathbf{C}$ are system matrices. Each spike train $\mathbf{S}(t)$ is represented as $S_i(t) = \sum_k \delta(t - t_i^k)$, where $\delta(t)$ denotes the Dirac delta function.

The **Spike-Aware HiPPO (SA-HiPPO)** (Figure 1(c)) mechanism adapts memory retention dynamically using a decay matrix $\mathbf{F}(\Delta t)$, which depends on inter-spike intervals ($\Delta t$). Thus, the state matrix $\mathbf{A_S}$ is modified as:

$$\mathbf{A_S} = \mathbf{A} \circ \mathbf{F}(\Delta t), \quad F_{ij}(\Delta t) = e^{-\alpha_{ij} \Delta t},$$

where $\circ$ denotes the Hadamard (element-wise) product, $\Delta t = t_j - t_i$ is the time difference between spikes $i$ and $j$, and $\alpha_{ij}$ is a decay parameter. The mechanism ensures that recent spikes have a stronger influence on the hidden state, while older spikes decay exponentially, preserving stability and responsiveness. State evolution operates in two modes: continuous dynamics and updates at spike times. Between spikes, the state evolves as:

$$\dot{\mathbf{x}}(t) = \mathbf{A_S} \cdot \mathbf{x}(t).$$

At spike times $t_k$, the state is updated using:

$$\mathbf{x}(t_{k+1}) = e^{\mathbf{A_S} \Delta t_k} \cdot \mathbf{x}(t_k) + \mathbf{A_S}^{-1} \big( e^{\mathbf{A_S} \Delta t_k} - \mathbf{I} \big) \cdot \mathbf{B} \cdot \mathbf{S}(t_k),$$

where $\Delta t_k = t_{k+1} - t_k$. For computational efficiency, the matrix exponential $e^{\mathbf{A_S} \Delta t_k}$ is approximated via a truncated Taylor series:

$$e^{\mathbf{A_S} \Delta t_k} \approx \mathbf{I} + \mathbf{A_S} \cdot \Delta t_k + \frac{\mathbf{A_S}^2 \cdot (\Delta t_k)^2}{2}.$$

**Efficiency via NPLR Decomposition.** To scale computations efficiently, the SPLR Convolution employs **Normal Plus Low-Rank (NPLR) Decomposition**:

$$\mathbf{A_S} = \mathbf{V} \mathbf{\Lambda} \mathbf{V}^* - \mathbf{P} \mathbf{Q}^*,$$

where $\mathbf{V}$ is a unitary matrix, $\mathbf{\Lambda}$ is diagonal, and $\mathbf{P}, \mathbf{Q}$ are low-rank matrices with rank $r \ll N$. This decomposition reduces matrix-vector multiplication complexity from $O(N^2)$ to $O(Nr)$, making it feasible for large state spaces.

**Long-Range Dependencies via FFT Convolution.** Long-range temporal dependencies are captured using **FFT-based convolution**. The system's impulse response is precomputed as:

$$\mathbf{K}(\omega) = \frac{1}{\omega - \mathbf{\Lambda}},$$

where $\omega$ represents the frequency and $\mathbf{\Lambda}$ is the diagonal matrix of eigenvalues from the NPLR decomposition. The state update is efficiently computed in the frequency domain as:

$$\mathbf{x}(t) = \text{IFFT}\big(\text{FFT}(\mathbf{K}(\omega)) \odot \text{FFT}(\mathbf{x}(t))\big),$$

where $\odot$ denotes element-wise multiplication in the frequency domain. By leveraging FFT and IFFT operations, the model efficiently handles high-resolution temporal sequences and captures long-range dependencies while maintaining computational efficiency.

The SPLR Convolution Layer integrates three key innovations: 1. **Temporal Adaptation:** SA-HiPPO dynamically adjusts memory retention, capturing spike timing dependencies. 2. **Computational Efficiency:** NPLR Decomposition ensures scalability by reducing computational overhead. 3. **Scalable Convolution:** FFT-based convolution accelerates long-range temporal modeling.

**5. Normalization:** To maintain stability and ensure efficient learning, *Layer Normalization (LN)* is applied after each SPLR convolution layer:

$$\hat{\mathbf{x}}_l = \frac{\mathbf{x}_l - \mu_l}{\sqrt{\sigma_l^2 + \epsilon}} \cdot \gamma + \beta,$$

where $\mu_l$ and $\sigma_l^2$ are the mean and variance of activations at layer $l$, respectively, and $\gamma, \beta$ are learnable parameters. Normalization reduces variability in activations, providing stable training regardless of input fluctuations.

**6. Readout Layer:** The readout layer is inspired by the *Event-SSM* architecture and employs an *event-pooling mechanism* to subsample the temporal sequence length. The pooled output is given as:

$$\mathbf{x}_{\text{pooled},k} = \frac{1}{p} \sum_{i=kp}^{(k+1)p-1} \hat{\mathbf{x}}_i,$$

where $p$ is the pooling factor. This operation retains the most relevant temporal features, reducing computational burden while preserving key information. The resulting pooled sequence is passed through a linear transformation: $\mathbf{y} = \mathbf{W} \cdot \mathbf{x}_{\text{pooled}} + \mathbf{b}$, where $\mathbf{W}$ and $\mathbf{b}$ are learnable parameters. The combination of event pooling and linear transformation provides an efficient means for deriving a final representation suitable for downstream tasks, maintaining scalability with longer event sequences.

## 4 THEORETICAL DISCUSSION

In this section, we analyze the computational complexity, temporal dependency preservation, and stability of the SPLR model. We derive theoretical bounds and discuss how the model's components interact to ensure efficient processing and robust memory retention in SNNs. The detailed proofs of these theorems are given in Suppl Sec. 7)

**Lemma 1.** *(Computational Complexity of SPLR) Let the spike-driven SSM be given as:*

$$\dot{\mathbf{x}}(t) = \mathbf{A} \cdot \mathbf{x}(t) + \mathbf{B} \cdot \mathbf{S}(t),$$

*where $\mathbf{x}(t) \in \mathbb{R}^N$ is the internal state, $\mathbf{A} \in \mathbb{R}^{N \times N}$ is the state transition matrix, and $\mathbf{S}(t) \in \mathbb{R}^M$ is the input spike train. The computational complexity of updating the internal state $\mathbf{x}(t)$ at each spike event is $O(N^2)$.*

*Intuitive Explanation*: This result shows that the computational cost for updating the SPLR model at each spike event scales with the square of the state's dimensionality. By leveraging techniques like low-rank decomposition, reducing the matrix density makes computations more efficient.

**Theorem 1.** *(Long-Range Temporal Dependency Preservation via Spike-Aware HiPPO) Let $\mathbf{x}(t) \in \mathbb{R}^N$ evolve according to:*

$$\dot{\mathbf{x}}(t) = \mathbf{A} \cdot \mathbf{x}(t) + \mathbf{B} \cdot \mathbf{S}(t),$$

*where $\mathbf{A} \in \mathbb{R}^{N \times N}$ is a HiPPO matrix with all eigenvalues satisfying $Re(\lambda_i) < 0$ for $i = 1, \ldots, N$; $\mathbf{B} \in \mathbb{R}^{N \times M}$ is the input matrix; $\mathbf{S}(t) \in \mathbb{R}^M$ is a bounded input spike train, i.e., $\|\mathbf{S}(t)\| \le S_\infty$ for all $t \ge 0$; and $\mathbf{x}_0 = \mathbf{x}(0) \in \mathbb{R}^N$ is the initial state. Then, the SPLR preserves long-range temporal dependencies in $\mathbf{S}(t)$, and the state $\mathbf{x}(t)$ satisfies:*

$$\|\mathbf{x}(t)\| \le e^{-\alpha t} \|\mathbf{x}_0\| + \frac{\|\mathbf{B}\| S_\infty}{\alpha} \left(1 - e^{-\alpha t}\right),$$

*where $\alpha = \min_i |Re(\lambda_i)| > 0$ is the memory retention factor determined by $\mathbf{A}$.*

---

**Algorithm 1** SPLR Model Training

---

**Require:** Training dataset $\mathcal{D} = \{(\mathbf{X}_i, \mathbf{y}_i)\}_{i=1}^N$, learning rate $\eta$, total epochs $E$, threshold potential $V_{\text{th}}$, decay factors $\alpha_d, \beta$

1: Initialize weights $\mathbf{W}$, dendritic timing factors $\tau_d$, SPLR matrices $\mathbf{A}, \mathbf{B}, \mathbf{C}$, low-rank matrices $\mathbf{P}, \mathbf{Q}$, and kernel $\mathbf{K}(\omega)$
2: Initialize coupling strengths $\mathbf{g}_d$ for each dendrite $d$
3: **for** epoch = 1 to $E$ **do**
4:     **for** each $(\mathbf{X}, \mathbf{y}) \in \mathcal{D}$ **do**
    **Input Representation:** Prepare input events for processing
5:         Parse input event sequence $\mathbf{X} = \{(x_i, y_i, t_i, p_i)\}$, where $(x_i, y_i)$ are spatial coords, $t_i$ is time, $p_i$ is polarity.
    **Dendrite Attention Layer:** Update dendritic currents and aggregate at soma
6:         **for** each $t_i$ in spike event sequence **do**
7:             **for** each dendrite $d$ **do**
8:                 Update dendritic current: $\mathbf{i}_d(t_i + 1) = \alpha_d \cdot \mathbf{i}_d(t_i) + \sum_{j \in \mathcal{N}_d} \mathbf{w}_j \cdot p_j$
9:             **end for**
10:             Aggregate currents at soma: $V(t_i + 1) = \beta \cdot V(t_i) + \sum_d \mathbf{g}_d \cdot \mathbf{i}_d(t_i)$
11:             **if** $V(t_i + 1) > V_{\text{th}}$ **then**
12:                 Generate spike and reset potential: $V(t_i + 1) \leftarrow 0$
13:             **end if**
14:         **end for**
    **Spatial Pooling Layer:** Reduce spatial dimensionality while preserving temporal resolution
15:         Apply max pooling: $I_{\text{pooled}}(x', y', t) = \max_{(x,y) \in P(x', y')} I(x, y, t)$
    **SPLR Conv. Layer:** Apply SA-HiPPO, NPLR, & FFT for event dynamics
16:         Initialize state vector $\mathbf{x}(0)$
17:         **for** each spike time $t_k$ in $I_{\text{pooled}}$ **do**
18:             Compute $\Delta t_k = t_{k+1} - t_k$, decay $\mathbf{F}_{ij}(\Delta t_k) = e^{-\alpha_{ij} \cdot \Delta t_k}$
19:             Compute spike-aware HiPPO: $\mathbf{A_S} = \mathbf{A} \circ \mathbf{F}(\Delta t_k)$
20:             Decompose: $\mathbf{A_S} = \mathbf{V} \mathbf{\Lambda} \mathbf{V}^* - \mathbf{P} \mathbf{Q}^*$
21:             $e^{\mathbf{A_S} \Delta t_k} \approx \mathbf{I} + \mathbf{A_S} \Delta t_k + \dfrac{\mathbf{A_S}^2 (\Delta t_k)^2}{2}$
22:             Update: $\mathbf{x}(t_{k+1}) = e^{\mathbf{A_S} \Delta t_k} \cdot \mathbf{x}(t_k) + \mathbf{A_S}^{-1} (e^{\mathbf{A_S} \Delta t_k} - \mathbf{I}) \cdot \mathbf{B} \cdot \mathbf{S}(t_k)$
23:             FFT-based convolution: $\mathbf{x}(t_{k+1}) = \text{IFFT}(\text{FFT}(\mathbf{K}(\omega)) \odot \text{FFT}(\mathbf{x}(t_{k+1})))$
24:         **end for**
25:         Compute continuous output: $\mathbf{y}(t) = \mathbf{C} \cdot \mathbf{x}(t)$
26:         **Thresholding:** Convert $\mathbf{y}(t)$ to spikes by applying $y_{\text{spike}}(t) = \mathbb{I}(\mathbf{y}(t) > V_{\text{th}})$
    **Normalization:** Reduce variability in activations
27:         Apply layer normalization: $\hat{\mathbf{x}}_l = \dfrac{\mathbf{x}_l - \mu_l}{\sqrt{\sigma_l^2 + \epsilon}} \cdot \gamma + \beta$
    **Readout Layer:** Compute final output and update model parameters
28:         Compute pooled state: $\mathbf{x}_{\text{pooled}, k} = \frac{1}{p} \sum_{i=kp}^{(k+1)p-1} \hat{\mathbf{x}}_i$
29:         Final output: $\mathbf{y}_{\text{pred}} = \mathbf{W} \cdot \mathbf{x}_{\text{pooled}} + \mathbf{b}$
30:         Compute loss $\mathcal{L}(\mathbf{y}_{\text{pred}}, \mathbf{y})$, update $\mathbf{W} \leftarrow \mathbf{W} - \eta \cdot \frac{\partial \mathcal{L}}{\partial \mathbf{W}}$
31:     **end for**
32: **end for**

---

*Intuitive Explanation*: This theorem establishes that SPLR effectively retains temporal dependencies over time by controlling the decay of older information. This ensures that recent input spikes have a stronger influence on the state than older inputs, providing the model with long-range memory.

**Lemma 2.** *(Error Bound for Spike-Driven Matrix Exponential Approximation) Let the matrix exponential be approximated using a Taylor expansion up to the $n$-th term:*

$$e^{\mathbf{A} \Delta t} \approx \mathbf{I} + \mathbf{A} \Delta t + \frac{\mathbf{A}^2 \Delta t^2}{2!} + \cdots + \frac{\mathbf{A}^n \Delta t^n}{n!}.$$

*Assume that the matrix norm $\|\cdot\|$ is submultiplicative, i.e., $\|\mathbf{A}\cdot\mathbf{B}\| \leq \|\mathbf{A}\|\|\mathbf{B}\|$ for any compatible matrices $\mathbf{A}$ and $\mathbf{B}$. Then, the error $\mathbf{E}_n$ of this approximation satisfies:*

$$\|\mathbf{E}_n\| \leq \frac{\|\mathbf{A}\Delta t\|^{n+1}}{(n+1)!}.$$

*Intuitive Explanation*: This lemma provides a bound on the error when approximating the matrix exponential with a Taylor series. It helps balance computational efficiency with accuracy, showing that including more terms reduces the error. This is particularly useful for efficient, real-time state updates in spike-driven models.

**Theorem 2.** *(Boundedness of State Trajectories in the Presence of Spiking Inputs) For a given initial condition $\mathbf{x}_0$, the state trajectory $\mathbf{x}(t)$ of the SPLR model driven by the spike input $\mathbf{S}(t)$ is bounded, i.e., $\|\mathbf{x}(t)\| \leq C$, for some constant $C > 0$, provided that:*

1. *The input spikes $\mathbf{S}(t)$ are of finite magnitude, i.e., $\|\mathbf{S}(t)\| \leq S_\infty$ for all $t \geq 0$.*

2. *The decay matrix $\mathbf{A_S}$ is Hurwitz, meaning all its eigenvalues have negative real parts.*

3. *There exists a positive definite matrix $\mathbf{P}$ satisfying the Lyapunov equation $\mathbf{A_S}^T\mathbf{P} + \mathbf{P}\mathbf{A_S} = -\mathbf{Q}$, for some positive definite matrix $\mathbf{Q}$.*

*Intuitive Explanation*: This theorem guarantees that the SPLR model's state remains bounded over time when spike inputs are limited in magnitude. It ensures stability, meaning the state won't grow indefinitely, making the model reliable for continuous, real-time spike inputs.

## 5 EXPERIMENTS AND RESULTS

**Experimental Setup:** We evaluate the SPLR model on a variety of datasets to demonstrate its effectiveness in processing asynchronous, event-driven data. For all experiments, the SPLR model processes inputs on an event-by-event basis, dynamically updating its hidden state with each incoming spike. This approach preserves high temporal resolution and captures fine-grained spatio-temporal dependencies without accumulating events into frames. Below, we summarize the experimental setup for the primary datasets. Details for additional datasets, including Sequential CIFAR-10 and CIFAR-100 Krizhevsky et al. (2009), SHD, and SSC Cramer et al. (2020), is given in Suppl. Sec. 8.

*DVS Gesture Dataset Amir et al. (2017):* Contains event streams of 11 hand gestures from 29 subjects recorded with a Dynamic Vision Sensor. The SPLR model processes real-time spikes, capturing temporal dynamics for accurate gesture classification.

*HAR-DVS Dataset Wang et al. (2024b):* Comprises event streams of six human activities, including walking and running, with spatial coordinates, timestamps, and polarity. SPLR dynamically handles these sparse streams to enable real-time classification of complex activities.

*Celex-HAR Dataset Wang et al. (2024a):* Utilizes high-resolution CeleX event streams of actions such as sitting and walking. The SPLR model updates its state with each event, effectively modeling fine-grained temporal structures.

*Long Range Arena (LRA) Tay et al. (2020):* Serves as a benchmark for long-range dependency modeling. Tasks like ListOps and Path-X are transformed into event-driven formats, with SPLR sequentially processing tokens to capture extended temporal dependencies.

**Long-Range Dependencies:** We evaluate the ability of the proposed *SPLR* model to capture long-range dependencies using the **Long Range Arena (LRA)** dataset Tay et al. (2020). The LRA benchmark evaluates models on tasks requiring long-context understanding, where Transformer-based non-spiking models often exhibit suboptimal performance due to the computational overhead of attention mechanisms, which scales poorly with increasing sequence lengths. As shown in Table 3, we benchmark our method against state-of-the-art alternatives, including the LMU-based spiking model, SpikingLMUFormer Liu et al. (2024b), and the BinaryS4D model Stan & Rhodes (2024). While BinaryS4D is not fully spiking—it relies on floating-point MAC operations for matrix multiplications—it incorporates LIF neurons to spike from an underlying SSM, providing a hybrid approach to handling long-range dependencies.

Table 3: Results comparing the accuracy of our model against some spiking and non-spiking architectures on test sets of LRA benchmark tasks.

| Model | SNN | ListOps | Text | Retrieval | Image | Pathfinder |
|---|---|---|---|---|---|---|
| S4 (Original) Gu et al. (2022) | No | 58.35 | 76.02 | 87.09 | 87.26 | 86.05 |
| S4 (Improved) Gu et al. (2022) | No | 59.60 | 86.82 | 90.90 | 88.65 | 94.20 |
| Transformer Vaswani et al. (2017) | No | 36.37 | 64.27 | 57.46 | 42.44 | 71.40 |
| Sparse Transformer Tay et al. (2020) | No | 17.07 | 63.58 | 59.59 | 44.24 | 71.71 |
| Linformer Wang et al. (2020) | No | 35.70 | 53.94 | 52.27 | 38.56 | 76.34 |
| Linear Transformer Tay et al. (2020) | No | 16.13 | 65.90 | 53.09 | 42.34 | 75.30 |
| FLASH-quad Hua et al. (2022) | No | 42.20 | 64.10 | 83.00 | 48.30 | 83.62 |
| Spiking LMUFormer Liu et al. (2024b) | Yes | 37.30 | 65.80 | 79.76 | 55.65 | 72.68 |
| TransNormer T2 Qin et al. (2022) | No | 41.60 | 72.20 | 83.82 | 49.60 | 76.60 |
| BinaryS4D Stan & Rhodes (2024) | Partial | 54.80 | 82.50 | 85.30 | 82.00 | 82.60 |
| **SPLR (Our Model)** | **Yes** | **59.08** | **79.41** | **89.62** | **79.88** | **86.47** |

**Event Dataset Results:** Figure 3(a) presents the performance of our proposed SPLR models on the DVS Gesture 128 dataset, comparing accuracy versus number of parameters with other state-of-the-art models. We evaluated three variants of SPLR—Tiny, Small, and Normal—each designed to understand scalability and efficiency (Details of model architectures given in Suppl. Sec. 9). The SPLR Normal variant achieved an accuracy of 96.5%, effectively capturing the complex temporal dependencies in event-driven tasks. SPLR Small and SPLR Tiny also demonstrated competitive performance with accuracies of 93.7% and 89.2%, respectively, maintaining a balance between reduced parameter count and performance. Compared to other architectures like EventMamba Ren et al. (2024), TBR+I3D Innocenti et al. (2021), and PointNet++ Qi et al. (2017), our SPLR variants consistently showed a favorable trade-off between model complexity and accuracy. Notably, SPLR Normal matched or even exceeded the performance of larger CNN and ViT models, such as Event Frames + I3DBi et al. (2020) and RG-CNN Miao et al. (2019), with significantly fewer parameters, emphasizing its efficiency. We conducted an ablation study to evaluate the contribution of specific architectural components in the SPLR models, focusing on the Dendrite Attention Layer and the SA-HiPPO matrix. Removing the dendrite mechanism led to a significant drop in accuracy across all variants, with SPLR Normal reducing to 95.2%. Similarly, replacing SA-HiPPO with standard LIF neurons further reduced accuracy to 90.4%, indi-

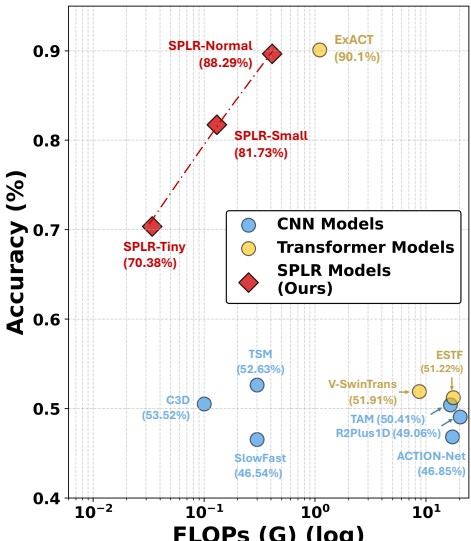

Figure 2: Accuracy vs. FLOPS (G) on HAR-DVS DatasetWang et al. (2024b) comparing SPLR variants with other SOTA models.

cating the crucial role of SA-HiPPO in maintaining long-range temporal dependencies. (Complete results are shown in Table 8 in Suppl. Sec 8). The dashed lines in Figure 3(a) illustrate the impact of these architectural components, demonstrating the critical contribution of both Dendrite Attention and SA-HiPPO in achieving high accuracy. These results highlight the importance of each component in enabling efficient spatiotemporal learning, allowing SPLR models to outperform other methods while maintaining fewer parameters.

We also evaluate the effectiveness of *dendritic mechanisms* combined with *SPLR convolutions* across SHD, SSC, and DVS Gesture datasets as detailed in Tables 7, 8 (Suppl. Sec. 8). The SSC dataset, requiring the capture of long-range temporal dependencies, proves to be more challenging than SHD and DVS Gesture. Figure 5 demonstrate that SPLR's performance gains are most pronounced in SSC, underscoring its capability in handling complex temporal patterns. Moreover, incorporating dendritic attention consistently enhances accuracy across all datasets, especially when using fewer channels.

**Scaling to HD Event Streams:** To evaluate the scalability of the proposed *SPLR* model, we utilized the *Celex HAR* datasetWang et al. (2024a), a high-resolution human activity recognition benchmark (1280 × 800). This dataset presents significant challenges in maintaining accuracy and efficiency with large-scale spatial and temporal data. As shown in Figure 3(b), *SPLR* achieves superior accuracy compared to baseline SNNs and DNNs, maintaining high performance even at increased

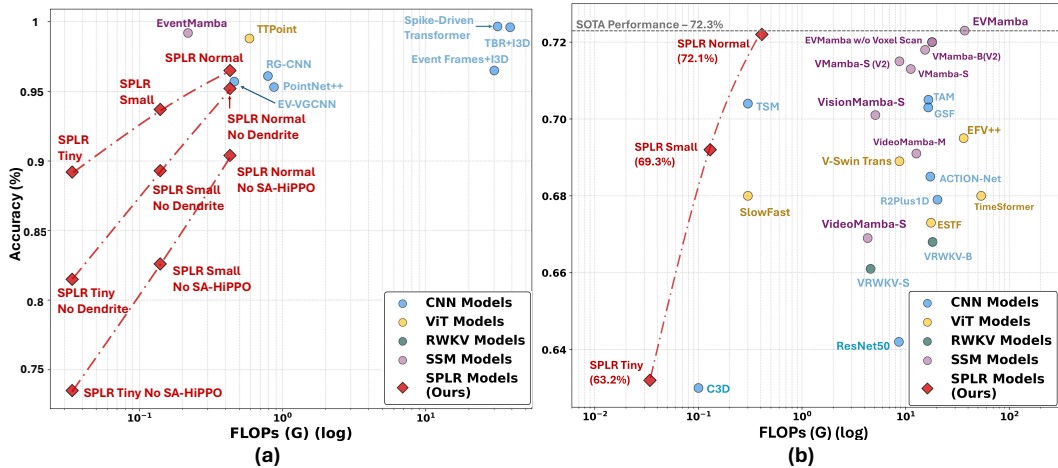

Figure 3: Accuracy vs. FLOPS (G) on (a) *DVSGesture128* and (b) *Celex-HAR* datasets comparing SPLR variants with other SOTA models. Figure (a) shows the ablation studies showing the impact of removing the Dendrite Attention Layer or replacing SA-HiPPO with standard LIF neurons. Note: There are no spike-based designs for Celex-HAR

resolutions where other methods struggle. The *SPLR convolution layer* effectively manages both spatial and temporal complexities, enabling real-time processing of HD event streams with minimal computational overhead. **To our knowledge, no prior work has demonstrated results on Celex HAR using spiking-based models.** Figure 3(b) also illustrates the trade-off between accuracy and computational cost (FLOPs), with *SPLR Tiny*, *Small*, and *Normal* achieving competitive or better accuracy compared to models like *SlowFast* Feichtenhofer et al. (2019) and *C3D* Tran et al. (2015), but with significantly lower computational requirements. *SPLR Normal* exceeds the performance of *TSM* Lin et al. (2019) and *VisionMamba-S* Zhu et al. (2024) at a reduced cost, highlighting the efficiency of the event-driven state-space approach.

**HAR-DVS:** We also evaluated on the HAR-DVS dataset Wang et al. (2024b) (Fig. 2). We see that our SPLR models outperform other state-of-the-art DNN models. Unlike frame-based methods, SPLR employs event-by-event processing to preserve temporal dynamics and introduces a novel dendritic attention mechanism, enabling efficient and robust spatio-temporal modeling. This makes SPLR particularly well-suited for real-time event-driven applications. [See Suppl. Sec. 8]

## 6 CONCLUSION

This work presents the **SPLR** model, which integrates the novel **SA-HiPPO** mechanism with fully event-driven processing to overcome the limitations of existing approaches in SNNs. By dynamically adapting memory retention to inter-spike intervals, SA-HiPPO enables precise modeling of long-range dependencies while preserving the sparsity and efficiency inherent to SNNs. Empirical evaluations highlight SPLR's superior performance across a range of benchmarks. On the **Long Range Arena (LRA)**, SPLR demonstrates significantly higher accuracy than methods like BinaryS4D and SpikingLMU, achieving state-of-the-art results with lower computational cost compared to dense Transformer-based architectures. On real-world event datasets such as **DVS Gesture** and **HAR-DVS**, SPLR leverages its efficient spatio-temporal feature extraction to outperform other state-of-the-art models like EventMamba. Furthermore, SPLR scales effectively on high-resolution benchmarks such as **Celex-HAR**, maintaining high performance under increased spatial and temporal complexities where traditional methods degrade.

The entire SPLR pipeline, including components such as the Dendrite Attention Layer, Spatial Pooling, and SPLR Convolution, is critical for enabling the precise modeling of long-range dependencies in event-driven systems. Each component plays a complementary role in achieving robust, scalable, and efficient processing. The key novelty lies in the formulation of the **SA-HiPPO** mechanism, which addresses a major challenge in the field by overcoming scalability limitations in SNNs while preserving their asynchronous, low-latency nature. By preserving the fully asynchronous, event-driven nature of SNNs, SPLR achieves a transformative balance of scalability, low latency, and computational efficiency. These results establish SPLR as a robust and scalable solution for neuromorphic computing, unlocking new capabilities for long-range dependency modeling and real-time event-driven systems.

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

CONTENTS

# 7 SUPPLEMENTARY SECTION A: DETAILED PROOFS

## 7.1 COMPUTATIONAL COMPLEXITY OF SPIKE-DRIVEN SSMs

**Lemma 3.** *Let the spike-driven state-space model be governed by:*

$$\dot{x}(t) = Ax(t) + BS(t),$$

*where $x(t) \in \mathbb{R}^N$ is the internal state, $A \in \mathbb{R}^{N \times N}$ is the state transition matrix, and $S(t) \in \mathbb{R}^M$ is the input spike train. The computational complexity of updating the internal state $x(t)$ at each spike event is $O(N^2)$.*

*Proof.* The spike-driven state-space model is governed by:

$$\dot{x}(t) = Ax(t) + BS(t),$$

where $x(t) \in \mathbb{R}^N$ represents the internal state of the system, $A \in \mathbb{R}^{N \times N}$ is the state transition matrix, and $S(t) \in \mathbb{R}^M$ represents the input spike train. When a spike event occurs at time $t_i$, the state update can be represented by the following integral equation for $t \in [t_i, t_{i+1})$:

$$x(t_i^+) = e^{A \Delta t_i} x(t_i^-) + \int_{t_i^-}^{t_i^+} e^{A(t_i^+ - \tau)} BS(\tau)\, d\tau,$$

where:

- $t_i^-$ and $t_i^+$ are the times just before and after the spike at $t_i$, - $\Delta t_i = t_i^+ - t_i^-$ is infinitesimal, - $S(\tau)$ contains Dirac delta functions at spike times and is zero elsewhere.

For simplicity, we focus on the update at the spike time $t_i$ to approximate the state transition at each event.

The update of the internal state $x(t)$ requires computing the matrix exponential $e^{A \Delta t}$, where $\Delta t = t - t_i$ represents the time interval between successive spikes. Computing the exact matrix exponential for a general matrix $A \in \mathbb{R}^{N \times N}$ is computationally expensive, involving $O(N^3)$ operations using standard algorithms such as diagonalization or the Schur decomposition.

To reduce the computational cost, we approximate the matrix exponential using a truncated Taylor series expansion:

$$e^{A \Delta t_i} \approx I + A \Delta t_i + \frac{1}{2} A^2 \Delta t_i^2.$$

where $I$ is the identity matrix of size $N \times N$. This approximation is typically sufficient for small $\Delta t$, which is common between spike events.

In the Taylor series expansion approximation of $e^{A \Delta t}$, the dominant computational cost arises from multiplying the matrix $A \in \mathbb{R}^{N \times N}$ by itself and by the state vector $x(t) \in \mathbb{R}^N$.

The product $Ax(t)$, where $A \in \mathbb{R}^{N \times N}$ and $x(t) \in \mathbb{R}^N$, requires $N^2$ multiplications. Thus, the computational cost for this step is $O(N^2)$.

The term $A^2$ is computed by multiplying $A$ by itself. Since $A$ is an $N \times N$ matrix, computing $A^2$ explicitly would have a computational cost of $O(N^3)$. However, we avoid this by computing $A(Ax(t))$, which involves two sequential matrix-vector products, each costing $O(N^2)$. Therefore, the computational cost of computing $A^2 x(t)$ is $O(N^2)$.

The term $BS(t)$, where $B \in \mathbb{R}^{N \times M}$ and $S(t) \in \mathbb{R}^M$, involves $O(NM)$ operations. Assuming $M$ is proportional to $N$ or smaller, this computation contributes $O(N^2)$ to the overall complexity.

To update the internal state $x(t)$, we perform the following operations: First, we multiply $A$ by $x(t)$: $O(N^2)$; then multiply $A^2$ by $x(t)$: $O(N^2)$; followed by multiplying $B$ by $S(t)$: $O(NM)$ and finally add the resulting vectors.

Thus, the overall computational complexity for updating the internal state $x(t)$ at each spike event is $O(N^2)$.

In the general case, where $A$ is a dense matrix, the cost of updating the state is $O(N^2)$. If the matrix $A$ has a specific structure, such as being sparse or block-diagonal, the computational cost can be

reduced. For example: - If $A$ is sparse with $k$ non-zero entries per row, the cost of multiplying $A$ by $x(t)$ becomes $O(kN)$, which can be significantly lower than $O(N^2)$ when $k \ll N$. - If $A$ is block-diagonal, the cost can be reduced to $O(N)$ per block, depending on the number and size of the blocks. However, for the general case where no such structure is assumed, the computational complexity remains $O(N^2)$. The computational complexity of updating the internal state $x(t)$ at each spike event, using the matrix exponential approximation with a Taylor series expansion, is dominated by the matrix-vector multiplication operations. Additionally, accounting for the $BS(t)$ term maintains the overall complexity at $O(N^2)$. Therefore, the overall computational complexity for updating the internal state at each spike event is $O(N^2)$.

$\square$

## 7.2 LONG-RANGE TEMPORAL DEPENDENCY PRESERVATION VIA SPIKE-BASED HIPPO

**Theorem 3.** *Let $x(t) \in \mathbb{R}^N$ evolve according to*

$$\dot{x}(t) = Ax(t) + BS(t),$$

*where: - $A \in \mathbb{R}^{N \times N}$ is a HiPPO matrix with all eigenvalues satisfying $Re(\lambda_i) < 0$ for $i = 1, 2, \ldots, N$, - $B \in \mathbb{R}^{N \times M}$ is the input matrix, - $S(t) \in \mathbb{R}^M$ is the input spike train, assumed to be bounded, i.e., there exists a constant $S_\infty > 0$ such that $\|S(t)\| \le S_\infty$ for all $t \ge 0$, - $x_0 = x(0) \in \mathbb{R}^N$ is the initial state.*

*Then, the spike-driven SSM preserves long-range temporal dependencies in the input spike train $S(t)$, and the state $x(t)$ satisfies the bound:*

$$\|x(t)\| \le e^{-\alpha t}\|x_0\| + \frac{\|B\|S_\infty}{\alpha}\left(1 - e^{-\alpha t}\right),$$

*where $\alpha = \min_i |Re(\lambda_i)| > 0$ is the memory retention factor determined by the eigenvalues of the HiPPO matrix $A$.*

*Proof.* To establish the theorem, we will analyze the evolution of the internal state $x(t)$ governed by the differential equation:

$$\dot{x}(t) = Ax(t) + BS(t),$$

with initial condition $x(0) = x_0$.

The differential equation is a non-homogeneous linear ordinary differential equation (ODE). Using the variation of parameters method, the solution can be expressed as:

$$x(t) = e^{At}x_0 + \int_0^t e^{A(t-\tau)}BS(\tau)\, d\tau,$$

where: - $e^{At}x_0$ is the solution to the homogeneous equation $\dot{x}(t) = Ax(t)$ with initial condition $x(0) = x_0$, - $\int_0^t e^{A(t-\tau)}BS(\tau)\, d\tau$ accounts for the particular solution due to the input $S(t)$.

Given that $A$ is a HiPPO matrix, all its eigenvalues satisfy $Re(\lambda_i) < 0$ for $i = 1, 2, \ldots, N$. This implies that $A$ is a Hurwitz matrix, ensuring that the system is asymptotically stable. Define the memory retention factor $\alpha$ as:

$$\alpha = \min_i |Re(\lambda_i)| > 0.$$

This factor dictates the rate at which the influence of the initial state $x_0$ decays over time.

Consider the homogeneous solution $e^{At}x_0$. Since all eigenvalues of $A$ have negative real parts, the matrix exponential $e^{At}$ satisfies:

$$\|e^{At}\| \le e^{-\alpha t},$$

where $\|\cdot\|$ denotes an operator norm (e.g., the induced 2-norm). This inequality leverages the spectral bound of $A$ to provide an exponential decay rate.

Therefore, the contribution of the initial state is bounded by:

$$\|e^{At}x_0\| \le \|e^{At}\| \cdot \|x_0\| \le e^{-\alpha t}\|x_0\|.$$

Next, consider the particular solution:

$$\int_0^t e^{A(t-\tau)} B S(\tau)\, d\tau.$$

To bound its norm, apply the triangle inequality and properties of operator norms:

$$\left\| \int_0^t e^{A(t-\tau)} B S(\tau)\, d\tau \right\| \le \int_0^t \|e^{A(t-\tau)}\| \cdot \|B\| \cdot \|S(\tau)\|\, d\tau.$$

Given that $\|S(\tau)\| \le S_\infty$ and $\|e^{A(t-\tau)}\| \le e^{-\alpha(t-\tau)}$, we have:

$$\left\| \int_0^t e^{A(t-\tau)} B S(\tau)\, d\tau \right\| \le \|B\| S_\infty \int_0^t e^{-\alpha(t-\tau)}\, d\tau.$$

Evaluate the integral:

$$\int_0^t e^{-\alpha(t-\tau)}\, d\tau = \int_0^t e^{-\alpha s}\, ds = \frac{1 - e^{-\alpha t}}{\alpha}.$$

Thus, the bound becomes:

$$\left\| \int_0^t e^{A(t-\tau)} B S(\tau)\, d\tau \right\| \le \frac{\|B\| S_\infty}{\alpha} \left(1 - e^{-\alpha t}\right).$$

Combining the bounds for the homogeneous and particular solutions, we obtain:

$$\|x(t)\| \le \|e^{At} x_0\| + \left\| \int_0^t e^{A(t-\tau)} B S(\tau)\, d\tau \right\| \le e^{-\alpha t} \|x_0\| + \frac{\|B\| S_\infty}{\alpha} \left(1 - e^{-\alpha t}\right).$$

This inequality demonstrates that: - The influence of the initial state $x_0$ decays exponentially at rate $\alpha$, - The accumulated influence of the input spike train $S(t)$ is bounded and grows to a steady-state value determined by $\|B\|$, $S_\infty$, and $\alpha$.

The derived bound:

$$\|x(t)\| \le e^{-\alpha t} \|x_0\| + \frac{\|B\| S_\infty}{\alpha} \left(1 - e^{-\alpha t}\right),$$

reveals that the term $e^{-\alpha t} \|x_0\|$ signifies that the system "forgets" its initial state exponentially fast, ensuring that old information does not dominate the state indefinitely. Also, the integral term captures the accumulated influence of the input spike train $S(t)$. Since $S(t)$ is bounded, the state $x(t)$ can retain and reflect information from the input over extended periods without being overwhelmed by the initial condition.

Therefore, the spike-driven SSM governed by a HiPPO matrix $A$ effectively preserves long-range temporal dependencies in the input spike train $S(t)$, while ensuring that the memory of the initial state $x_0$ decays at an exponential rate determined by $\alpha$.

$\square$

## 7.3 ERROR BOUND FOR SPIKE-DRIVEN MATRIX EXPONENTIAL APPROXIMATION

**Lemma 4.** *Let the matrix exponential be approximated using a Taylor expansion up to the $n$-th term:*

$$e^{A\Delta t} \approx I + A\Delta t + \frac{A^2 \Delta t^2}{2!} + \cdots + \frac{A^n \Delta t^n}{n!}.$$

*Assume that the matrix norm $\|\cdot\|$ is submultiplicative, i.e., $\|AB\| \le \|A\|\|B\|$ for any matrices $A$ and $B$ of compatible dimensions. Then, the error $E_n$ of this approximation satisfies*

$$\|E_n\| \le \frac{\|A\Delta t\|^{n+1}}{(n+1)!} e^{\|A\Delta t\|}.$$

*Proof.* The matrix exponential can be expressed as an infinite Taylor series:

$$e^{A\Delta t} = \sum_{k=0}^{\infty} \frac{(A\Delta t)^k}{k!}.$$

If we truncate this series after the $n$-th term, the remainder $E_n$ is given by:

$$E_n = e^{A\Delta t} - \sum_{k=0}^{n} \frac{(A\Delta t)^k}{k!} = \sum_{k=n+1}^{\infty} \frac{(A\Delta t)^k}{k!}.$$

To bound the norm of the error $E_n$, we apply the submultiplicative property of the matrix norm:

$$\|E_n\| = \left\| \sum_{k=n+1}^{\infty} \frac{(A\Delta t)^k}{k!} \right\| \le \sum_{k=n+1}^{\infty} \frac{\|A\Delta t\|^k}{k!}.$$

Using the submultiplicative property of the matrix norm:

$$\|E_n\| \le \sum_{k=n+1}^{\infty} \frac{\|A\Delta t\|^k}{k!}.$$

Let $x = \|A\Delta t\| \ge 0$. Then:

$$\|E_n\| \le \sum_{k=n+1}^{\infty} \frac{x^k}{k!}.$$

Since

$$\sum_{k=n+1}^{\infty} \frac{x^k}{k!} = e^x - \sum_{k=0}^{n} \frac{x^k}{k!} = R_n(x),$$

where $R_n(x)$ is the remainder of the Taylor series expansion of $e^x$.

According to Taylor's Remainder Theorem (Lagrange's form), there exists $\xi \in [0, x]$ such that:

$$R_n(x) = \frac{x^{n+1}}{(n+1)!} e^{\xi}.$$

Since $\xi \le x$ and $e^{\xi} \le e^x$ for $x \ge 0$, we have:

$$R_n(x) \le \frac{x^{n+1}}{(n+1)!} e^x.$$

Therefore:

$$\|E_n\| \le \frac{x^{n+1}}{(n+1)!} e^x = \frac{\|A\Delta t\|^{n+1}}{(n+1)!} e^{\|A\Delta t\|}.$$

Thus, the error $E_n$ satisfies:

$$\|E_n\| \le \frac{\|A\Delta t\|^{n+1}}{(n+1)!} e^{\|A\Delta t\|}.$$

$\square$

——

## 7.4 BOUNDEDNESS OF STATE TRAJECTORIES IN THE PRESENCE OF SPIKING INPUTS

**Theorem 4.** *Boundedness of State Trajectory in Spike-Driven State-Space Models*

*For a given initial condition $x_0$, the state trajectory $x(t)$ of the SPLR model driven by the spike input $S(t)$ is bounded, i.e., $\|x(t)\| \leq C$, for some constant $C > 0$, provided that:*

1. *The input spikes $S(t)$ are of finite magnitude, i.e., $\|S(t)\| \leq S_\infty$ for all $t \geq 0$.*

2. *The decay matrix $A_S$ is Hurwitz, meaning all its eigenvalues have negative real parts.*

3. *There exists a positive definite matrix $P$ satisfying the Lyapunov equation $A_S^T P + P A_S = -Q$, for some positive definite matrix $Q$.*

*Proof.* Consider the SPLR governed by:

$$\dot{x}(t) = A_S x(t) + B S(t),$$

where $A_S$ is a Hurwitz matrix, $B$ is the input matrix, and $S(t)$ is a bounded input spike train with $\|S(t)\| \leq S_\infty$ for all $t \geq 0$.

We define a Lyapunov function $V(x) = x^T P x$, where $P$ is a positive definite matrix satisfying the Lyapunov equation:

$$A_S^T P + P A_S = -Q,$$

with $Q$ being a positive definite matrix. Such a $P$ exists because $A_S$ is Hurwitz. The derivative of $V(x)$ along the system trajectories is computed:

$$\dot{V}(x) = \frac{d}{dt}(x^T P x) = x^T \dot{P} x + x^T P \dot{x} + \dot{x}^T P x.$$

Since $P$ is constant ($\dot{P} = 0$), and $\dot{x} = A_S x + B S(t)$, this simplifies to:

$$\dot{V}(x) = x^T P(A_S x + B S(t)) + (A_S x + B S(t))^T P x.$$

Recognizing that $P$ is symmetric ($P^T = P$), we can write:

$$\dot{V}(x) = x^T (A_S^T P + P A_S) x + 2 x^T P B S(t).$$

Substituting the Lyapunov equation $A_S^T P + P A_S = -Q$:

$$\dot{V}(x) = -x^T Q x + 2 x^T P B S(t).$$

The term $2 x^T P B S(t)$ is bounded using the Cauchy-Schwarz inequality as

$$2 x^T P B S(t) \leq 2 \|x\| \cdot \|PB\| \cdot \|S(t)\| \leq 2 \|PB\| S_\infty \|x\|.$$

Next, let us define $\gamma = 2 \|PB\| S_\infty$ The derivative $\dot{V}(x)$ becomes:

$$\dot{V}(x) \leq -x^T Q x + \gamma \|x\|.$$

Since $Q$ is positive definite, $x^T Q x \geq \lambda_{\min}(Q) \|x\|^2$, where $\lambda_{\min}(Q)$ is the smallest eigenvalue of $Q$. Therefore:

$$\dot{V}(x) \leq -\lambda_{\min}(Q) \|x\|^2 + \gamma \|x\|.$$

Completing the square:

$$\dot{V}(x) \leq -\lambda_{\min}(Q) \left( \|x\|^2 - \frac{\gamma}{\lambda_{\min}(Q)} \|x\| \right) = -\lambda_{\min}(Q) \left( \|x\| - \frac{\gamma}{2\lambda_{\min}(Q)} \right)^2 + \frac{\gamma^2}{4\lambda_{\min}(Q)}.$$

This inequality indicates that $\dot{V}(x) < 0$ whenever $\|x\| > \frac{\gamma}{2\lambda_{\min}(Q)}$. Since $V(x) \geq 0$ and $\dot{V}(x)$ is negative outside a ball of radius $C = \frac{\gamma}{2\lambda_{\min}(Q)}$, the state $x(t)$ will ultimately remain within this bounded region. Therefore, $\|x(t)\| \leq C$ for all $t \geq 0$

$\square$

# 8    SUPPLEMENTARY SECTION B: EXTENDED EXPERIMENTAL RESULTS

## 8.1    DATASETS AND TASKS

In this study, we evaluate the performance of the SPLR model across a diverse set of datasets, each presenting unique challenges in event-driven processing. The datasets include Sequential CIFAR-10, Sequential CIFAR-100 Krizhevsky et al. (2009), DVS Gesture Amir et al. (2017), HAR-DVS Wang et al. (2024b), Celex-HAR Wang et al. (2024a), Long Range Arena (LRA) Tay et al. (2020), Spiking Heidelberg Digits (SHD) Cramer et al. (2020), and Spiking Speech Commands (SSC). For all experiments, the SPLR model processes inputs on an event-by-event basis, leveraging its temporal dynamics to handle fine-grained temporal dependencies without accumulating events into frames. Below, we provide detailed descriptions of each dataset and the corresponding experimental setups.

**Sequential CIFAR-10 and CIFAR-100**: The CIFAR-10 and CIFAR-100 datasets Krizhevsky et al. (2009) consist of $32 \times 32$ RGB images across 10 and 100 classes, respectively. To simulate a temporal sequence, each image is divided into 16 non-overlapping patches of size $8 \times 8$ pixels. These patches are presented to the model sequentially in a raster-scan order, from top-left to bottom-right. Each patch is treated as an independent event in the sequence. The task involves classifying the image based on the full sequence of patches, requiring the model to integrate information over the entire sequence. This setup evaluates the model's ability to process spatial information in a temporal context.

**DVS Gesture Dataset**: The DVS Gesture dataset Amir et al. (2017) comprises recordings from a Dynamic Vision Sensor (DVS), capturing 11 hand gestures performed by 29 subjects under varying lighting conditions. Each event is characterized by its spatial location $(x, y)$, timestamp $t$, and polarity $p$ (on/off). The dataset provides a challenging benchmark for models to recognize dynamic gestures from sparse, asynchronous event streams. In our experiments, the SPLR model processes each event individually as it occurs, without accumulating them into temporal frames, thereby maintaining high temporal resolution and reducing latency.

**HAR-DVS Dataset**: The HAR-DVS dataset Wang et al. (2024b) contains neuromorphic event streams representing human activities, recorded with a DVS. Activities include walking, running, and other movement-based tasks. Each event is defined by its spatial coordinates, timestamp, and polarity. The dataset tests the model's ability to recognize complex human activities from sparse event streams. The SPLR model processes each spike event-by-event, dynamically updating its internal state for each incoming spike, enabling precise temporal modeling of the activity sequences.

**Celex-HAR Dataset**: The Celex-HAR dataset Wang et al. (2024a) consists of high-resolution event streams captured with a CeleX camera for human activity recognition. Activities include actions such as sitting, standing, and walking. Each event is represented by its spatial coordinates, timestamps, and polarity. The dataset provides a comprehensive benchmark for evaluating models on high-resolution event-based data. The SPLR model processes each spike event-by-event, allowing it to capture the fine-grained temporal dynamics of human activities.

**Long Range Arena (LRA)**: The Long Range Arena benchmark Tay et al. (2020) evaluates a model's ability to process long sequences and capture dependencies over extended temporal horizons. Tasks such as ListOps and Path-X involve sequence lengths ranging from hundreds to thousands of tokens. Although these tasks involve discrete tokens rather than spikes, we simulate event-driven processing by treating each token as an individual event presented sequentially. The SPLR model leverages its temporal dynamics to capture long-range dependencies efficiently.

**Spiking Heidelberg Digits (SHD) and Spiking Speech Commands (SSC)**: The SHD and SSC datasets Cramer et al. (2020) are benchmarks for spiking neural networks, containing neuromorphic spike streams derived from speech datasets. SHD consists of spoken digit recordings converted to spike trains using the CochleaAMS model, while SSC contains spiking representations of spoken command audio, representing keywords like "yes," "no," and "stop." Each event is characterized by its spatial location, timestamp, and polarity. The datasets evaluate the model's performance on tasks involving complex spatio-temporal patterns in speech data. The SPLR model processes each spike event as it occurs, dynamically updating its state, ensuring high temporal resolution and efficient processing for speech recognition tasks.

Across all datasets, the SPLR model processes inputs on an event-by-event basis. This approach allows it to maintain high temporal resolution and capture fine-grained spatio-temporal patterns,

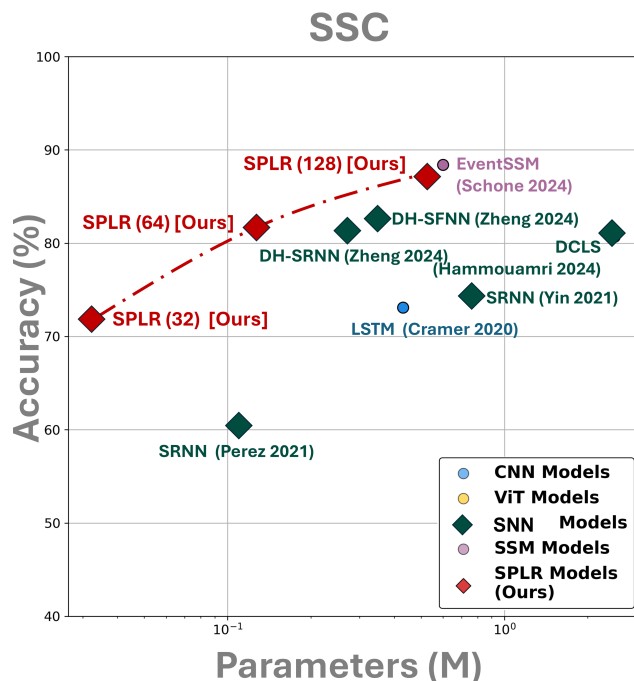

Figure 4: Figure showing the performance comparison of different state of the art models wrt the SPLR models on the SSC dataset Cramer et al. (2020)

distinguishing it from frame-based methods. The event-by-event design also reduces computational overhead and ensures low latency, making the model well-suited for real-time applications.

Table 4: Comparison of SPLR models with state-of-the-art on the HARDVS dataset. Accuracy is measured in percentage, and computational cost is in GFLOPs.

| Model | GFLOPs | Accuracy (%) |
|---|---|---|
| C3D Tran et al. (2015) | 0.1 | 50.52 |
| R2Plus1D Tran et al. (2018) | 20.3 | 49.06 |
| TSM Lin et al. (2019) | 0.3 | 52.63 |
| ACTION-Net Wang et al. (2021) | 17.3 | 46.85 |
| TAM Liu et al. (2021) | 16.6 | 50.41 |
| V-SwinTrans Liu et al. (2022c) | 8.7 | 51.91 |
| SlowFast Feichtenhofer et al. (2019) | 0.3 | 46.54 |
| ESTF Wang et al. (2024b) | 17.6 | 51.22 |
| ExACT Zhou et al. (2024) | 1.3 | 90.10 |
| **SPLR-Tiny** *[Ours]* | **0.034** | **65.42** |
| **SPLR-Small** *[Ours]* | **0.13** | **79.36** |
| **SPLR-Normal** *[Ours]* | **0.41** | **88.29** |

Table 5: Detailed Architecture of SPLR Models (Tiny, Small, and Normal)

| Layer Type | SPLR Tiny | SPLR Small | SPLR Normal |
|---|---|---|---|
| **Input Representation** | Asynchronous Spike Events $(x, y, t, p)$ | | |
| **Dendrite Attention Layer** | 16 dendritic branches $\tau_d = [\tau_1, \ldots, \tau_{16}]$ | 32 dendritic branches $\tau_d = [\tau_1, \ldots, \tau_{32}]$ | 64 dendritic branches $\tau_d = [\tau_1, \ldots, \tau_{64}]$ |
| **Convolutional Block 1** | Conv2D (32 filters, 3x3) Batch Norm, Max Pool (2x2) | Conv2D (64 filters, 3x3) Batch Norm, Max Pool (2x2) | Conv2D (128 filters, 3x3) Batch Norm, Max Pool (2x2) |
| **Convolutional Block 2** | Conv2D (32 filters, 3x3) Batch Norm, Max Pool (2x2) | Conv2D (64 filters, 3x3) Batch Norm, Max Pool (2x2) | Conv2D (128 filters, 3x3) Batch Norm, Max Pool (2x2) |
| **Spatial Pooling Layer** | Pool (2x2) | Pool (2x2) | Pool (2x2) |
| **SPLR Convolution** | State Update using Spike-Aware HiPPO and NPLR decomposition for efficient event-driven convolution | | |
| **Normalization Layer** | Layer Norm | Layer Norm | Layer Norm |
| | Normalizes the state variables to stabilize training | | |
| **Readout Layer** | Fully Connected (256 neurons) Softmax for classification | Fully Connected (512 neurons) Softmax for classification | Fully Connected (1024 neurons) Softmax for classification |

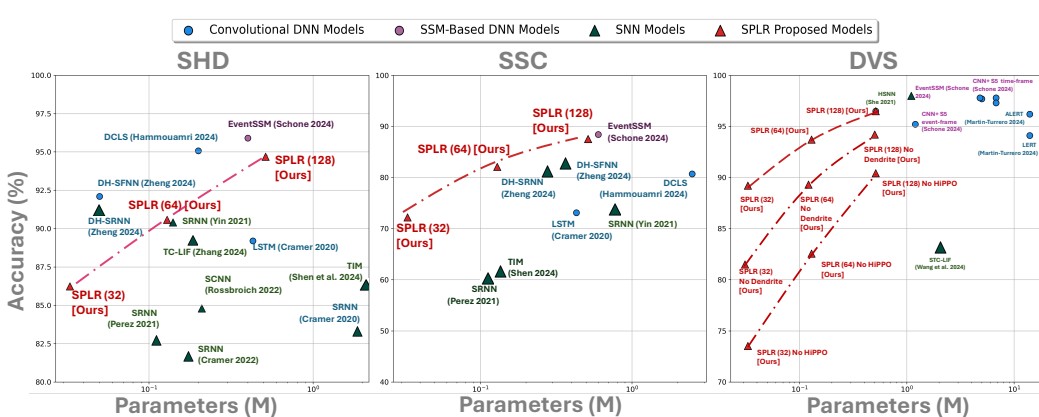

Figure 5: Comparison of our SPLR to the state-of-the-art on DVS128-GestureAmir et al. (2017), Spiking Heidelberg digits (SHD) and Spiking Speech Commands (SSC) Cramer et al. (2020) datasets

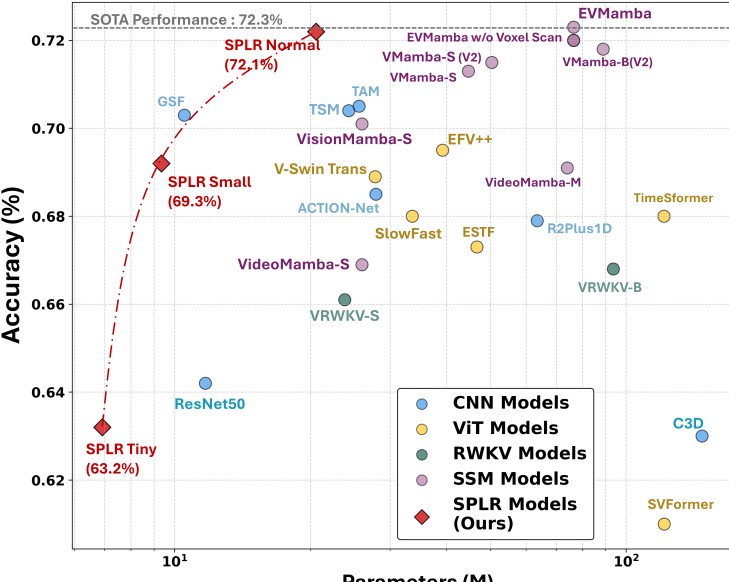

Figure 6: Figure showing the Parameters vs Accuracy of different state of the art DNN and SNN models on the Celex-HAR Wang et al. (2024a) dataset wrt the SPLR models

Table 6: Experimental results on CeleX-HAR dataset.

| No. | Algorithm | Publish | Arch. | FLOPs | Params | acc/top-1 | Code |
|---|---|---|---|---|---|---|---|
| 01 | ResNet-50 He et al. (2016) | CVPR-2016 | CNN | 8.6G | 11.7M | 0.642 | URL |
| 02 | ConvLSTM Shi et al. (2015) | NIPS-2015 | CNN, LSTM | - | - | 0.539 | URL |
| 03 | C3D Tran et al. (2015) | ICCV-2015 | CNN | 0.1G | 147.2M | 0.630 | URL |
| 04 | R2Plus1D Tran et al. (2018) | CVPR-2018 | CNN | 20.3G | 63.5M | 0.679 | URL |
| 05 | TSM Lin et al. (2019) | ICCV-2019 | CNN | 0.3G | 24.3M | 0.704 | URL |
| 06 | ACTION-Net Wang et al. (2021) | CVPR-2021 | CNN | 17.3G | 27.9M | 0.685 | URL |
| 07 | TAM Liu et al. (2021) | ICCV-2021 | CNN | 16.6G | 25.6M | 0.705 | URL |
| 08 | GSF Sudhakaran et al. (2023) | TPAMI-2023 | CNN | 16.5G | 10.5M | 0.703 | URL |
| 09 | V-SwinTrans Liu et al. (2022c) | CVPR-2022 | ViT | 8.7G | 27.8M | 0.689 | URL |
| 10 | TimeSformer Bertasius et al. (2021) | ICML-2021 | ViT | 53.6G | 121.2M | 0.680 | URL |
| 11 | SlowFast Feichtenhofer et al. (2019) | ICCV-2019 | ViT | 0.3G | 33.6M | 0.680 | URL |
| 12 | SVFormer Xing et al. (2023) | CVPR-2023 | ViT | 196.0G | 121.3M | 0.610 | URL |
| 13 | EFV++ Chen et al. (2024) | arXiv-2024 | ViT, GNN | 36.3G | 39.2M | 0.695 | URL |
| 14 | ESTF Wang et al. (2024b) | AAAI-2024 | ViT, CNN | 17.6G | 46.7M | 0.673 | URL |
| 15 | VRWKV-S Duan et al. (2024) | arXiv-2024 | RWKV | 4.6G | 23.8M | 0.661 | URL |
| 16 | VRWKV-B Duan et al. (2024) | arXiv-2024 | RWKV | 18.2G | 93.7M | 0.668 | URL |
| 17 | Vision Mamba-S Zhu et al. (2024) | ICML-2024 | SSM | 5.1G | 26.0M | 0.701 | URL |
| 18 | VMamba-S Liu et al. (2024a) | arXiv-2024 | SSM | 11.2G | 44.7M | 0.713 | URL |
| 19 | VMamba-S(V2) Liu et al. (2024a) | arXiv-2024 | SSM | 8.7G | 50.4M | 0.715 | URL |
| 20 | VMamba-B Liu et al. (2024a) | arXiv-2024 | SSM | 18.0G | 76.5M | 0.720 | URL |
| 21 | VMamba-B(V2) Liu et al. (2024a) | arXiv-2024 | SSM | 15.4G | 88.9M | 0.718 | URL |
| 22 | VideoMamba-S Li et al. (2024) | ECCV-2024 | SSM | 4.3G | 26.0M | 0.669 | URL |
| 23 | VideoMamba-M Li et al. (2024) | ECCV-2024 | SSM | 12.7G | 74.0M | 0.691 | URL |
| 24 | EVMamba | arXiv-2024 | SSM | 37.2G | 76.5M | **0.723** | URL |
| 25 | EVMamba *w/o* Voxel Scan | arXiv-2024 | SSM | 18.0G | 76.5M | 0.720 | URL |
| 26 | **SPLR-Tiny (Ours)** | - | SSM | 0.034G | 7.91M | 0.632 | - |
| 27 | **SPLR-Small (Ours)** | - | SSM | 0.13G | 13.35M | 0.692 | - |
| 28 | **SPLR-Normal (Ours)** | - | SSM | 0.41G | 25.57M | 0.722 | - |

Table 7: Comparison of classification accuracy and parameters of different models across SHD and SSC datasets.

| Model | SHD | | SSC | |
|---|---|---|---|---|
| | #Parameters | Accuracy (%) | #Parameters | Accuracy (%) |
| SFNN Cramer et al. (2020) | 0.09 M | 48.1 | 0.09 M | 32.5 |
| SRNN Cramer et al. (2020) | 1.79 M | 83.2 | - | - |
| SRNN Cramer et al. (2022) | 0.17 M | 81.6 | - | - |
| SRNN Perez-Nieves et al. (2021) | 0.11 M | 82.7 | 0.11 M | 60.1 |
| SCNN Rossbroich et al. (2022) | 0.21 M | 84.8 | - | - |
| SRNN Yin et al. (2021) | 0.14 M | 90.4 | 0.77 M | 74.2 |
| HRSNN Chakraborty & Mukhopadhyay (2023) | - | 80.01 | - | 59.28 |
| LSTM Cramer et al. (2020) | 0.43 M | 89.2 | 0.43 M | 73.1 |
| DH-SRNN Zheng et al. (2024b) | 0.05 M | 91.34 | 0.27 M | 81.03 |
| DH-SFNN Zheng et al. (2024b) | 0.05 M | 92.1 | 0.35 M | 82.46 |
| ASGL Wang et al. (2023) | - | 78.90 | - | 78.90 |
| DCLS Hammouamri et al. (2024) | 0.2 M | 95.07 | 2.5 M | 80.69 |
| TIM Shen et al. (2024b) | 2.59 M | 86.3 | 0.111 M | 61.09 |
| TC-LIF Zhang et al. (2024) | 0.142 M | 88.91 | - | - |
| SPLR-Normal (128) [Ours] | 0.513 M | 94.68 | 0.513 M | 87.52 |
| SPLR-Small (64) [Ours] | 0.129 M | 90.57 | 0.129 M | 82.08 |
| SPLR-Tiny (32) [Ours] | 0.033 M | 86.24 | 0.033 M | 72.19 |

## 8.2 ABLATION STUDIES:

To evaluate the contribution of individual components in the SPLR model, we performed extensive ablation studies on the sequential CIFAR-10 dataset. Specifically, we analyzed the impact of removing or replacing key components such as the dendritic attention layer, Spike-Aware HiPPO (SA-HiPPO), NPLR decomposition, and FFT convolution. The results of these experiments, along with the corresponding model parameters and computational costs (in GFLOPs), are summarized in Table 9.

**Impact of Dendritic Attention Layer** Removing the dendritic attention mechanism leads to a reduction in both accuracy and model parameters. The accuracy drops across all channel configurations, with the largest channels (128) seeing a decrease from 90.25% to 85.83%. The smaller channel configurations (64 and 32) experience similar drops, highlighting the dendritic attention's role in

Table 8: Comparison of classification accuracy, parameters, and FLOPs of different models across the DVS128-Gesture dataset.

| Model | #Parameters (M) | GFLOPs | Accuracy (%) |
|---|---|---|---|
| Yousefzadeh et al. Yousefzadeh et al. (2019) | 1.2 | - | 95.2 |
| Xiao et al. Xiao et al. (2022) | - | - | 96.9 |
| RTRL Subramoney (2023) | 4.8 | - | 97.8 |
| She et al. She et al. (2021b) | 1.1 | - | 98.0 |
| Liu et al. Liu et al. (2022a) | - | - | 98.8 |
| Chakraborty et al. Chakraborty & Mukhopadhyay (2022) | - | - | 96.5 |
| Martin-Turrero et al. Turrero et al. (2024) | 14 | - | 96.2 |
| Martin-Turrero et al. Turrero et al. (2024) | 14 | - | 94.1 |
| CNN + S5 (time-frames) Schöne et al. (2024) | 6.8 | - | 97.8 |
| Event-SSM Schöne et al. (2024) | 5 | - | 97.7 |
| CNN + S5 (event-frames) Schöne et al. (2024) | 6.8 | - | 97.3 |
| TBR+I3D Innocenti et al. (2021) | 12.25 | 38.82 | 99.6 |
| Event Frames + I3D Bi et al. (2020) | 12.37 | 30.11 | 96.5 |
| EV-VGCNN Deng et al. (2022) | 0.82 | 0.46 | 95.7 |
| RG-CNN Miao et al. (2019) | 19.46 | 0.79 | 96.1 |
| PointNet++ Wang et al. (2019) | 1.48 | 0.872 | 95.3 |
| PLIF Fang et al. (2021) | 1.7 | - | 97.6 |
| GET Peng et al. (2023) | 4.5 | - | 97.9 |
| Swin-T v2 Liu et al. (2022b) | 7.1 | - | 93.2 |
| TTPOINT Ren et al. (2024) | 0.334 | 0.587 | 98.8 |
| EventMamba Ren et al. (2024) | 0.29 | 0.219 | 99.2 |
| STC-LIF Zuo et al. (2024) | 3.922 | - | 83.0 |
| Spike-Driven Transformer Yao et al. (2024) | 36.01 | 33.32 | 99.3 |
| SPLR-Normal (128) **[Ours]** | 0.513 | 0.43 | 96.5 |
| SPLR-Small (64) **[Ours]** | 0.129 | 0.14 | 93.7 |
| SPLR-Tiny (32) **[Ours]** | 0.033 | 0.07 | 89.2 |
| SPLR-Normal (128 Channels) No Dendrite **[Ours - Ablation]** | 0.501 | 0.43 | 95.2 |
| SPLR-Small (64 Channels) No Dendrite **[Ours - Ablation]** | 0.121 | 0.14 | 89.3 |
| SPLR-Tiny (32 Channels) No Dendrite **[Ours - Ablation]** | 0.031 | 0.07 | 81.5 |
| SPLR-Normal (128 Channels) No HiPPO **[Ours - Ablation]** | 0.501 | 0.43 | 90.4 |
| SPLR-Small (64 Channels) No HiPPO **[Ours - Ablation]** | 0.121 | 0.14 | 82.6 |
| SPLR-Tiny (32 Channels) No HiPPO **[Ours - Ablation]** | 0.031 | 0.07 | 73.5 |

improving the spatio-temporal feature representation. Interestingly, removing this mechanism slightly reduces the model's GFLOPs since the computations associated with the dendritic layer are avoided.

**Impact of Spike-Aware HiPPO** Replacing SA-HiPPO with a simple LIF-based mechanism leads to a moderate drop in accuracy (e.g., from 90.25% to 87.62% for 128 channels). However, this modification does not alter the computational cost (GFLOPs), as SA-HiPPO primarily affects the temporal memory adaptation rather than the core matrix or convolution operations. These results emphasize SA-HiPPO's critical role in retaining and managing temporal dynamics effectively.

**Impact of NPLR Decomposition** The NPLR decomposition significantly reduces the computational complexity of state-space updates. Removing NPLR decomposition results in a notable increase in GFLOPs across all configurations (e.g., from 0.43 GFLOPs to 1.8 GFLOPs for 128 channels) due to the quadratic complexity of dense matrix operations. Despite this computational overhead, the accuracy remains relatively stable, highlighting that NPLR's primary advantage is computational efficiency rather than feature extraction performance.

**Impact of FFT Convolution** FFT convolution is integral to efficiently handling long-range temporal dependencies. Replacing FFT convolution with standard time-domain convolution increases the GFLOPs substantially (e.g., from 0.43 GFLOPs to 1.2 GFLOPs for 128 channels). Furthermore, the accuracy sees a more pronounced decline (e.g., from 90.25% to 86.47%), particularly in tasks requiring high temporal resolution. These results underscore FFT convolution's dual role in reducing computational cost and maintaining temporal modeling performance.

**Summary of Findings** The ablation studies validate the critical importance of each component in the SPLR model:

- The dendritic attention layer enhances the spatio-temporal feature representation, significantly improving accuracy.

- SA-HiPPO dynamically adjusts temporal memory retention, contributing to performance robustness without additional computational overhead.

Table 9: Updated Ablation Study for SPLR Variants on seqCIFAR-10 with FLOPs

| Model Variant | Channels | Accuracy (%) | Params (M) | FLOPs (GFLOPs) |
|---|---|---|---|---|
| SPLR (Full) | 128 | 90.25 | 0.513 | 0.43 |
| SPLR (No SA-HiPPO) | 128 | 87.62 | 0.501 | 0.43 |
| SPLR (No NPLR Decomposition) | 128 | 88.05 | 0.513 | 1.8 |
| SPLR (No FFT Convolution) | 128 | 86.47 | 0.513 | 1.2 |
| SPLR (No Dendrite) | 128 | 85.83 | 0.501 | 0.43 |
| SPLR (Full) | 64 | 88.62 | 0.129 | 0.14 |
| SPLR (No SA-HiPPO) | 64 | 86.14 | 0.121 | 0.14 |
| SPLR (No NPLR Decomposition) | 64 | 86.72 | 0.129 | 0.56 |
| SPLR (No FFT Convolution) | 64 | 85.23 | 0.129 | 0.32 |
| SPLR (No Dendrite) | 64 | 84.65 | 0.121 | 0.14 |
| SPLR (Full) | 32 | 83.15 | 0.033 | 0.034 |
| SPLR (No SA-HiPPO) | 32 | 81.75 | 0.031 | 0.034 |
| SPLR (No NPLR Decomposition) | 32 | 82.12 | 0.033 | 0.12 |
| SPLR (No FFT Convolution) | 32 | 80.62 | 0.033 | 0.08 |
| SPLR (No Dendrite) | 32 | 80.05 | 0.031 | 0.034 |

- NPLR decomposition ensures scalability by reducing the computational cost of state-space updates, making the model efficient for large-scale tasks.

- FFT convolution is indispensable for capturing long-range dependencies efficiently while keeping computational complexity low.

The full SPLR model represents a carefully optimized design that balances accuracy, efficiency, and scalability, making it suitable for real-time and resource-constrained spiking neural network applications.

### 8.3 LONG-RANGE DEPENDENCIES

**Sequential CIFAR Datasets** The first set of experiments evaluates the ability of the proposed *SPLR* model to effectively capture long-range dependencies in sequential data. This is crucial for applications involving event-driven data spanning extended periods, such as continuous gesture recognition and video analysis. To simulate long-term temporal relationships, we conduct experiments using the *Sequential CIFAR-10* and *Sequential CIFAR-100* datasets, where each image is transformed into a sequence of frames.

In these experiments, we compare the performance of *SPLR* against several baselines, including traditional SNN models. The key focus is on assessing the effectiveness of our *Spike-Aware HiPPO (SA-HiPPO)* dynamics in retaining temporal memory over long sequences. The results are presented in Table 10, which includes classification accuracy for different sequence lengths, as well as model complexity in terms of the number of parameters.

As seen in Table 10, *SPLR* significantly outperforms the baselines in capturing long-range dependencies. The *SPLR* model with 128 channels achieves an accuracy of 90.25% on the *Sequential CIFAR-10* dataset and 65.33% on *Sequential CIFAR-100*, which surpasses the performance of all baseline models by a substantial margin. These results indicate that *SPLR* not only maintains memory over extended input sequences but also converges faster, achieving higher accuracy with fewer epochs compared to traditional spiking and hybrid models.

The ablation study further reveals that the SA-HiPPO matrix incorporated in *SPLR* plays a pivotal role in enhancing temporal filtering capabilities, leading to improved convergence rates and more robust performance in long-range dependency tasks. This improvement is evident in the accuracy gains observed in *SPLR* compared to other models, including those using mechanisms like GLIF and PLIF.

Moreover, even when model complexity is reduced, as seen in the *SPLR* variants with 64 and 32 channels, our model maintains superior accuracy compared to all baseline architectures. For instance, the *SPLR* with 64 channels achieves 88.% accuracy on *Sequential CIFAR-10*, outperforming other models with similar parameter counts, demonstrating the efficiency and scalability of the proposed SA-HiPPO dynamics for capturing long-term dependencies in sequential data.

Table 10: Comparison of Architectures on Sequential CIFAR-10 and CIFAR-100

| Architecture | Channels | Layer Type | seqCIFAR10 Accuracy (%) | seqCIFAR100 Accuracy (%) |
|---|---|---|---|---|
| 6Conv+FC | 128 | *PSN* Fang et al. (2023) | 88.45 | 62.21 |
| | | *masked PSN* Fang et al. (2023) | 85.81 | 60.69 |
| | | *GLIF* Yao et al. (2022) | 83.66 | 58.92 |
| | | *KLIF* Jiang & Zhang (2023) | 83.26 | 57.37 |
| | | *PLIF* Fang et al. (2021) | 83.49 | 57.55 |
| | | *LIF* | 81.50 | 55.45 |
| | | *SPLR* | 90.25 | 65.33 |
| | 64 | *SPLR* | **88.62** | **63.57** |
| | 32 | *SPLR* | **83.15** | **56.32** |

These findings validate the superior temporal modeling capabilities of *SPLR*, making it well-suited for tasks that require efficient and scalable handling of long-range dependencies in sequential, event-driven data.

**Long Range Arena Datasets:** We evaluate the ability of the proposed *SPLR* model to capture long-range dependencies using the **Long Range Arena (LRA)** dataset Tay et al. (2020). The LRA benchmark evaluates models on tasks requiring long-context understanding, where Transformer-based non-spiking models often exhibit suboptimal performance due to the computational overhead of attention mechanisms, which scales poorly with increasing sequence lengths. As shown in Table 3, we benchmark our method against state-of-the-art alternatives, including the LMU-based spiking model, SpikingLMUFormer Liu et al. (2024b), and the BinaryS4D model Stan & Rhodes (2024). While BinaryS4D is not fully spiking—it relies on floating-point MAC operations for matrix multiplications—it incorporates LIF neurons to spike from an underlying state-space model (SSM), providing a hybrid approach to handling long-range dependencies.

## 8.4 DVS Gesture Recognition

To further investigate the combined effectiveness of dendritic mechanisms and *SPLR* convolutions in event-based processing, we evaluate our model on the *DVS Gesture* dataset. This dataset consists of event streams recorded from a Dynamic Vision Sensor (DVS) at a resolution of $128 \times 128$, providing a challenging benchmark for evaluating temporal dynamics in gesture recognition tasks involving varying speeds and motions.

Our goal is to assess how the integration of dendritic mechanisms with *SPLR* convolution layers enhances the model's ability to capture multi-scale temporal dependencies. Specifically, we examine how dendrites can serve as a temporal attention mechanism that helps *SPLR* effectively focus on the most relevant events, while *SPLR* convolutions manage the overall temporal and spatial evolution of features.

The experiment involves training variants of our model—one incorporating both dendritic mechanisms and *SPLR* convolutions, and the other using only *SPLR*—to determine the contribution of dendritic attention. Table 8 summarizes the test accuracy of our models compared to other state-of-the-art approaches. The results are measured in terms of classification accuracy, along with the number of parameters, to highlight model efficiency.

As shown in Table 8, the *SPLR* model with 128 channels, incorporating dendritic attention, achieves 96.5% accuracy while maintaining a significantly lower parameter count compared to many other state-of-the-art models. This shows that our approach effectively utilizes sparse event-driven inputs to achieve high accuracy with reduced computational complexity. The use of dendritic mechanisms allows the model to dynamically adjust its focus on different temporal scales, thus improving gesture recognition even in scenarios with rapid motion changes.

The variant without dendritic attention, while still competitive, lags behind in adapting to the multi-scale nature of the event data, especially for gestures with complex temporal characteristics. This indicates that the dendritic mechanism plays a crucial role in adaptively filtering relevant temporal features, which is essential for handling the asynchronous, irregular inputs typical of event cameras.

In addition, visualizations of the learned dendritic activity reveal how the model attends to different time segments, effectively filtering the incoming spike streams to prioritize the most relevant events.

This adaptive filtering complements the *SPLR* convolutional operations, leading to more robust and efficient temporal feature extraction.

Overall, the results validate the utility of combining dendritic mechanisms with *SPLR* convolutions for event-driven tasks, making the model well-suited for gesture recognition from DVS inputs. The joint use of these components allows for efficient temporal modeling, maintaining a favorable trade-off between accuracy and parameter efficiency.

## 8.5 SCALING TO HD EVENT STREAMS

The scalability of the proposed *SPLR* model is evaluated on the *Celex HAR* dataset, a human activity recognition dataset recorded at a high resolution of $1280 \times 800$. This dataset serves as a challenging benchmark for assessing the model's ability to maintain high accuracy and computational efficiency when processing large-scale spatial and temporal data.

In this experiment, *SPLR* is used for action recognition on HD event streams, and its performance is compared to that of baseline Spiking Neural Networks (SNNs) and State-Space Models (SSMs). As shown in Figure 3, the results demonstrate that *SPLR* maintains high accuracy even at increased resolutions, whereas the baseline models experience significant performance degradation due to heightened computational demands. The integration of the *SPLR convolution layer* proves effective in managing the complex spatial and temporal components of HD event data, providing robust real-time processing capabilities with minimal computational overhead.

Figure 3 illustrates the trade-off between accuracy and computational cost, measured in terms of FLOPs, for our *SPLR* models compared to state-of-the-art methods on the *Celex-HAR* dataset. The *SPLR* variants—*SPLR Tiny*, *SPLR Small*, and *SPLR Normal*—demonstrate superior efficiency by achieving competitive or better accuracy while utilizing significantly fewer computational resources.

Key observations from Figure 3 are as follows:

- **Efficiency at Different Scales**: *SPLR Tiny* achieves approximately 63.8% accuracy with a fraction of the computational cost compared to larger models such as *SlowFast* and *C3D*. As the model scales to *SPLR Small* and *SPLR Normal*, accuracy improves to 69.3% and 72.1%, respectively, while maintaining a favorable computational cost profile.

- **Performance with Reduced Complexity**: *SPLR Normal* matches or exceeds the accuracy of models like *TSM* and *VisionMamba-S* but at a substantially lower computational cost. This efficiency is attributed to the integration of event-driven processing and effective state-space dynamics.

The improved efficiency of *SPLR* can be credited to the event-based processing capabilities of the **SPLR architecture** and the **SPLR convolution layer**, which optimally manage state-space

Table 11: Latency Comparison on Celex-HAR (in microseconds)

| Algorithm | Latency (us) |
|---|---|
| **SPLR-Tiny** | **0.162** |
| **SPLR-Small** | **0.582** |
| **SPLR-Normal** | **1.867** |
| ResNet-50 | 41.575 |
| C3D | 0.473 |
| R2Plus1D | 94.264 |
| TSM | 1.4266 |
| ACTION-Net | 81.035 |
| TAM | 76.012 |
| GSF | 75.558 |
| V-SwinTrans | 39.837 |
| TimeSformer | 255.425 |
| SlowFast | 1.118 |
| EFV++ | 166.23 |
| ESTF | 80.61 |
| SVFormer | 897.455 |
| VRWKV-S | 21.091 |
| VRWKV-B | 86.346 |
| Vision Mamba-S | 23.88 |
| VMamba-S | 53.302 |
| VMamba-S(V2) | 39.848 |
| VMamba-B | 82.421 |
| VMamba-B(V2) | 70.514 |
| VideoMamba-S | 19.707 |
| VideoMamba-M | 58.164 |
| EVMamba | 170.34 |
| EVMamba w/o Voxel Scan | 82.423 |

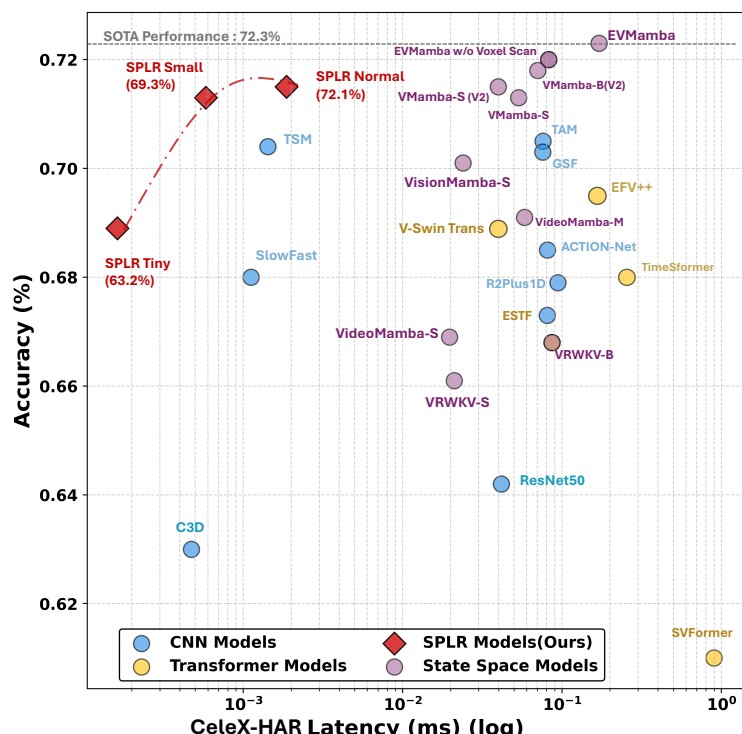

Figure 7: Figure showing Accuracy vs Inference Latency for different models on the Celex-HAR dataset.

evolution without relying on dense operations. These features allow the model to capture complex temporal dependencies while minimizing computational requirements, making *SPLR* particularly effective for high-resolution event-based datasets like *Celex-HAR*.

**HAR-DVS Results:** The HAR-DVS dataset results underscore the advantages of our SPLR models, achieving accuracies of **70.38%**, **81.73%**, and **88.29%** for SPLR-Tiny, SPLR-Small, and SPLR-Normal, respectively, while maintaining substantially lower computational costs compared to other state-of-the-art models. Unlike traditional deep neural networks such as C3D and R2Plus1D, which struggle to model the complex temporal relationships inherent in event streams, SPLR leverages a novel *event-by-event processing approach*, preserving fine-grained temporal dynamics essential for accurate action recognition.

Moreover, SPLR employs a unique *dendritic attention mechanism* that enhances its ability to capture long-range spatio-temporal dependencies efficiently. The prolonged and complex actions in HAR-DVS demand robust temporal attention mechanisms, as highlighted in prior studies. SPLR's dendritic-inspired design meets these requirements while offering a computationally efficient solution, making it particularly suitable for real-time, low-latency applications in dynamic event-driven environments.

It is important to note that HAR-DVS provides frame-based data, as raw event data was unavailable for download. Since SPLR is designed for event-by-event processing, we treated all events arriving at the same timestamp as a single batch for processing, adhering to the event-driven principles of the model.

## 8.6 LATENCY RESULTS

Table 11 presents the latency results (in microseconds) for SPLR and various state-of-the-art methods on the Celex-HAR dataset. SPLR outperforms all competing models in terms of latency, with SPLR-Tiny achieving the lowest latency of 0.162 $\mu$s. SPLR-Small and SPLR-Normal maintain low latencies of 0.582 $\mu$s and 1.867 $\mu$s, respectively, while providing competitive accuracy. In contrast, high-performing models such as TimeSformer (255.425 $\mu$s), EFV++ (166.23 $\mu$s), and R2Plus1D (94.264 $\mu$s) exhibit significantly higher latencies. Even latency-optimized models like VideoMamba-S

(19.707 $\mu$s) and SlowFast (1.118 $\mu$s) are surpassed by SPLR configurations, demonstrating SPLR's exceptional efficiency in the latency-accuracy trade-off. These results highlight SPLR's suitability for real-time, resource-constrained applications.

# 9 SUPPLEMENTARY SECTION C: METHODS AND ARCHITECTURAL DETAILS

## BACKGROUND AND PRELIMINARIES

**State-Space Models:** A state-space model (SSM) is a mathematical framework for modeling systems that evolve over time. The dynamics of such systems are described by a set of first-order differential equations, often expressed in continuous time as:

$$\dot{x}(t) = Ax(t) + Bu(t), \quad y(t) = Cx(t) + Du(t)$$

where:

- $x(t) \in \mathbb{R}^N$ is the hidden state vector, representing the internal state of the system at time $t$,
- $u(t) \in \mathbb{R}^M$ is the input signal, such as sensory data or external stimuli,
- $y(t) \in \mathbb{R}^P$ is the output signal or observable state,
- $A \in \mathbb{R}^{N \times N}$, $B \in \mathbb{R}^{N \times M}$, $C \in \mathbb{R}^{P \times N}$, and $D \in \mathbb{R}^{P \times M}$ are learned system matrices.

State-space models are often used in signal processing and control systems to model systems with temporal dependencies. In many practical scenarios, however, the continuous-time formulation is discretized:

$$x_{k+1} = A_d x_k + B_d u_k, \quad y_k = C_d x_k + D_d u_k$$

where $A_d$, $B_d$, $C_d$, and $D_d$ are the corresponding discrete-time matrices, and $k$ indexes the discrete time steps.

**Spiking Neural Networks (SNNs):** SNNs are a class of neural networks that more closely mimic biological neurons. In SNNs, information is transmitted as spikes, or binary events, at discrete times, as opposed to continuous activations in traditional neural networks. A typical neuron in an SNN, such as the *Leaky Integrate-and-Fire (LIF)* neuron, is governed by the following dynamics:

$$\tau_m \frac{dV_i(t)}{dt} = -V_i(t) + I_i(t)$$

where $V_i(t)$ is the membrane potential of neuron $i$, $\tau_m$ is the membrane time constant, and $I_i(t)$ is the input current, typically derived from presynaptic neurons or external stimuli.

A spike is emitted when the membrane potential exceeds a threshold $\theta_i$. After a spike, the membrane potential is reset, and a refractory period prevents immediate re-firing.

Despite their potential for efficient temporal data processing, SNNs are difficult to train due to the non-differentiability of spikes and the complex membrane potential dynamics.

**Highly Optimized Polynomial Projection (HiPPO):** The *HiPPO* framework provides a method for approximating the continuous history of an input signal by projecting it onto a set of polynomial basis functions. The HiPPO matrix $A$ is designed to optimally compress the history of the input into a state vector $x(t)$, allowing the model to retain relevant temporal dependencies over long time scales. For example, the HiPPO-Legendre (HiPPO-LegS) matrix $A$ is defined as:

$$A_{nk} = \begin{cases} -\sqrt{(2n+1)(2k+1)} & \text{if } n > k \\ n + 1 & \text{if } n = k \\ 0 & \text{if } n < k \end{cases}$$

This matrix governs the dynamics of how the internal state evolves to represent the history of the input in a compressed manner.

## MATHEMATICAL MODELING AND SPIKE GENERATION MECHANISM

Spikes in the SPLR model are generated through the dynamics of LIF neurons. The spike generation process is described in detail below:

- **Dendritic Current Integration**: Each DH-LIF neuron integrates incoming spikes through its dendritic branches:

$$i_d(t+1) = \alpha_d i_d(t) + \sum_{j \in N_d} w_j p_j, \tag{1}$$

where $\alpha_d = e^{-\frac{1}{\tau_d}}$ represents the decay rate, $w_j$ is the synaptic weight, and $p_j$ is the input spike value.

- **Soma Potential Update and Spike Generation**: The soma potential is updated based on the integrated dendritic currents:

$$V(t+1) = \beta V(t) + \sum_d g_d i_d(t), \tag{2}$$

where $\beta = e^{-\frac{1}{\tau_s}}$ is the decay rate of the soma, and $g_d$ is the coupling strength of each dendrite. A spike is generated if $V(t)$ exceeds the threshold $V_{\text{th}}$.

- **Spike Propagation**: The generated spikes propagate through the network according to:

$$x(t_{k+1}) = e^{A\Delta t_k} x(t_k) + A^{-1}(e^{A\Delta t_k} - I)BS(t_k), \tag{3}$$

preserving both spatial and temporal information.

METHODS

The proposed model is designed to handle sparse, asynchronous event-based inputs effectively while being scalable to high-definition (HD) event streams. It leverages *Dendrite Heterogeneity Leaky Integrate-and-Fire (DH-LIF)* neurons in the first layer to capture *multi-scale temporal dynamics*, crucial for preserving temporal details inherent in event streams while reducing spatial and computational redundancy. The model then utilizes a series of *spiking state-space convolution* layers, enabling efficient integration of both local and global temporal relationships. The final *readout layer* employs event pooling and a linear transformation to produce a compact and meaningful representation for downstream tasks such as classification or regression. This architecture ensures robustness and scalability, making it suitable for high-resolution inputs.

VARIABLES AND NOTATIONS

To ensure clarity, we provide definitions for all variables and notation used in the equations:

**Input Representation:**

- $x, y$: Spatial coordinates of the spike event.
- $t$: Timestamp of the spike.
- $p$: Magnitude or polarity of the spike.

**Dendrite Attention Layer:**

- $\tau_d$: Dendritic timing factor, representing the temporal scale of each dendrite.
- $i_d(t)$: Dendritic current for branch $d$ at time $t$.
- $\alpha_d$: Decay rate for dendritic branch $d$, defined as $\alpha_d = e^{-\frac{1}{\tau_d}}$.
- $\mathcal{N}_d$: Set of presynaptic inputs connected to dendrite $d$.
- $w_j$: Synaptic weight associated with presynaptic input $p_j$.
- $V(t)$: Membrane potential of the soma at time $t$, aggregated from all dendritic currents.
- $\beta$: Decay rate of the soma, defined as $\beta = e^{-\frac{1}{\tau_s}}$, where $\tau_s$ is the soma's time constant.
- $g_d$: Coupling strength of dendrite $d$ to the soma.
- $V_{\text{th}}$: Threshold potential for spike generation.

**Spatial Pooling Layer:**

- $I(x, y, t)$: Initial spike activity at location $(x, y)$ and time $t$.
- $I_{\text{pooled}}(x', y', t)$: Spatially pooled spike activity at location $(x', y')$ and time $t$.
- $P(x', y')$: Pooling window centered at $(x', y')$.

**SPLR Convolution Layer:**

- $x(t)$: Internal state vector at time $t$.
- $S(t)$: Input spike train, where $S_i(t) = \sum_k \delta(t - t_i^k)$ and $\delta(t)$ is the Dirac delta function.
- $A_S$: Spike-Aware HiPPO (SA-HiPPO) matrix, dynamically adapted based on inter-spike intervals.
- $B, C$: Input and output coupling matrices.
- $\Delta t$: Inter-spike interval, defined as the time difference between consecutive spikes.
- $F(\Delta t)$: Decay matrix for SA-HiPPO, where $F_{ij}(\Delta t) = e^{-\alpha_{ij}\Delta t}$.
- $V, \Lambda, P, Q$: Components of NPLR decomposition:
    - $V$: Unitary matrix.
    - $\Lambda$: Diagonal matrix of decay rates.
    - $P, Q$: Low-rank matrices, where $r \ll N$.
- $K(\omega)$: FFT convolution kernel, defined as $K(\omega) = \frac{1}{\omega - \Lambda}$.
- $\text{FFT}(\cdot), \text{IFFT}(\cdot)$: Fast Fourier Transform and its inverse.

**Normalization Layer:**

- $x_l$: Input to the normalization layer at layer $l$.
- $\mu_l, \sigma_l^2$: Mean and variance of the activations at layer $l$.
- $\gamma, \beta$: Learnable scale and shift parameters for layer normalization.

**Readout Layer:**

- $x_{\text{pooled},k}$: Pooled state vector, computed as $x_{\text{pooled},k} = \frac{1}{p} \sum_{i=kp}^{(k+1)p-1} x_i$, where $p$ is the pooling factor.
- $W, b$: Learnable weight matrix and bias for the linear transformation.
- $y$: Final output of the model, computed as $y = W x_{\text{pooled}} + b$.

OVERVIEW OF THE SPLR MODEL

The proposed Spiking Network for Learning Long-Range Relations (SPLR) addresses the limitations of conventional spiking neural networks (SNNs) in capturing long-range temporal dependencies while maintaining event-driven efficiency. The SPLR model is composed of the following key components:

---

**Algorithm 2** SPLR Model Processing

---

**Require:** Input spike event sequence $X = \{(x_i, y_i, t_i, p_i)\}$
1: **Initialize** model parameters
2: **Process** input through **Dendrite Attention Layer** (Algorithm 3)
3: **Apply Spatial Pooling Layer** to reduce spatial dimensions (Algorithm 4)
4: **Pass** output to **SPLR Convolution Layer** to capture temporal dynamics (Algorithm 5)
5: **Update** state using **Spike-Aware HiPPO** mechanism (Algorithm 5)
6: **Aggregate** information in the **Readout Layer** for final output (Algorithm 6)
7: **Output**: Model prediction $y$

---

## 9.1 INPUT REPRESENTATION

The input to the model is represented as a sequence of spike events, each defined by the tuple $(x, y, t, p)$, where $(x, y)$ are the spatial coordinates, $t$ is the timestamp, and $p$ represents the magnitude or polarity of the spike. These events are streamed asynchronously, reflecting the sparse nature of the data. The model is also designed to handle higher resolutions, allowing scalability to HD event streams. This input representation emphasizes the need for efficient aggregation of both spatial and temporal information while minimizing computational load.

## 9.2 DENDRITE ATTENTION LAYER

The model begins by passing the input through the *Dendrite Attention Layer*, constructed using DH-LIF neurons as shown in Fig. 1. Each DH-LIF neuron features multiple dendritic branches, each with a unique timing factor $\tau_d$, enabling the capture of temporal dynamics across a range of timescales, which is essential for accommodating the diverse timescales present in asynchronous spike inputs. The dynamics of the dendritic current $i_d(t)$ are governed by $i_d(t+1) = \alpha_d i_d(t) + \sum_{j \in \mathcal{N}_d} w_j p_j$, where

$\alpha_d = e^{-\frac{1}{\tau_d}}$ is the decay rate for branch $d$, and $w_j$ represents the synaptic weight associated with presynaptic input $p_j$. The set $\mathcal{N}_d$ represents the presynaptic inputs connected to dendrite $d$, ensuring that each dendrite captures temporal features independently, functioning as independent temporal filters. Unlike a standard CUBA LIF neuron model, which integrates all inputs uniformly at the soma with a single timescale, the dendritic attention layer introduces multiple dendritic branches, each independently filtering inputs at different temporal scales. This design enables the neuron to selectively process asynchronous inputs and retain information across diverse temporal windows, providing greater flexibility and adaptability.

The dendritic currents from each branch are aggregated at the soma, resulting in the membrane potential $V(t+1) = \beta V(t) + \sum_d g_d i_d(t)$, where $\beta = e^{-\frac{1}{\tau_s}}$ represents the soma's decay rate, and $g_d$ represents the coupling strength of dendrite $d$ to the soma. A spike is generated whenever the membrane potential exceeds a threshold $V_{\text{th}}$, allowing the neuron to selectively fire only when sufficiently excited.

---
**Algorithm 3** Dendrite Attention Layer

---
**Require:** Input spike events $X = \{(x_i, y_i, t_i, p_i)\}$, dendritic timing factors $\{\tau_d\}$, synaptic weights $\{w_j\}$, coupling strengths $\{g_d\}$, threshold $V_{\text{th}}$
 1: **Initialize** dendritic currents $i_d(0)$ and membrane potential $V(0)$
 2: **for** each time step $t$ **do**
 3:     **for** each dendrite $d$ **do**
 4:         Compute decay rate: $\alpha_d \leftarrow e^{-\frac{1}{\tau_d}}$
 5:         Update dendritic current: $i_d(t+1) \leftarrow \alpha_d i_d(t) + \sum_{j \in \mathcal{N}_d} w_j p_j$
 6:     **end for**
 7:     Compute soma decay rate: $\beta \leftarrow e^{-\frac{1}{\tau_s}}$
 8:     Update membrane potential: $V(t+1) \leftarrow \beta V(t) + \sum_d g_d i_d(t)$
 9:     **if** $V(t+1) > V_{\text{th}}$ **then**
10:         Generate spike at time $t+1$
11:         Reset membrane potential: $V(t+1) \leftarrow 0$
12:     **end if**
13: **end for**
14: **Output:** Spatio-temporal features $I(x, y, t)$

---

## 9.3 SPATIAL POOLING LAYER

Following the dendritic attention layer, a *Spatial Pooling Layer* is introduced to reduce the spatial dimensionality of the resulting output. Given the initial spike activity $I(x, y, t)$ at location $(x, y)$, the pooling operation reduces spatial dimensions while preserving temporal resolution:

$$I_{\text{pooled}}(x', y', t) = \max_{(x,y) \in P(x', y')} I(x, y, t)$$

where $P(x', y')$ is a pooling window centered at $(x', y')$. Pooling reduces spatial complexity, simplifying subsequent processing in the network while retaining key features. This is especially useful for HD event streams with extensive spatial information.

## 9.4 SPLR CONVOLUTION

The **Spiking Process with Long-term Recurrent dynamics (SPLR) Convolution Layer** is a critical component of the SPLR model, specifically designed for processing event-based spiking inputs. It

---

**Algorithm 4** Spatial Pooling Layer

---

**Require:** Input spike activity $I(x, y, t)$ from Dendrite Attention Layer, pooling window $P(x', y')$
 1: **for** each spatial location $(x', y')$ **do**
 2:      **for** each time step $t$ **do**
 3:          Pool activity: $I_{\text{pooled}}(x', y', t) \leftarrow \max\limits_{(x,y) \in P(x',y')} I(x, y, t)$
 4:      **end for**
 5: **end for**
 6: **Output**: Pooled spike activity $I_{\text{pooled}}(x', y', t)$

---

captures long-range dependencies and asynchronous dynamics by integrating mechanisms such as the **Spike-Aware HiPPO (SA-HiPPO)** framework, **Normal Plus Low-Rank (NPLR) Decomposition**, and **Fast Fourier Transform (FFT) Convolution**. These innovations collectively enable efficient and robust temporal feature extraction.

### Overview and Intuition

Traditional convolutional layers are adept at extracting spatial features but often fail to capture complex temporal dependencies, especially in asynchronous, sparse spiking data. The SPLR Convolution Layer overcomes this limitation by incorporating state-space models that inherently manage temporal dynamics. Leveraging the SA-HiPPO mechanism, the layer dynamically adapts memory retention based on spike timings, emphasizing recent events while allowing older information to decay. The use of NPLR Decomposition and FFT-based convolution further enhances computational efficiency, enabling scalability to high-dimensional, long-range temporal data.

**Spiking State-Space Model:** The temporal dynamics of the SPLR Convolution Layer are governed by the **Spiking State-Space Model**:

$$\dot{x}(t) = A_S x(t) + B S(t), \quad y(t) = C x(t), \tag{4}$$

where:

- $x(t) \in \mathbb{R}^N$ represents the internal state vector,
- $S(t) \in \mathbb{R}^M$ is the input spike train, with each component $S_i(t) = \sum_k \delta(t - t_i^k)$, where $\delta(t)$ is the Dirac delta function,
- $A_S \in \mathbb{R}^{N \times N}$ is the **Spike-Aware HiPPO** matrix,
- $B \in \mathbb{R}^{N \times M}$ and $C \in \mathbb{R}^{P \times N}$ are the input and output coupling matrices.

This framework ensures that temporal dependencies inherent in spiking data are captured effectively.

**Spike-Aware HiPPO Mechanism**: The *Spike-Aware HiPPO (SA-HiPPO)* (Fig. 8) mechanism is a core component of the SPLR model, designed to efficiently capture long-term temporal dependencies in the presence of sparse, event-based spiking inputs. The HiPPO (Highly Optimized Polynomial Projection) framework, originally developed to approximate continuous input signals, projects them onto polynomial bases, enabling efficient temporal compression of input history. However, when dealing with spike-driven dynamics, where inputs are discrete and irregular, the conventional HiPPO formulation must be adapted to properly address these challenges. The SA-HiPPO adapts the HiPPO framework to efficiently handle discrete, spike-driven inputs by introducing a decay matrix $F(\Delta t)$. This matrix adjusts memory retention based on the time elapsed between spikes $(\Delta t)$, ensuring more recent spikes have a greater influence while older information gradually decays. The Hadamard product with the original HiPPO matrix enables adaptive modulation of memory, making it more stable and suitable for asynchronous events. In a spike-driven scenario, the input signal is represented as a vector of spike trains $S(t) \in \mathbb{R}^M$, with each element $S_i(t)$ defined by $S_i(t) = \sum_k \delta(t - t_i^k)$,

where $\delta(t)$ is the Dirac delta function, and $t_i^k$ denotes the time of the $k$-th spike for input $i$. Given the irregular and sparse nature of these spike-driven inputs, we introduce a *Spike-Aware HiPPO (SA-HiPPO)* matrix $A_S$ that extends the dynamics of the standard HiPPO to efficiently process spikes. The SA-HiPPO matrix $A_S$ modifies the original HiPPO dynamics to adapt to the nature of spiking events by incorporating a decay function that accounts for the time elapsed between successive spikes. Specifically, the state evolution in the presence of spikes is modeled by $\dot{x}(t) = A_S x(t) + B S(t)$. The matrix $A_S$ is defined as $A_S = A \circ F(\Delta t)$, where $A \in \mathbb{R}^{N \times N}$ is the original HiPPO matrix, and

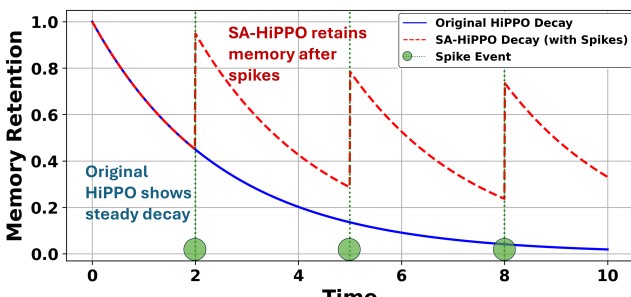

Figure 8: The SA-HiPPO decay is needed to adapt the memory retention dynamically to the irregular timing of spike events, allowing the system to prioritize recent spikes while efficiently managing the decay of older information, which enhances stability and responsiveness for event-driven inputs.

$F(\Delta t) \in \mathbb{R}^{N \times N}$ is a decay matrix that weights the original HiPPO dynamics based on the inter-spike interval $\Delta t$. The operator $\circ$ denotes the element-wise (Hadamard) product. The decay matrix $F(\Delta t)$ is formulated as $F_{ij}(\Delta t) = e^{-\alpha_{ij}\Delta t}$, where $\Delta t = t_j - t_i$ represents the time difference between spike $i$ and spike $j$, and $\alpha_{ij}$ is a decay parameter that controls how the influence of past spikes diminishes over time. The exponential decay function ensures that the impact of previous spikes decreases exponentially, allowing more recent spikes to have a stronger influence on the current state. This weighting mechanism makes the HiPPO dynamics more adaptable to spiking inputs, capturing both the recency and relevance of spikes for efficient temporal representation.

The state vector $x(t)$ thus evolves in two distinct modes: continuous evolution between spikes and instantaneous updates at spike times. Between spikes, the state evolves according to the homogeneous equation $\dot{x}(t) = A_S x(t)$. When a spike occurs at time $t_k$, the state is updated as:

$$x(t_{k+1}) = e^{A_S \Delta t_k} x(t_k) + A_S^{-1} \left( e^{A_S \Delta t_k} - I \right) B S(t_k)$$

where $\Delta t_k = t_{k+1} - t_k$ represents the time difference between successive spikes. To make the state update computationally feasible, the matrix exponential $e^{A_S \Delta t_k}$ is approximated using a truncated Taylor series expansion:

$$e^{A_S \Delta t_k} \approx I + A_S \Delta t_k + \frac{A_S^2 \Delta t_k^2}{2}$$

This first-order or second-order approximation provides a good balance between computational efficiency and accuracy, especially in scenarios with small inter-spike intervals.

The SA-HiPPO mechanism effectively extends the temporal memory capabilities of the original HiPPO framework by introducing a spike-sensitive adaptation. It ensures that the state vector $x(t)$ retains relevant temporal information while accommodating the asynchronous nature of spike inputs. The decay function embedded within $F(\Delta t)$ provides a means to dynamically adjust the influence of past inputs, thereby making the model more responsive to recent events.

**Normal Plus Low-Rank (NPLR) Decomposition**: The **NPLR Decomposition** reduces computational complexity by expressing $A_S$ as:

$$A_S = V \Lambda V^* - P Q^*, \tag{5}$$

where:

- $V \in \mathbb{C}^{N \times N}$ is a unitary matrix,
- $\Lambda \in \mathbb{C}^{N \times N}$ is a diagonal matrix of decay rates,
- $P, Q \in \mathbb{C}^{N \times r}$ are low-rank matrices, with $r \ll N$.

This decomposition reduces the complexity of matrix-vector multiplications from $O(N^2)$ to $O(Nr)$, facilitating scalability to large state spaces.

**Fast Fourier Transform (FFT) Convolution**: Long-range temporal dependencies are handled efficiently using FFT-based convolution. The convolution operation is performed as follows:

1. Transform the state vector $x(t)$ and convolution kernel $K(\omega)$ into the frequency domain using FFT.
2. Perform element-wise multiplication in the frequency domain.
3. Apply the inverse FFT (IFFT) to obtain the updated state vector in the time domain.

This approach significantly accelerates the processing of long temporal sequences by leveraging frequency-domain efficiencies. The SPLR Convolution Layer integrates these components to achieve robust spatio-temporal feature extraction:

- **Temporal Dynamics Modeling**: SA-HiPPO captures spike timing dependencies while balancing memory retention and decay.
- **Computational Efficiency**: NPLR Decomposition and FFT convolution ensure scalability and rapid processing.
- **Efficient State Management**: The state-space formulation ensures accurate updates for spiking inputs.

### 9.4.1 SPLR CONVOLUTION LAYER

Using all these concepts of SA-Hippo, NPLR Decomposition and FFT Convolution, we introduce *SPLR Convolution (SPLRConv)* layers, which generalize the spike-aware state-space operations into a convolutional framework. These layers are designed to extend the capabilities of SPLR by transforming the temporal memory operations into a convolutional form, thus allowing for more efficient feature extraction in both temporal and spatial domains. The *SPLR Conv* layer incorporates spike-based input while retaining the convolutional structure, enabling the model to operate efficiently over high-dimensional data while capturing complex temporal dependencies. The continuous-time state-space dynamics are given by:

$$\frac{d}{dt}x(t) = Ax(t) + Bu(t)$$

where $x(t) \in \mathbb{R}^N$ represents the state vector, $u(t) \in \mathbb{R}^M$ is the input, $A \in \mathbb{R}^{N \times N}$ is the state transition matrix, and $B \in \mathbb{R}^{N \times M}$ is the input coupling matrix. The state evolves based on both the internal dynamics and the influence of incoming spikes. The Spike-Aware dynamics incorporate both decay and event-driven updates

$$\dot{x}(t) = A_{\text{spike}}(t)x(t) + B_{\text{spike}}(t)u(t), \tag{6}$$

where $A_{\text{spike}}(t) = A_{\text{decay}} + A_{\text{timing}}(t)$. The matrix $A_{\text{decay}} = -\frac{1}{\tau_m}I$ models natural decay, while $A_{\text{timing}}(t)$ represents spike-driven effects and depends on the inter-spike intervals. The model discretizes these dynamics for efficient implementation, using a fixed time step $\Delta t$:

$$x_{k+1} = x_k + \Delta t(A_{\text{spike},k}x_k + B_{\text{spike},k}u_k) \tag{7}$$

At each spike time $t_i$, the state undergoes an instantaneous update $x(t_i^+) = x(t_i^-) + B_{\text{spike}}(t_i)$. To improve computational efficiency, the spiking state matrix $A_{\text{spike}}$ is decomposed using the *Normal Plus Low-Rank (NPLR) decomposition*: $A_{\text{spike}} = V\Lambda V^* - PQ^*$

where $V \in \mathbb{C}^{N \times N}$ is a unitary matrix, $\Lambda \in \mathbb{C}^{N \times N}$ represents the decay, and $P, Q \in \mathbb{C}^{N \times r}$ are low-rank matrices. This reduces the cost of matrix-vector products from $O(N^2)$ to $O(Nr)$, where $r$ is the rank of the low-rank perturbation. The resulting state update rule becomes:

$$x_{k+1} = x_k + \Delta t\left((V\Lambda V^* - PQ^*)x_k + B_{\text{spike}}u_k\right)$$

The convolution operation in these layers is realized by transforming recurrent state-space updates into a convolutional form, with the system's impulse response precomputed. Using the *Fast Fourier Transform (FFT)*, the convolution kernel $K(\omega)$ can be efficiently calculated as $K(\omega) = \frac{1}{\omega - \Lambda}$. This transformation allows the model to handle long-range temporal dependencies efficiently, even in high-resolution event-based streams.

**Computational Efficiency:** The layer achieves notable computational advantages:

- **Reduced Complexity**: NPLR Decomposition transforms operations from $O(N^2)$ to $O(Nr)$.
- **Accelerated Convolutions**: FFT convolution rapidly processes long temporal sequences.
- **Parallelization**: FFT operations are well-suited for parallel hardware architectures, enhancing performance.

**Spike Generation in SPLR Convolution Layers**: Spikes in the SPLR model are generated through the interaction of dendritic and soma compartments in the DH-LIF neurons. These neurons are integral to the Dendrite Attention Layer, which precedes each SPLR convolution layer, ensuring asynchronous and event-driven signal processing.

The dendritic branches act as independent temporal filters, accumulating and processing inputs over time:

$$i_d(t+1) = \alpha_d i_d(t) + \sum_{j \in \mathcal{N}_d} w_j p_j,$$

where $\alpha_d = e^{-\frac{1}{\tau_d}}$ is the decay rate determined by the dendritic branch's time constant $\tau_d$, $w_j$ is the synaptic weight, and $p_j$ is the presynaptic spike.

The soma aggregates these currents, with its membrane potential evolving as:

$$V(t+1) = \beta V(t) + \sum_d g_d i_d(t),$$

where $\beta = e^{-\frac{1}{\tau_s}}$ represents the soma's decay factor, and $g_d$ is the coupling strength of each dendrite $d$.

A spike is produced when the soma's membrane potential $V(t)$ exceeds the threshold $V_{\text{th}}$. After firing, the potential resets, and these spikes serve as inputs to the next SPLR convolution layer. This mechanism ensures the model maintains its asynchronous event-driven processing nature while enabling precise temporal modeling across layers.

The **SPLR Convolution Layer** combines the strengths of SA-HiPPO, NPLR Decomposition, and FFT Convolution to process asynchronous spiking inputs effectively. This integration enables the model to extract meaningful spatio-temporal features while maintaining computational efficiency and scalability, making it ideal for high-resolution, real-world applications.

## 9.5 NORMALIZATION AND RESIDUAL

To maintain stability and ensure efficient learning, *Layer Normalization (LN)* is applied after each spiking SSM convolution layer: $\hat{x}_l = \dfrac{x_l - \mu_l}{\sqrt{\sigma_l^2 + \epsilon}} \cdot \gamma + \beta$, where $\mu_l$ and $\sigma_l^2$ are the mean and variance of activations at layer $l$, respectively, and $\gamma, \beta$ are learnable parameters. Normalization reduces variability in activations, providing stable training regardless of fluctuations in inputs.

Additionally, *residual connections* help propagate information across layers by defining $x_{l+1} = f(x_l) + x_l$, where $f(x_l)$ represents the transformation applied by the spiking convolution at layer $l$. Residual connections prevent vanishing gradients, allow lower-level feature retention, and enhance learning efficiency.

Table 12: Input-Output Descriptions for Each Block in the SPLR Model

| Block | Input | Output |
|---|---|---|
| **Input Representation** | Spike events $(x,y,t,p)$: $(x,y)$ (spatial), $t$ (time), $p$ (magnitude/polarity) | Preprocessed spike events for subsequent layers |
| | Spike event stream with spatial and temporal coordinates $(x,y,t,p)$ | Aggregated membrane potential $\mathbf{v}(t)$, capturing spatio-temporal features at multiple timescales. |
| **Dendrite Attention Layer** | *Dendritic Current Update:* Previous dendritic current $\mathbf{i}_d(t)$, synaptic weights $\mathbf{w}_j$, and decay factor $\alpha_d$ | Updated dendritic current $\mathbf{i}_d(t+1) = \alpha_d \mathbf{i}_d(t) + \sum_{j \in \mathcal{N}_d} \mathbf{w}_j p_j$ |
| | *Soma Aggregation:* Inputs from dendritic currents $\mathbf{i}_d(t+1)$, soma decay factor $\beta$, and coupling strengths $\mathbf{g}_d$ | Aggregated membrane potential $\mathbf{v}(t+1) = \beta \mathbf{v}(t) + \sum_d \mathbf{g}_d \mathbf{i}_d(t)$ |
| | *Spike Generation:* | Spike output if $\mathbf{v}(t+1) > V_{\text{th}}$, and reset potential ($\mathbf{v}(t+1) \leftarrow 0$) |
| **Spatial Pooling Layer** | Aggregated spikes $\mathbf{I}(x,y,t)$ from the Dendrite Attention Layer | Pooled spatio-temporal representation $\mathbf{I}_{\text{pooled}}(x',y',t)$, with reduced spatial dimensions |
| | Pooled spike features $\mathbf{I}_{\text{pooled}}(x',y',t)$ | Processed state $\mathbf{y}(t)$, thresholded to generate spikes |
| **SPLR Convolution Layer** | *SA-HiPPO:* Spike features and inter-spike intervals ($\Delta t$) | Adjusted state-space matrix $\mathbf{A_S}$, incorporating memory retention through a decay matrix |
| | *NPLR Decomposition:* Adjusted state-space matrix $\mathbf{A_S}$ | Decomposed matrix $\mathbf{A_S} = \mathbf{V} \mathbf{\Lambda} \mathbf{V}^* - \mathbf{P} \mathbf{Q}^*$, reducing computational complexity |
| | *Matrix Exponential Approximation:* Decomposed state-space matrix $\mathbf{A_S}$, time step $\Delta t_k$ | Approximated exponential $e^{\mathbf{A_S} \Delta t_k}$ for efficient state updates |
| | *FFT Convolution:* State vector $\mathbf{x}(t_k)$ and precomputed impulse response $\mathbf{K}(\omega)$ | Updated state vector $\mathbf{x}(t_{k+1})$ after efficient frequency-domain convolution |
| **Layer Normalization** | Intermediate activations $\mathbf{x}_l$ from the SPLR Convolution Layer | Normalized activations $\hat{\mathbf{x}}_l$, ensuring stable training by reducing variability in activations |
| **Readout Layer** | Normalized features $\hat{\mathbf{x}}_l$ | Final output $\mathbf{y}$, generated via event pooling and a linear transformation |

---

**Algorithm 5** SPLR Convolution Layer

---

**Require:** Spike train input $S(t)$, HiPPO base matrix $\mathbf{A}$, input coupling matrix $\mathbf{B}$, output coupling matrix $\mathbf{C}$, decay function $\mathbf{F}(\Delta t)$, time step $\Delta t$, low-rank matrices $\mathbf{P}$, $\mathbf{Q}$, total time $T$, rank $r$, state space dimension $N$, FFT convolution kernel $\mathbf{K}(\omega)$, threshold potential $V_{\text{th}}$

**Ensure:** Output spike map $Y_{\text{spike}}(t)$

    **Initialization**

1: Initialize state vector $\mathbf{x} \leftarrow 0$            *(N-dimensional state vector)*

2: Initialize output $Y_{\text{spike}} \leftarrow []$         *(Empty list to store spike outputs)*

    **Precomputations**

3: Compute spike-aware HiPPO matrix: $\mathbf{A}_{\text{spike}} \leftarrow \mathbf{A} \circ \mathbf{F}(\Delta t)$     *(Hadamard product with decay function)*

4: Perform eigendecomposition: $\mathbf{V}, \mathbf{\Lambda} \leftarrow \text{eig}(\mathbf{A}_{\text{spike}})$

5: Decompose using NPLR: $\mathbf{A}_{\text{NPLR}} \leftarrow \mathbf{V}\mathbf{\Lambda}\mathbf{V}^* - \mathbf{P}\mathbf{Q}^*$

6: **for** $t = 1$ **to** $T$ **do**

    **Spike-Driven Dynamics**

7:     **if** $S(t)$ contains spikes **then**

8:         Compute time difference: $\Delta t_k = t_{k+1} - t_k$

9:         Approximate matrix exponential:

$$e^{\mathbf{A}_{\text{spike}}\Delta t_k} \approx \mathbf{I} + \mathbf{A}_{\text{spike}}\Delta t_k + \frac{(\mathbf{A}_{\text{spike}})^2(\Delta t_k)^2}{2}$$

10:       Update state vector:

$$\mathbf{x}(t_{k+1}) \leftarrow \mathbf{x}(t_k) + \Delta t_k\left((\mathbf{V}\mathbf{\Lambda}\mathbf{V}^* - \mathbf{P}\mathbf{Q}^*)\mathbf{x}(t_k) + \mathbf{B}S(t_k)\right)$$

11:     **else**

12:         Update state for continuous dynamics: $\mathbf{x} \leftarrow e^{\mathbf{A}_{\text{spike}}\Delta t}\mathbf{x}$

13:     **end if**

    **FFT-Based Convolution for Temporal Dependencies**

14:     Transform state and kernel to frequency domain:

$$\mathbf{X}_{\text{freq}} \leftarrow \text{FFT}(\mathbf{x}), \quad \mathbf{K}_{\text{freq}} \leftarrow \text{FFT}(\mathbf{K}(\omega))$$

15:     Perform element-wise multiplication in frequency domain:

$$\mathbf{Y}_{\text{freq}} \leftarrow \mathbf{X}_{\text{freq}} \cdot \mathbf{K}_{\text{freq}}$$

16:     Transform back to time domain:

$$\mathbf{x}(t_{k+1}) \leftarrow \text{IFFT}(\mathbf{Y}_{\text{freq}})$$

17:     Compute continuous output: $y_t \leftarrow \mathbf{C} \cdot \mathbf{x}(t)$

18:     Threshold the output to generate spikes:

$$y_{\text{spike}}(t) \leftarrow \mathbb{I}(y_t > V_{\text{th}})$$

19:     Append $y_{\text{spike}}(t)$ to $Y_{\text{spike}}$

20: **end for**

    **Output:** $Y_{\text{spike}}$, the final spike map

---

## 9.6 READOUT LAYER

The readout layer is inspired by the *Event-SSM* architecture and employs an *event-pooling mechanism* to subsample the temporal sequence length. The pooled output is computed as $x_{\text{pooled},k} = \frac{1}{p}\sum_{i=kp}^{(k+1)p-1} x_i$, where $p$ is the pooling factor. This operation ensures only the most relevant temporal features are retained, reducing computational burden while preserving key information. The resulting pooled sequence is passed through a linear transformation as $y = Wx_{\text{pooled}} + b$ where $W$ and $b$ are learnable parameters. The combination of event pooling and linear transformation provides an efficient means for deriving a final representation suitable for downstream tasks, maintaining scalability even with longer event sequences.

---

**Algorithm 6** Readout Layer

---

**Require:** State vectors $\{x(t)\}$, pooling factor $p$, weights $W$, bias $b$
1: **for** each pooled time step $k$ **do**
2:        Compute pooled state:

$$x_{\text{pooled},k} \leftarrow \frac{1}{p} \sum_{i=kp}^{(k+1)p-1} x(t_i)$$

3: **end for**
4: Compute final output:

$$y \leftarrow W x_{\text{pooled}} + b$$

5: **Output**: Model prediction $y$

---

## 10 SUPPLEMENTARY SECTION D: RELATED WORKS

**Spiking Neural Networks**

Spiking Neural Networks (SNNs) are biologically inspired models that process information through discrete spike events, offering a more energy-efficient alternative to traditional artificial neural networks (ANNs) Ponulak & Kasinski (2011). They employ learning mechanisms such as spike-timing-dependent plasticity (STDP) for unsupervised training Gerstner & Kistler (2002); Chakraborty & Roy (2023) and surrogate gradient descent for supervised learning Neftci et al. (2019). These approaches have enabled SNNs to be deployed in neuromorphic hardware like TrueNorth Akopyan et al. (2015) and Loihi Davies et al. (2018), achieving substantial energy savings compared to ANNs. Recent efforts to improve the learning capacity of SNNs for long-range temporal dependencies have explored architectures that integrate heterogeneous neuronal dynamics, achieving significant improvements in spatiotemporal tasks Perez-Nieves et al. (2021); Chakraborty & Mukhopadhyay (2022; 2023); She et al. (2021a).

However, many traditional SNNs still struggle to model long-range dependencies effectively. This limitation is often due to the short-term memory characteristics of spiking neuron models, which focus on local temporal processing and struggle with maintaining information over extended periods Bellec et al. (2018); Fang et al. (2023). To address this, recent research has explored the integration of state-space dynamics within SNNs. For instance, Stan and Rhodes Stan & Rhodes (2024) proposed a model that combines state-space models (SSMs) with spiking architectures, demonstrating improved performance in sequence modeling tasks compared to other SNN models. Our work builds on this by introducing a novel state-space approach specifically tailored for efficient, asynchronous processing in neuromorphic contexts, enabling accurate and scalable temporal modeling.

### EVENT-BY-EVENT PROCESSING

Event-based processing in SNNs leverages the asynchronous nature of the spiking activity to process dynamic visual scenes efficiently, a concept widely explored in neuromorphic vision. Prior approaches, such as in Gehrig & Scaramuzza (2024), employ hybrid event- and frame-based systems to capture high-speed, low-latency visual data, while other works like Schöne et al. Schöne et al. (2024) utilize deep state-space models for long-term event-driven data processing. These models manage dynamic temporal dependencies over extensive event sequences, essential for real-time neuromorphic applications. However, a critical limitation remains in scaling these approaches for complex dependencies without excessive computational costs. Our SPLR framework enhances event-driven processing capabilities by integrating state-space dynamics with spike-aware temporal mechanisms, preserving the asynchronous, efficient qualities of SNNs.

### SPIKING NETWORKS WITH DENDRITIC AND TEMPORAL HETEROGENEITY

Temporal dendritic heterogeneity has emerged as a powerful tool to enhance SNNs' temporal processing capabilities. The DH-LIF model by Zheng et al. Zheng et al. (2024b) leverages this heterogeneity to model multi-timescale dependencies within SNNs effectively, achieving robust performance in sequential tasks. Other recent works, Pagkalos et al. (2023); Shen et al. (2024a), propose dendrite-based SNN models, demonstrating how multi-compartment neurons improve computational efficiency by capturing temporal features across diverse timescales. While these

models achieve notable gains in temporal modeling, they often introduce significant computational overhead, limiting scalability. Our SPLR model addresses this by incorporating dendritic-inspired pooling mechanisms that retain temporal features with reduced computational demands, enabling scalable processing for complex neuromorphic tasks.

