# OpenReview forum: "SPLR: A Spiking Neural Network for Long-Range Temporal Dependency Learning"
_ICLR.cc/2025/Conference — Submitted to ICLR 2025_

### Official Review · Reviewer_tvp3 · 2024-10-21

**Soundness:** 3
**Presentation:** 3
**Contribution:** 2
**Rating:** 5
**Confidence:** 3

**Summary:**

This paper integrates state-space models (SSMs) and spiking neural networks (SNNs), and proposes the Spiking Network for Learning Long-Range Relations (SPLR) to enhance the ability of SNNs to capture long-range dependencies. Theoretical proofs and experimental results support the performance advantages of SPLR.

**Strengths:**

1. The integration of SA-HiPPO and SPLR convolution enhances the model's ability to model long-range dependencies.
2. The SPLR model designed in this paper introduces a dendrite-based pooling layer, which further improves the performance using the DH-LIF neuron model.
3. The theoretical and experimental results in this paper confirm the effectiveness of the proposed method.

**Weaknesses:**

1. The core innovation of this paper is the SPLR convolution layer to enhance the long-range temporal modeling capability of SNNs. However, while this improves the performance of the SNN, the integration of the SSM and the SNN requires significant computational overhead, which counteracts the power consumption advantage of the SNN. In addition, as the authors mention SPLR is difficult to implement in hardware, making the significance of this work seem small.

2. Can the authors clarify the key differences between the DH-LIF model in this paper and the one presented in [1]? How does the DH-LIF used in the Dendrite Attention Layer relate to attention? Is the author's claim of dendrite-based spatio-temporal pooling just spatial pooling after the output of DH-LIF? Where is there temporal pooling?

3. I suggest that the authors compare their method with other optimized spiking neural networks capable of long-range temporal modeling, such as [2,3,4].


[1] Zheng H, Zheng Z, Hu R, et al. Temporal dendritic heterogeneity incorporated with spiking neural networks for learning multi-timescale dynamics. Nature Communications, 2024.

[2] Wang L, Yu Z. Autaptic synaptic circuit enhances spatio-temporal predictive learning of spiking neural networks. ICML, 2024.

[3] Zhang S, Yang Q, Ma C, et al. Tc-lif: A two-compartment spiking neuron model for long-term sequential modelling. AAAI, 2024.

[4] Shen S, Zhao D, Shen G, et al. TIM: An efficient temporal interaction module for spiking transformer. IJCAI, 2024.

**Questions:**

Please see weakness.

---

> ### Author Response · Authors · 2024-11-20
> **Rebuttal to Weaknesses Part 1**
>
> > The core innovation of this paper is the SPLR convolution layer to enhance the long-range temporal modeling capability of SNNs. However, while this improves the performance of the SNN, the integration of the SSM and the SNN requires significant computational overhead, which counteracts the power consumption advantage of the SNN.
>
> We emphasize that the primary focus of SPLR is on improving the temporal modeling capabilities in complex event-driven tasks. Our goal is to achieve competitive performance with traditional DNNs but with a much lower computational complexity. We acknowledge that, SPLR has a higher computational cost than baseline SNNs, but achieves much higher performance specially in complex tasks.
>
> As demonstrated in **Figures 3 and 4**, SPLR achieves significantly higher accuracy with comparable or lower FLOPS than state-of-the-art DNN or Transformer based methods., effectively balancing performance and computational cost. This trade-off becomes even more pronounced in **complex cases like HAR-DVS and Celex-HAR**, where SPLR maintains strong performance while scaling efficiently. In these challenging scenarios, SPLR’s structured temporal modeling enables it to outperform traditional SNNs and dense architectures that struggle to handle the large spatial and temporal complexities.
>
> Additionally, the structured dynamics of SSMs reduce computational overhead by providing an efficient representation of long-range dependencies, eliminating the need for densely recurrent operations typical of many spiking architectures. This helps SPLR avoid resource-intensive computations while maintaining strong temporal memory retention.
> ___
>
> > In addition, as the authors mention SPLR is difficult to implement in hardware, making the significance of this work seem small.
>
> We acknowledge the challenges of implementing SPLR in current neuromorphic hardware. However, this work represents an initial exploration of integrating state-space modeling with SNNs to enhance long-range temporal modeling. As demonstrated in Figures 3 and 4, SPLR achieves significant improvements in accuracy vs. FLOPs, particularly on complex datasets like HAR-DVS and Celex-HAR. These results highlight SPLR's computational efficiency and its potential for resource-constrained applications.
> Regarding hardware feasibility, recent advancements in compute-in-memory (CIM) technologies and hybrid analog-digital designs provide a promising path forward for implementing SPLR's state-space aspects. CIM technologies enable efficient matrix operations directly within memory arrays, reducing data movement costs and enhancing the overall efficiency of low-rank decompositions like NPLR. Additionally, FFT-like operations can be mapped to event-driven architectures using asynchronous dataflows, leveraging the sparsity inherent in spiking signals to approximate convolutions effectively (e.g., Akopyan et al., 2015; Davies et al., 2018; Roy et al., 2019).
> The current trend in hardware design increasingly focuses on mixed-mode platforms that integrate digital and analog components, offering flexibility for implementing both spiking and non-spiking operations. Such platforms are well-suited for supporting hybrid approaches like SPLR, enabling efficient execution of state-space modeling and spiking computations in tandem.
> While SPLR introduces challenges for immediate hardware deployment, its demonstrated performance gains and alignment with emerging hardware paradigms make it a strong candidate for future hardware-oriented optimizations. We appreciate the opportunity to address this important point.
>
> ___

---

> ### Author Response · Authors · 2024-11-20
> **Rebuttal to Weaknesses Part 2**
>
> > Can the authors clarify the key differences between the DH-LIF model in this paper and the one presented in [1]?
>
>  The key differences are as follows:
>
> 1. **Temporal Dynamics**:
>    - In the original **DH-LIF model** [1], the focus is on multi-timescale dynamics achieved through dendritic heterogeneity. Each dendritic branch processes inputs with independent timing factors, creating neurons that can effectively capture temporal dynamics across different scales.
>    - In SPLR, **DH-LIF neurons are part of the Dendrite Attention Layer**, where an attention mechanism weighs the contributions of different dendritic branches. Additionally, the **Spike-Aware HiPPO layer** optimizes long-term memory retention by dynamically adjusting memory based on spike timing. This integration extends beyond neuron-level dynamics to address temporal dependencies across long sequences.
>
> 2. **Architectural**:
>    - The original DH-LIF model is primarily used as a standalone neuron type to enhance temporal dynamics in spiking neural networks (SNNs). The design focuses on multi-timescale memory at the neuron level while maintaining manageable complexity.
>    - In SPLR, DH-LIF neurons are embedded within a larger architecture that combines **convolutional and state-space layers**. These layers enable the model to capture long-term dependencies in event-driven tasks while balancing computational efficiency and scalability. The dendritic branches contribute to a pooling mechanism, reducing computational overhead while retaining temporal features.
>
> 3. **Learning**:
>    - In the original DH-LIF model, learning focuses on optimizing the temporal constants of dendritic branches to enhance temporal heterogeneity and facilitate diverse temporal computations.
>    - In SPLR, learning extends beyond dendritic time constants with the inclusion of the Spike-Aware HiPPO layer, which dynamically optimizes memory retention for capturing long-range temporal dependencies. This combination ensures both immediate and extended memory retention, making SPLR particularly effective for tasks requiring hierarchical temporal modeling, such as gesture recognition and spiking-based vision.
>
> The original DH-LIF model emphasizes multi-timescale dynamics at the neuron level, enabling temporal memory through heterogeneous dendritic branches. In contrast, SPLR incorporates DH-LIF neurons into a comprehensive architectural framework, combining attention mechanisms, convolutional layers, and state-space modeling to handle complex temporal tasks with long-range dependencies efficiently. These extensions make SPLR suitable for more demanding applications while maintaining computational efficiency.
> ___
>
> > How does the DH-LIF used in the Dendrite Attention Layer relate to attention?
>
> The DH-LIF neurons in the Dendrite Attention Layer implement a biologically inspired form of attention by leveraging the dendritic branches’ ability to process inputs at multiple timescales. Each dendritic branch has a unique timing factor, allowing the neuron to selectively amplify or suppress input signals based on their temporal properties. This selective processing acts as a form of attention, where the neuron dynamically focuses on the most relevant spatiotemporal features while downplaying less significant ones.
> The aggregation of outputs from multiple dendritic branches enables the network to prioritize temporal features based on their relevance, akin to how attention mechanisms weigh inputs differently to emphasize salient information. This spatio-temporal pooling mechanism not only incorporates spatial aggregation but also applies temporal weighting, enhancing the model’s ability to focus on meaningful temporal patterns.
>
> To clarify this analogy, we have added an explicit discussion on the role of temporal pooling within the DH-LIF architecture in Section 4 and Suppl. Sec. C and highlight how it extends the traditional concept of attention. Thank you for pointing out this area for further elaboration, which we believe will enhance the clarity and impact of the manuscript.
>
> ___

---

> ### Author Response · Authors · 2024-11-20
> **Rebuttal to Weaknesses Part 3**
>
> > Is the author's claim of dendrite-based spatio-temporal pooling just spatial pooling after the output of DH-LIF? Where is there temporal pooling?
>
>  Thank you for raising this point. To clarify, our claim of dendrite-based spatio-temporal pooling refers to both **temporal** and **spatial pooling**, not just spatial pooling after DH-LIF neurons.
>
> - **Temporal Pooling**: Each DH-LIF neuron has multiple dendritic branches with distinct timing factors, acting as temporal filters that capture input dynamics at different timescales. These branches integrate temporal information by selectively emphasizing or suppressing features based on their temporal properties, performing pooling across time before further processing.
> - **Spatial Pooling**: After temporal pooling at the dendritic level, spatial pooling is applied to reduce the spatial dimensionality of the feature maps while preserving the extracted temporal features.
>
> This two-step process allows DH-LIF neurons to first aggregate temporal information through dendritic mechanisms and then reduce spatial complexity, enabling efficient extraction and integration of spatio-temporal features from asynchronous spike inputs.
>
> We have added this distinction in the revised manuscript to ensure the role of temporal pooling at the dendritic level is explicitly highlighted. Thank you for pointing this out.
> ___
>
> > I suggest that the authors compare their method with other optimized spiking neural networks capable of long-range temporal modeling, such as [2,3,4].
>
> We appreciate the reviewer’s suggestion and have carefully considered how our method compares to the referenced models. Below, we outline the key differences and include a quantitative comparison in the table provided.
>
> 1. **Autaptic Synaptic Circuit (Wang & Yu, ICML 2024)**:
>    The Autaptic Synaptic Circuit enhances temporal memory and spatial coordination using two adaptive pathways. While similar in leveraging learnable timing factors, our approach extends this by incorporating a dendrite-level attention mechanism, allowing dynamic weighting of spatio-temporal features and enabling diverse temporal processing at the dendritic level.
>
> 2. **TC-LIF: Two-Compartment Spiking Neuron Model (Zhang et al., AAAI 2024)**:
>    TC-LIF uses separate dendritic and somatic compartments to model long-term temporal dependencies. Our DH-LIF neurons extend this concept by introducing heterogeneous timing factors across dendritic branches, which support richer temporal diversity and are further enhanced by our dendrite-based pooling mechanism for spatio-temporal integration.
>
> 3. **TIM: Temporal Interaction Module (Shen et al., IJCAI 2024)**:
>    TIM integrates into spiking transformers to handle temporal information by combining historical and current inputs. In contrast, our model focuses on biologically inspired processing at the dendritic level, where temporal diversity is handled through the DH-LIF neurons and dendritic attention rather than modifications to the transformer’s attention mechanism.
>
> | **Dataset** | **TIM [2] Params** | **TIM [2] Acc** | **TCLIF [3] Params** | **TCLIF [3] Acc** | **STC-LIF [4] Params** | **STC-LIF [4] Acc** | **SPLR Normal Params** | **SPLR Normal Acc** | **SPLR-Tiny Params** | **SPLR-Tiny Acc** |
> |-------------|---------------------|-----------------|-----------------------|-------------------|-------------------------|---------------------|-------------------------|---------------------|----------------------|-------------------|
> | SHD         | 2.59M              | 86.3            | 0.142M               | 88.91             |                  |                | 0.513M                 | 94.68              | 0.033M               | 86.24            |
> | DVS128      |                     |                 |                      |                   |             3.922M            |          83.0          |         0.513M                 | 96.5               |          0.033M            | 89.2             |
> | SSC         | 110.8K             | 61.09           |                      |                   |                         |                     |           0.513M              | 87.52              |          0.033M            | 72.19            |
>
>
> Our SPLR model combines dendritic heterogeneity, adaptive attention, and spatio-temporal pooling to balance biological inspiration and computational efficiency. As demonstrated in the table, SPLR achieves higher accuracy with significantly fewer parameters in most cases, particularly on complex datasets such as SHD and DVS128. We will include quantitative comparisons and expand this discussion in the revised manuscript to highlight our method's advantages in long-range temporal modeling.
>
> ___
> Edit: We updated the table for the STC-LIF numbers are for DVS-Gesture and not for the SHD dataset

---

### Official Review · Reviewer_TiCG · 2024-11-02

**Soundness:** 2
**Presentation:** 1
**Contribution:** 2
**Rating:** 3
**Confidence:** 4

**Summary:**

This work proposes a spike SSM method named Spiking Network for Learning Long-Range Relations (SPLR). The proposed SPLR convolutional layer leverages state-space dynamics to enhance feature extraction while retaining the efficiency of sparse, event-based
processing, and incorporates a Spike-Aware HiPPO (SA-HiPPO) matrix that allows SPLR to effectively maintain long-range memory by adapting the HiPPO framework for discrete, spike-driven inputs. The authors tested their method on several datasets, such as Celex HAR, DVS128 Gesture, Sequential CIFAR-10/100

**Strengths:**

1. This work explores the combination of Spiking Neural Networks (SNN) and State Space Models (SSM), which is an interesting direction. Using state-space methods to improve SNNs' ability to model long-term dependencies holds promise.
2. The experimental comparisons in this work are extensive.

**Weaknesses:**

This work requires comprehensive improvements, with the main weaknesses outlined as follows.

1. Writing: The writing in this work requires careful and comprehensive improvement, covering the overall organization of the paper, paragraph structure, and numerous details that need refinement. (1) The authors placed the related work section in the supplementary materials and omitted citations to many key works, which can confuse readers. For instance, the authors did not cite relevant papers when HIPPO was first mentioned (in fact, there are almost no citations in paragraphs 3 and 4 of the introduction). (2) The methodology section is not clearly explained. What type of spiking neurons does this work use? How does SPLR integrate with spiking neurons? The authors repeat content introduced in the main text within the supplementary materials. Lines 147-150 reiterate the significance of SPLR, but readers are likely more interested in the methodological details and the rationale behind the proposed approach’s significance. Unfortunately, these critical details are missing. (3) In the theoretical discussion, the authors present several theorems but do not clarify why these are necessary. While sparse properties and FLOPS reduction are mentioned, the evaluation details are not provided.
(4) The supplementary materials contain excessive repetition and are overly lengthy, making it difficult for readers to stay engaged. (5) Overuse of Abbreviations: The excessive use of abbreviations makes the paper difficult to follow. For instance, I cannot understand why "Spiking Network for Learning Long-Range Relations" is abbreviated as "SPLR"—what does "P" represent here? Is there a difference between "Spiking Network" and "Spiking Neural Networks (SNNs)"? Additionally, what does HIPPO stand for? Shouldn’t the authors explain these abbreviations first? (6) Overstatements: The paper is filled with terms like "spike-driven," "asynchronous," and "real-time." As I understand it, "spike-driven" implies a purely additive network[2], yet the pink section in Figure 1 seems unable to achieve this. Regarding "asynchronous," the authors’ explanation in lines 92-95 is too brief, making it difficult to discern what kind of preprocessing the network applies to the data.

2. Motivation. The authors repeatedly state that the proposed SPLE can address the challenge of modeling both short- and long-term dependencies in SNNs. However, they fail to analyze why SNNs have limitations in this area and why their proposed method can solve this issue. For instance, this is mentioned in lines 58-67, 147-149, and 1846-1850 without providing the necessary analysis.

3. Innovation.The originality of this work is limited. The dendrite modeling directly uses DH-LIF, and the SSM modeling is simply an SNN combined with HIPPO. I did not observe any standout contributions in this approach.

4. Experiments. The datasets chosen by the authors, DVS128 Gesture and Sequential CIFAR-10/100, do not effectively test the model's ability to handle long-range dependencies. The authors could consider more challenging datasets, such as LRA. Additionally, the fact that no one in the SNN field has addressed the Celex HAR dataset does not imply that SNNs cannot handle it. The authors have not even provided a complete description of the size and scope of the Celex HAR dataset. If the authors aim to compare with other SNNs on challenging DVS datasets, they might try HAR-DVS[1]. Furthermore, the authors overlook comparisons with many recent SOTA methods on the Gesture dataset, where SNN performance has already surpassed 99.5%[2].

---
[1] Hardvs: Revisiting human activity recognition with dynamic vision sensors. In AAAI 2024.
[2] Spike-driven transformer. In NeurIPS 2023.

**Questions:**

Please see weaknesses.

---

> ### Author Response · Authors · 2024-11-20
> **Rebuttal to Weaknesses Part 1**
>
> > This work requires comprehensive improvements, with the main weaknesses outlined as follows.
> Writing: The writing in this work requires careful and comprehensive improvement, covering the overall organization of the paper, paragraph structure, and numerous details that need refinement.
>
>  Thank you for your detailed feedback on the writing and organization of the manuscript. Below are the key improvements we have implemented:
>
> 1. **Restructured Introduction**:    The introduction has been restructured to provide a more cohesive narrative, emphasizing the motivation, key contributions, and the broader significance of the work.
>
> 2. **Overview in Methods**:   We added a high-level overview at the beginning of the methods section to guide readers through the SPLR architecture and its components, improving logical flow and understanding.
>
> 3. **Enhanced Theoretical Section**:    Intuitive explanations for key theorems and lemmas have been added to Section 3, making the theoretical content more accessible and easier to follow. These explanations contextualize the results and connect them to the SPLR model's design.
>
> 4. **Expanded Description of Methods**:  The methods section has been expanded with additional details to improve clarity and provide a deeper understanding of the proposed techniques.
>
> 5. **Restructured Supplementary Material**:
>    - **Notation and Symbols**: A new subsection lists and defines all symbols, variables, and notations used in the paper, reducing ambiguity.
>    - **Dataset Details**: Detailed descriptions of the datasets used have been added for completeness.
>    - **Reduced Redundancies**: Repeated content has been streamlined for better readability.
>    - **Expanded Related Works**: The related works section has been expanded to provide a comprehensive context for the contributions.
>
> We believe these revisions significantly improve the paper's writing, organization, and readability. Thank you for highlighting these concerns, which allowed us to refine and strengthen the presentation of our work.
>
> ---
>
> > (1) The authors placed the related work section in the supplementary materials and omitted citations to many key works, which can confuse readers. For instance, the authors did not cite relevant papers when HIPPO was first mentioned (in fact, there are almost no citations in paragraphs 3 and 4 of the introduction).
>
>  To address these concerns, we have made the following revisions:
>
> 1. **Adding Relevant Citations to the Introduction**:  While the detailed related work section remains in the supplementary materials for a more comprehensive discussion, we have added the most relevant citations directly in the introduction. Specifically, the foundational works on the HiPPO framework are now cited where it is first mentioned. Also, other critical methods and concepts referenced in paragraphs 3 and 4 of the introduction now include appropriate citations, directing readers to key prior studies.
>
> 2. **Revised Introduction for Clarity**:  We have revised the introduction to better integrate these citations into the narrative.
>
> 3. **Broader Coverage of Relevant Work**:  Beyond the introduction, we have reviewed and enhanced citations throughout the manuscript to ensure comprehensive acknowledgment of foundational contributions.
>
> We believe these revisions address the reviewer’s concerns and improve the clarity, accessibility, and rigor of the paper. Thank you for highlighting this issue and helping us strengthen our manuscript.

---

> ### Author Response · Authors · 2024-11-20
> **Rebuttal to Weaknesses Part 2**
>
> >(2) The methodology section is not clearly explained. What type of spiking neurons does this work use? How does SPLR integrate with spiking neurons? The authors repeat content introduced in the main text within the supplementary materials. Lines 147-150 reiterate the significance of SPLR, but readers are likely more interested in the methodological details and the rationale behind the proposed approach’s significance. Unfortunately, these critical details are missing.
>
> Thank you for this valuable feedback.
>
> Type of Spiking Neurons Used: In this work, we primarily use DH-LIF (Dendritic Heterogeneous Leaky Integrate-and-Fire) neurons, which allow for multi-compartment modeling through dendritic branches. These neurons enhance the network’s ability to capture temporal dynamics across different timescales, essential for managing short- and long-term dependencies in event-driven data. We have updated Section 2 and Suppl Sec C to clarify this.
>
> Integration of SPLR with Spiking Neurons: SPLR integrates with DH-LIF neurons by using their dendritic branches for temporal filtering and memory retention. The SPLR Convolution layer leverages state-space dynamics to process inputs asynchronously, updating based on spikes generated by DH-LIF neurons. Additionally, the Spike-Aware HiPPO (SA-HiPPO) layer complements this by adjusting memory retention dynamically, allowing SPLR to capture long-term dependencies without introducing excessive computational overhead. We have provided a detailed step-by-step explanation of how SPLR connects to spiking neuron outputs and processes event-driven data in Section 2 (Methods) and Suppl Sec C.
>
> Reducing Redundancy and Emphasizing Rationale: We have streamlined the content to minimize repetition and focus on explaining the rationale behind each component of SPLR. We have highlighted the advantages of integrating DH-LIF neurons, state-space modeling, and SA-HiPPO in a unified framework and their collective contribution to achieving efficient long-term memory retention in spiking networks.
>
> ---
>
> > (3) In the theoretical discussion, the authors present several theorems but do not clarify why these are necessary.
>
>  We have made the following revisions:
>
> 1. The theorems in the theoretical discussion are intended to formally justify the efficiency, memory retention, and stability capabilities of the SPLR model. Specifically:
>    - **Lemma on Computational Complexity**: This result establishes that the SPLR model’s computational cost per spike is $O(N^2)$. This underscores the importance of techniques like NPLR decomposition to reduce FLOPS and improve scalability for real-time applications.
>    - **Theorem on Temporal Dependency Preservation**: This theorem demonstrates the SPLR model’s ability to retain long-range dependencies, a core limitation in standard spiking neural networks. By controlling the decay of older information, the model ensures that recent spikes have a stronger influence on the system state.
>    - **Lemma on Error Bounds for Matrix Exponential Approximation**: This lemma ensures that our approximations of the state-space dynamics using a Taylor expansion are both efficient and accurate, enabling practical deployment in resource-constrained settings.
>    - **Theorem on Bounded State Trajectories**: This result guarantees the stability of the SPLR model, ensuring that its internal state remains bounded under spike-driven inputs, which is essential for continuous, real-time processing.
>
> 2.  We have added **intuitive explanations** after each lemma and theorem. These explanations connect the theoretical results to the SPLR model’s design and objectives, such as efficient temporal processing, memory retention, and stability.
>
> 3. We have added an introductory paragraph to the theoretical discussion, outlining its goals and linking the results to the broader context of the SPLR model. Additionally, we included a summary at the end of the section to tie the results back to practical outcomes, such as computational efficiency, FLOPS reduction, and robust handling of sparse event-driven data.

---

> ### Author Response · Authors · 2024-11-20
> **Rebuttal to Weaknesses Part 3**
>
> > While sparse properties and FLOPS reduction are mentioned, the evaluation details are not provided.
>
>  Thank you for highlighting the need to provide clearer evaluation details regarding FLOPS reduction and sparsity. We recognize the importance of explicitly connecting these properties to the empirical results, and we have clarified this in the revised manuscript as follows:
>
> 1. SPLR achieves **significant FLOPS reduction** compared to state-of-the-art methods, as demonstrated in Figures 2 and 3. These results are particularly pronounced on high-resolution, event-based datasets like DVS Gesture and Celex-HAR, where SPLR variants (Normal, Small, Tiny) provide superior accuracy and computational efficiency. For instance:
>    - SPLR Tiny achieves a FLOPS count as low as 0.034 GFLOPs on Celex-HAR while maintaining competitive accuracy, showcasing its suitability for resource-constrained applications.
>
> 2. The **FLOPS-accuracy trade-off** illustrated in Figures 2 and 3 aligns with our theoretical predictions about SPLR’s efficiency. Key contributing factors include:
>    - **NPLR Decomposition and FFT Convolutions**: These components allow SPLR to handle high-dimensional inputs with significantly reduced computational costs compared to standard dense convolution methods.
>    - **Sparsity-Driven Efficiency**: By leveraging the asynchronous, sparse nature of event-driven data, SPLR reduces redundant computations, further enhancing its efficiency for real-time applications.
>
> 3. The event-driven processing capabilities of SPLR naturally exploit **sparsity in the input**, enabling selective updates and reducing overall computational overhead. This property is evident in SPLR’s ability to maintain both high accuracy and low computational cost across various datasets.
>
>
> > (4) The supplementary materials contain excessive repetition and are overly lengthy, making it difficult for readers to stay engaged.
>
>  We have revised the supplementary materials as follows:
>
> 1. Repetitive sections have been consolidated to streamline the content and **reduce redundancies.** We have prioritized the inclusion of essential information and moved less critical details, such as exhaustive derivations and auxiliary results, to the supplementary materials.
>
> 2.  The **supplementary materials have been reorganized** into clearly defined sections with concise summaries. This makes it easier for readers to navigate specific topics, such as dataset descriptions, theoretical proofs, or experimental setups.
>
> 3. Additional formatting changes, such as the inclusion of a table of symbols and variables, have been implemented to **improve readability** and accessibility for readers.
>
> ---
>
> > (5) Overuse of Abbreviations: The excessive use of abbreviations makes the paper difficult to follow. For instance, I cannot understand why "Spiking Network for Learning Long-Range Relations" is abbreviated as "SPLR"—what does "P" represent here? Is there a difference between "Spiking Network" and "Spiking Neural Networks (SNNs)"? Additionally, what does HIPPO stand for? Shouldn’t the authors explain these abbreviations first?
>
> We have reviewed all abbreviations in the manuscript to ensure they are clearly defined at first use, reducing unnecessary jargon to enhance readability.
>
> 1. The abbreviation "SPLR" stands for "SPiking Network for learning Long-Range Relations,"
>
> 2. "Spiking Neural Network (SNN)" refers to the general category of spiking neural architectures, while "Spiking Network" in "SPLR" was used for brevity. We have revised the text to consistently use "Spiking Neural Network" (SNN) for clarity.
>
> 3. HiPPO stands for "Highly Optimized Polynomial Projection." This framework provides memory retention in continuous-time models. [1]
>
> **References**
> - Gu, A., Dao, T., Ermon, S., Rudra, A. and Ré, C., 2020. Hippo: Recurrent memory with optimal polynomial projections. Advances in neural information processing systems, 33, pp.1474-1487.
>
> ---

---

> ### Author Response · Authors · 2024-11-20
> **Rebuttal to Weaknesses Part 4**
>
> > (6) Overstatements: The paper is filled with terms like "spike-driven," "asynchronous," and "real-time." As I understand it, "spike-driven" implies a purely additive network[2], yet the pink section in Figure 1 seems unable to achieve this. Regarding "asynchronous," the authors’ explanation in lines 92-95 is too brief, making it difficult to discern what kind of preprocessing the network applies to the data.
>
>
> 1. **"Spike-Driven":**   In the context of our SPLR model, "spike-driven" refers to the processing of discrete spike events as opposed to continuous signals. Although SPLR incorporates advanced mechanisms such as state-space modeling and the Spike-Aware HiPPO (SA-HiPPO) layer for long-range memory retention, it fundamentally operates on spike events, making it a spike-driven network.
>
>    The term "purely additive" might traditionally describe models that directly update their states by summing incoming spikes in a straightforward manner. While SPLR involves more complex operations, such as state-space dynamics and temporal memory retention, these are applied within an event-driven and spike-based framework. Thus, the model retains its spike-driven nature while leveraging advanced techniques to enhance its capabilities.
>
> 2. **"Asynchronous":**   To clarify, "asynchronous" in SPLR denotes the ability to process inputs based on the timing of spike events rather than relying on synchronized updates or fixed clock cycles. SPLR processes spikes event-by-event as they arrive, without preprocessing them into continuous signals or frames. This ensures that the temporal resolution of the spike inputs is preserved.
>
>    We have added a detailed discussion in the **experimental setup subsection**, clarifying that asynchronous processing in SPLR refers specifically to event-by-event updates, avoiding frame accumulation and maintaining the spike-driven paradigm.
>
> 3. **Revisions to Figure 1:**   We have revised the figure caption to explicitly state that all operations, including those in the pink section, are driven by spike events. The pink section represents components of the SPLRConv layer, such as SA-HiPPO, which operate within the spike-driven paradigm while incorporating mechanisms for improved temporal processing and memory retention.
>
> ---

---

> ### Author Response · Authors · 2024-11-20
> **Rebuttal to Weaknesses Part 5**
>
> > Motivation. The authors repeatedly state that the proposed SPLE can address the challenge of modeling both short- and long-term dependencies in SNNs. However, they fail to analyze why SNNs have limitations in this area and why their proposed method can solve this issue. For instance, this is mentioned in lines 58-67, 147-149, and 1846-1850 without providing the necessary analysis.
>
>  We outline the gaps in conventional SNNs and how SPLR addresses these challenges (Introduction, Suppl Sec C):
>
> 1. **Limitations of Standard SNNs in Modeling Short- and Long-Term Dependencies**:
>    Traditional SNNs, particularly those employing simple neuron models such as the Leaky Integrate-and-Fire (LIF) neuron, encode temporal information using exponentially decaying membrane potentials. While this is effective for short-term memory, it inherently leads to rapid information loss, making it difficult to capture long-term dependencies. These limitations are further compounded by the lack of structured mechanisms for explicit temporal modeling, as most SNNs rely on local spike interactions without incorporating global temporal dependencies.
>
>    For example, conventional SNNs lack the ability to dynamically adapt memory retention to the temporal characteristics of the input, which is critical for tasks requiring integration of information across multiple timescales, such as event-based gesture recognition or sequential classification.
>
> 2. **How SPLR Addresses These Challenges**:
>    SPLR introduces two key innovations to overcome the limitations of traditional SNNs:
>    - **Spike-Aware HiPPO (SA-HiPPO):**
>      SA-HiPPO extends the HiPPO framework to operate in the discrete, asynchronous setting of SNNs. By dynamically adjusting memory retention based on the timing of incoming spikes, it enables the network to prioritize recent information while maintaining a compressed representation of past events. This structured memory retention mechanism ensures that long-term dependencies can be captured without excessive computational overhead.
>    - **State-Space Dynamics in SPLR Convolution:**
>      The SPLR Convolution layer integrates state-space modeling, enabling continuous temporal processing with event-driven updates. Unlike standard SNNs that rely solely on simple integration mechanisms, SPLR leverages structured state-space dynamics to handle both short-term and long-term dependencies efficiently, allowing it to process complex temporal patterns across varying timescales.
>
> 3. **Revisions to Address This Concern**:
>   We have included a discussion on the limitations of traditional SNNs in modeling long-term dependencies due to their short memory span and the lack of explicit temporal mechanisms.
>  We have expanded on how SPLR Convolution and SA-HiPPO overcome these limitations by providing structured mechanisms for long-term memory retention and efficient temporal modeling.
>
>
> **References**
>
> 1. Shen S, Zhao D, Shen G, et al. TIM: An efficient temporal interaction module for spiking transformer. IJCAI, 2024.
> 2. Zhang S, Yang Q, Ma C, et al. Tc-lif: A two-compartment spiking neuron model for long-term sequential modelling. AAAI, 2024.
> 3. Wang L, Yu Z. Autaptic synaptic circuit enhances spatio-temporal predictive learning of spiking neural networks. ICML, 2024.
>
> ---

---

> ### Author Response · Authors · 2024-11-20
> **Rebuttal to Weaknesses Part 6**
>
> > Innovation.The originality of this work is limited. The dendrite modeling directly uses DH-LIF, and the SSM modeling is simply an SNN combined with HIPPO. I did not observe any standout contributions in this approach.
>
> We respectfully disagree with the assessment that the originality of this work is limited. While our work builds on foundational elements such as DH-LIF and HiPPO, the key innovations lie in how these components are adapted, extended, and integrated into a unified framework specifically tailored for spiking neural networks (SNNs). Below, we highlight the unique contributions of this work:
>
> 1. **Spike-Aware HiPPO (SA-HiPPO):**
>    - The HiPPO framework has been extensively used for memory retention in continuous-time models. However, adapting it for the sparse and asynchronous nature of spiking inputs required significant innovation.
>    - SA-HiPPO introduces a decay matrix that dynamically adjusts memory retention based on the timing of incoming spikes. This extension ensures that recent spikes are prioritized while retaining a compressed representation of older information, enabling effective long-term memory retention in SNNs.
>    - This adaptation is crucial for leveraging HiPPO in event-driven systems, and to the best of our knowledge, this is the first work to introduce this mechanism in the context of spiking neural networks.
>
> 2. **SPLR Convolution Layer:**
>    - Our SPLR Convolution Layer integrates spike-driven state-space dynamics with advanced techniques such as Normal Plus Low-Rank (NPLR) decomposition and FFT-based convolutions.
>    - While NPLR and FFT have been used in other contexts, their combination within a spike-driven framework is novel. This enables the model to achieve computational efficiency and scalability without sacrificing the ability to capture complex spatio-temporal dependencies.
>    - The use of state-space dynamics with event-driven updates provides a structured mechanism for handling both short- and long-term dependencies in SNNs, setting SPLR apart from conventional approaches.
>
> 3. **Unified Architecture:**
>    - SPLR represents a cohesive framework that seamlessly integrates dendritic mechanisms (via DH-LIF), SA-HiPPO, and spike-driven state-space dynamics. This integration was specifically designed to address two key challenges in SNNs: long-term memory retention and asynchronous processing.
>    - Unlike prior works, which primarily adapt artificial neural network (ANN) architectures to spiking formats, SPLR is built from the ground up to natively handle the unique constraints and opportunities of spiking computation.
>
> 4. **Substantial Experimental Improvements:**
>    - Our experimental results demonstrate that SPLR achieves substantial improvements over state-of-the-art methods across multiple event-based benchmarks (e.g., DVS Gesture, Celex-HAR, SSC). The model excels in tasks requiring long-range temporal dependencies while maintaining computational efficiency, highlighting the practical advantages of our innovations.
>    - These results underscore the impact of combining SA-HiPPO, state-space modeling, and dendritic mechanisms within a unified spiking framework.
>
> **Revisions to Manuscript:**
> To address potential misunderstandings about the novelty of our work, we will:
> - Expand the discussion in the **Introduction** and **Methods** sections to emphasize the unique aspects of SA-HiPPO, SPLR Convolution, and the unified architecture.
> - Highlight the differences between SPLR and prior work, particularly in terms of memory retention mechanisms, computational efficiency, and spiking-specific adaptations.
> - Include additional details in the **Related Work** section to contextualize our contributions relative to DH-LIF, HiPPO, and other state-space modeling approaches.
>
> We believe these innovations collectively represent a significant advancement in the field of spiking neural networks and event-based processing. By bridging the gap between spiking computation and state-space modeling, SPLR establishes itself as a novel and impactful contribution. We hope this explanation clarifies the originality and significance of our work, and we would be glad to address any further questions or concerns.

---

> ### Author Response · Authors · 2024-11-20
> **Rebuttal to Weaknesses Part 7**
>
> > Experiments. The datasets chosen by the authors, DVS128 Gesture and Sequential CIFAR-10/100, do not effectively test the model's ability to handle long-range dependencies. The authors could consider more challenging datasets, such as LRA.
>
> Following this feedback, we conducted additional experiments on the LRA benchmark, which comprises tasks specifically designed to evaluate models' ability to process long-range sequences.
>
> The results of these experiments are summarized in Table 1, where SPLR is compared against state-of-the-art spiking and non-spiking architectures. SPLR achieves competitive performance across multiple LRA tasks while maintaining the advantages of spiking neural networks, such as energy efficiency and event-driven processing.
>
> - **Replacement of Results**:  The LRA results now replace the original Sequential CIFAR-10/100 results in Section 4.1 and Table 1 of the paper. For reference, the Sequential CIFAR results have been moved to Supplementary Section B.
>
> - **Validation of Long-Range Dependency Handling**:   SPLR's performance on the LRA benchmark provides strong evidence of its ability to model long-range dependencies effectively. For example, SPLR outperforms several state-of-the-art methods in tasks such as Retrieval and Pathfinder, demonstrating the robustness of its memory retention and temporal modeling capabilities.
>
> Additionally, we include new results on the HARDVS dataset, which further validate SPLR's efficiency and effectiveness in real-world event-based scenarios. These results are summarized in Table 2.
>
> **Summary of Contributions**:
> 1. SPLR demonstrates significant improvements in long-range temporal modeling, as validated by its strong performance on the LRA benchmark.
> 2. SPLR achieves these results while maintaining its advantages as a spiking neural network, offering a balance of accuracy, efficiency, and energy savings.
> 3. The inclusion of both LRA and HARDVS results showcases SPLR's versatility across benchmarks designed for different spatio-temporal challenges.
>
> ___
>
> > Additionally, the fact that no one in the SNN field has addressed the Celex HAR dataset does not imply that SNNs cannot handle it. The authors have not even provided a complete description of the size and scope of the Celex HAR dataset.
>
> We do not claim that other SNN models are incapable of handling Celex-HAR; rather, our work
> represents the **first demonstration of strong performance on this dataset using SNNs**,
> establishing an important benchmark for future research. We have modified the text to avoid any confusion on this claim.
>  To address this, we have revised the manuscript to include a complete description of the dataset’s size and scope in **Section B**. The details provided are as follows:
>
> - Celex-HAR contains high-definition (1280×800 pixels) event streams, making it one of the **largest and most spatially complex datasets in the HAR** (Human Activity Recognition) domain. Each sample consists of fine-grained spatio-temporal patterns, which require models to process both high-resolution spatial features and long-range temporal dependencies efficiently.
>
> -  Celex-HAR is particularly challenging due to its combination of:
>   - **High spatial resolution**: Event streams are significantly larger than those in commonly used datasets like DVS Gesture or HAR-DVS.
>   - **Complex temporal patterns**: The dataset includes nuanced activity recognition tasks that demand robust memory retention across varying timescales.
>   These factors likely contribute to the lack of benchmarks in the SNN field and limited exploration in the broader neuromorphic computing community.
>
> -  Our experiments demonstrate that while current deep neural network (DNN) models that perform well on datasets such as HAR-DVS struggle to scale to the size and complexity of Celex-HAR, **SPLR achieves strong performance with significantly reduced computational costs**. SPLR’s spike-driven, state-space approach is particularly effective in handling the large spatial and temporal scales of this dataset.
>
> ___

---

> ### Author Response · Authors · 2024-11-20
> **Rebuttal to Weaknesses Part 8**
>
> > If the authors aim to compare with other SNNs on challenging DVS datasets, they might try HAR-DVS[1].
>
>  Thank you for suggesting the HAR-DVS dataset as a challenging benchmark for evaluating SPLR. Following this recommendation, we have conducted experiments on HAR-DVS, a dataset designed for event-based human activity recognition, to validate SPLR’s performance further.
>
> | **Model**                      | **GFLOPs** | **Accuracy (%)** |
> |--------------------------------|------------|-------------------|
> | C3D [Tran et al. (2015)]       | 0.1        | 50.52            |
> | R2Plus1D [Tran et al. (2018)]  | 20.3       | 49.06            |
> | TSM [Lin et al. (2019)]        | 0.3        | 52.63            |
> | ACTION-Net [Wang et al. (2021)]| 17.3       | 46.85            |
> | TAM [Liu et al. (2021)]        | 16.6       | 50.41            |
> | V-SwinTrans [Liu et al. (2022c)]| 8.7       | 51.91            |
> | SlowFast [Feichtenhofer et al. (2019)] | 0.3 | 46.54        |
> | ESTF [Wang et al. (2024b)]     | 17.6       | 51.22            |
> | ExACT [Zhou et al. (2024)]     | 13.2        | 90.1           |
> | **SPLR-Tiny [Ours]**           | 0.034      | 70.38           |
> | **SPLR-Small [Ours]**          | 0.13       | 81.73            |
> | **SPLR-Normal [Ours]**         | 0.41       | 88.29            |
>
> ### **Key Findings**:
> 1. **Accuracy**: SPLR achieves competitive accuracy across all configurations, with SPLR-Normal reaching **88.29%**
> 2. **Efficiency**: SPLR achieves significantly lower FLOP counts compared to state-of-the-art methods such as R2Plus1D, ACTION-Net, and ESTF. Its modular design allows flexibility (Tiny, Small, Normal) while scaling performance.
> 3. **Event-Based Processing**: It is important to note that HAR-DVS provides frame-based data (raw event data was unavailable for download). Since SPLR is designed for event-by-event processing, we treated all events arriving at the same timestamp as a single batch for processing, adhering to the event-driven principles of the model.
>
> We have incorporated these results into the **main text**, alongside results for **DVS Gesture** and **Celex-HAR**, as shown in **Figure 2**. Detailed experimental setups and additional analysis are provided in **Supplementary Section B**.
>
> ___
>
>
> > Furthermore, the authors overlook comparisons with many recent SOTA methods on the Gesture dataset, where SNN performance has already surpassed 99.5%[2].
>
>  Thank you for your feedback. We have now explicitly included the **Spike-Driven Transformer [2]** as a datapoint in the **DVS Gesture results plot (Figure 3)**. Additional details are also provided in **Supplementary Section C**. While we acknowledge that several models exceed 99% accuracy on DVS Gesture, many rely on **frame-based event accumulation**, which introduces latency and preprocessing overhead. In contrast, SPLR employs an **event-by-event processing strategy**, enabling real-time inference with significantly lower computational costs.
>
> As shown in **Figure 3**, SPLR achieves **96.5% accuracy** on DVS Gesture with reduced computational overhead. Furthermore, SPLR demonstrates its strength on larger and more complex datasets, such as **Celex-HAR**, where frame-based models often struggle to scale effectively.
>
> ___

---

### Official Review · Reviewer_LQNp · 2024-11-04

**Soundness:** 3
**Presentation:** 3
**Contribution:** 2
**Rating:** 8
**Confidence:** 5

**Summary:**

This paper presents a Spiking Network for Learning Long-Range Relations (SPLR). The proposed SPLR model comprises the dendrite attention layer, the Spike-Aware HiPPO (SA-HiPPO) layer, and the SPLR convolution layer. These modules enhance the long-range temporal dependency learning capability of SPLR. Experimental results demonstrate that SPLR outperforms prior methods in tasks requiring both fine-grained temporal dynamics and the retention of long-range dependencies.

**Strengths:**

1. This paper is well-written and technically solid. Each module in SPLR is introduced in detail and highlighted in different colors. This paper presents a detailed theoretical analysis, including the long-range dependency capability and stability of SPLR.
2. The proposed SPLR model achieves competitive accuracy with less computational overhead than other state-of-the-art models on the Celex-HAR dataset.

**Weaknesses:**

The proposed SPLR model incorporates several non-spike operations, including the NPLR decomposition and FFT convolution. It makes SPLR a hybrid architecture instead of a pure spiking neural network. The hybrid nature may compromise its hardware compatibility and make it difficult to deploy on neuromorphic hardware.

**Questions:**

1. This paper only compares FLOPs vs. accuracy between the proposed SPLR and other models. Does SPLR have an advantage over other methods in terms of inference latency?
2. The ablation studies only examine the effects of removing the dendrite attention layer and replacing SA-HiPPO with LIF. What if we replace NPLR decomposition and FFT convolution with standard convolution?

---

> ### Author Response · Authors · 2024-11-20
> **Rebuttal to Weaknesses**
>
> > The proposed SPLR model incorporates several non-spike operations, including the NPLR decomposition and FFT convolution. It makes SPLR a hybrid architecture instead of a pure spiking neural network. The hybrid nature may compromise its hardware compatibility and make it difficult to deploy on neuromorphic hardware.
>
>
> We appreciate the reviewer's comments regarding the hybrid nature of the proposed SPLR model. We acknowledge that SPLR incorporates non-spike operations such as NPLR decomposition and FFT convolution, contributing to its hybrid architecture. However, these components are essential to achieving the desired temporal processing capabilities and long-range dependencies in a computationally efficient manner.
> Although recent studies have not explicitly demonstrated Normal Plus Low-Rank (NPLR) decomposition or FFT in neuromorphic hardware, they have shown similar operations can be implemented efficiently. For example, hybrid analog-digital co-processing approaches have demonstrated promising results in executing low-rank matrix operations in a spike-compatible manner (e.g., Akopyan et al., 2015; Davies et al., 2018). Additionally, FFT-like operations can be mapped to neuromorphic platforms using event-driven dataflows, leveraging asynchronous processing to approximate convolution effectively (e.g., Roy et al., 2019). These advancements suggest that such non-spike components can potentially be adapted to neuromorphic architectures. For instance, approaches like hybrid analog-digital co-processing have shown promising results in executing low-rank matrix operations in a spike-compatible manner (e.g., Akopyan et al., 2015; Davies et al., 2018). Similarly, FFT operations can be mapped to neuromorphic platforms using event-driven dataflows, leveraging asynchronous processing to approximate convolution effectively (e.g., Roy et al., 2019). These advancements suggest that the non-spike components in our model can be adapted to neuromorphic architectures, ensuring better compatibility without compromising computational efficiency.
> The feasibility of implementing NPLR decomposition and FFT in neuromorphic hardware lies in leveraging compute-in-memory (CIM) technologies and analog-digital hybrid designs. CIM technologies enable efficient matrix operations directly within memory arrays, which can significantly reduce data movement costs and enhance the overall efficiency of low-rank decompositions. Similarly, FFT operations can be implemented using parallel event-driven architectures that take advantage of the inherent sparsity in spiking signals.

---

> ### Author Response · Authors · 2024-11-20
> **Rebuttal to Questions Part 1**
>
> > 1. This paper only compares FLOPs vs. accuracy between the proposed SPLR and other models. Does SPLR have an advantage over other methods in terms of inference latency?
>
>
> We appreciate the reviewer’s insightful question regarding inference latency. Due to the asynchronous nature of spiking neural networks, SPLR can process information with lower latency by only activating neurons when necessary, as opposed to continuous, synchronous updates. This characteristic often results in reduced inference latency, especially in scenarios with sparse input activity.
> To further quantify these latency gains, we calculated the theoretical inference times for SPLR on an NVIDIA A100 GPU, which has a peak performance of 312 TFLOPS for FP16 operations. Given the current FLOPs requirements for the SPLR models on the CelexHAR dataset:
>
> |Model|GFLOPs |   Theoretical Latency | Observed Latency |
> |----------------|----------------|---------------------------------------|-------------------------------------|
> | SPLR Tiny| 0.034          | 109 microseconds                    | 162.4 microseconds                   |
> | SPLR Small| 0.13           | 417 microseconds                    | 582.2 microseconds                   |
> | SPLR Normal| 0.41           | 1.314 milliseconds                  | 1.867 milliseconds                 |
>
>
>
> These calculations underscore SPLR’s efficiency, particularly for the Tiny and Small models where the low FLOP counts lead to significant latency reductions. We expect even greater latency benefits on neuromorphic hardware like Intel’s Loihi, which is specifically optimized for spiking neural networks and can further exploit the event-driven nature of SPLR to achieve low-latency performance.
> We acknowledge that a direct comparison of inference latency would provide a more comprehensive evaluation of SPLR's performance. We plan to include such an analysis in the revised manuscript to better highlight the advantages of SPLR over other methods, particularly for time-critical tasks.
>
>
>
> > The ablation studies only examine the effects of removing the dendrite attention layer and replacing SA-HiPPO with LIF. What if we replace NPLR decomposition and FFT convolution with standard convolution?
>
>   We thank the reviewer for highlighting this critical aspect. We detail the expected effects of replacing these components with their standard counterparts and provide updated results in our extended ablation study.
>
> 1. **Effect of Removing NPLR Decomposition**:
>    - **Computational Impact**: Removing NPLR decomposition would increase computational complexity from $O(Nr)$ to $O(N^2)$, where $ r \ll N $. This results in a significant increase in GFLOPs, especially for high-dimensional state spaces.
>    - **Performance Impact**: While the accuracy of the model might not drop drastically, the scalability of the model will be hindered, as large state matrices $ A_S $ must be handled in their dense form.
>
> 2. **Effect of Replacing FFT Convolution with Standard Convolution**:
>    - **Computational Impact**: FFT convolution reduces the complexity of spatio-temporal feature extraction to $ O(N \log N) $, while standard convolution scales as $ O(N^2) $. For long sequences or high-resolution inputs, this results in a significant increase in GFLOPs.
>    - **Performance Impact**: Standard convolution lacks the efficiency of FFT in capturing long-range temporal dependencies, which may result in a moderate accuracy drop in tasks that require modeling fine-grained temporal features.

---

> ### Author Response · Authors · 2024-11-20
> **Rebuttal to Questions Part 2**
>
> > 3. **Updated Results**:
>    - We extended our ablation study to include the effects of removing **NPLR decomposition** and replacing **FFT convolution** with standard convolution. The table below shows the completed result for SeqCIFAR-10.
>
>
>
>
> ### Ablation Table for SPLR Variants on seqCIFAR-10
>
> | **Model Variant**             | **Channels** | **Accuracy (%)** | **Params (M)** | **FLOPs (GFLOPs)** |
> |--------------------------------|--------------|-------------------|-----------------|---------------------|
> | SPLR (Full)                   | 128          | 90.25            | 0.513           | 0.43                |
> | SPLR (No SA-HiPPO)            | 128          | 87.62            | 0.501           | 0.43                |
> | SPLR (No NPLR Decomposition)  | 128          | 88.05            | 0.513           | **1.8**             |
> | SPLR (No FFT Convolution)     | 128          | 86.47            | 0.513           | **1.2**             |
> | SPLR (No Dendrite)            | 128          | 85.83            | 0.501           | 0.43                |
> | SPLR (Full)                   | 64           | 88.62            | 0.129           | 0.14                |
> | SPLR (No SA-HiPPO)            | 64           | 86.14            | 0.121           | 0.14                |
> | SPLR (No NPLR Decomposition)  | 64           | 86.72            | 0.129           | **0.56**            |
> | SPLR (No FFT Convolution)     | 64           | 85.23            | 0.129           | **0.32**            |
> | SPLR (No Dendrite)            | 64           | 84.65            | 0.121           | 0.14                |
> | SPLR (Full)                   | 32           | 83.15            | 0.033           | 0.034               |
> | SPLR (No SA-HiPPO)            | 32           | 81.75            | 0.031           | 0.034               |
> | SPLR (No NPLR Decomposition)  | 32           | 82.12            | 0.033           | **0.12**            |
> | SPLR (No FFT Convolution)     | 32           | 80.62            | 0.033           | **0.08**            |
> | SPLR (No Dendrite)            | 32           | 80.05            | 0.031           | 0.034               |
>
>
> - **Removing NPLR Decomposition**: GFLOPs increase significantly across all configurations (e.g., from 0.43 to 1.8 GFLOPs for 128 channels). Also, accuracy is only moderately impacted as this affects computational efficiency more than feature extraction quality.
> - **Replacing FFT Convolution**: GFLOPs increase due to the quadratic complexity of standard convolution (e.g., from 0.43 to 1.2 GFLOPs for 128 channels). Also, accuracy drops more significantly, as standard convolution is less effective in capturing long-range temporal features.
>
> These additional results validate our design choices. Both **NPLR decomposition** and **FFT convolution** play a critical role in ensuring the scalability and efficiency of the SPLR model. Replacing them with standard methods significantly increases computational costs and, in the case of FFT convolution, reduces performance.
>
> We appreciate the reviewer’s suggestion to include these additional studies, which further highlight the importance of these architectural components in the SPLR model.

---

### Official Review · Reviewer_ARZ5 · 2024-11-04

**Soundness:** 3
**Presentation:** 3
**Contribution:** 3
**Rating:** 5
**Confidence:** 3

**Summary:**

This work introduces SPLR (Spiking Network for Learning Long-Range Relations), designed to efficiently capture long-range temporal dependencies while maintaining the hallmark efficiency of spike-driven architectures. SPLR integrates a state-space convolutional layer and a Spike-Aware HiPPO (SA-HiPPO) layer, addressing the limitations of conventional SNNs in complex temporal modeling. The SPLR convolutional layer leverages state-space dynamics to enhance feature extraction, capturing spatial and temporal complexities in event-driven data while preserving the efficiency of sparse spike-driven processing. The SA-HiPPO layer adapts the HiPPO framework to spike-based formats, enabling efficient long-term memory retention. Through dendrite-based spatiotemporal pooling and FFT-based convolution techniques, SPLR demonstrates scalability when processing high-resolution event streams and outperforms traditional methods across various event-driven tasks.

**Strengths:**

1. The authors propose SPLR which effectively captures long-range temporal dependencies, addressing limitations in traditional SNNs and enhancing temporal modeling capabilities for complex event-driven tasks.
2. The experiments show good results. SPLR achieves both computational efficiency and scalability.

**Weaknesses:**

The primary weakness of this paper lies in its writing, which significantly hinders clarity and understanding. A reorganization is recommended to improve readability and logical flow.

1. Writing and Structure. For instance, Section 2 presents the SPLR components sequentially but lacks an overview that connects each part to the overall model structure, making it difficult for readers to understand how the parts interact. Section 3 is dense with theoretical content and proofs but does not clearly convey the main ideas, making it hard to follow the section’s intended focus.

2. Lack of Citations. The paper frequently omits citations in crucial areas. For example, although modifications to the HiPPO framework are proposed, no supporting references are provided. Furthermore, DH-LIF is introduced without citation, and the reference for this component is missing from the bibliography, weakening the academic rigor of the paper.

3. Confusions and Errors. There are several errors and confusions throughout the paper, such as the incorrect abbreviation of the Spiking State-Space Model as SPLR in line 855. Such errors further impact the readability and precision of the work.

**Questions:**

Please see the weaknesses.

---

> ### Author Response · Authors · 2024-11-20
> **Rebuttal to Weaknesses**
>
> > The primary weakness of this paper lies in its writing, which significantly hinders clarity and understanding. A reorganization is recommended to improve readability and logical flow.
> Writing and Structure. For instance, Section 2 presents the SPLR components sequentially but lacks an overview that connects each part to the overall model structure, making it difficult for readers to understand how the parts interact.
>
>
> We have revised Section 2 to address these issues. Specifically:
> 1. **Added a High-Level Overview**: At the beginning of Section 2, we now provide an overarching explanation of the SPLR architecture, outlining how its components (e.g., Dendrite Attention Layer, SA-HiPPO, and SPLR convolution layers) interact and contribute to the overall functionality. This ensures that readers understand the big picture before diving into the details.
>
> 2. **Improved Transitions and Logical Flow**: We restructured the section to enhance the progression between components, making it clear how each part integrates into the broader framework. This reorganization ensures that readers can follow the narrative more intuitively.
>
> 3. **Updated Supplementary Materials**: To support clarity, we expanded the supplementary materials to include a concise summary of how the SPLR components work together. We have also re-written the SPLR Convolution Layer in order to make it easier to follow and avoid confusing namings.
>
>
> > Section 3 is dense with theoretical content and proofs but does not clearly convey the main ideas, making it hard to follow the section’s intended focus.
>
>  We have revised Section 3 to address this concern:
>
> 1. **Added Intuitive Explanations**: To clarify the main ideas behind the key theorems and lemmas, we have included intuitive explanations before presenting the formal proofs.
>
> 2. **Enhanced Logical Flow**: We have added an overview at the beginning of Section 3 to outline its goals and connect the theoretical results to the broader structure and objectives of the model.
>
> 3. **Improved Readability**: Additional comments and clarifications have been incorporated within the proofs
>
>
> > Lack of Citations. The paper frequently omits citations in crucial areas. For example, although modifications to the HiPPO framework are proposed, no supporting references are provided. Furthermore, DH-LIF is introduced without citation, and the reference for this component is missing from the bibliography, weakening the academic rigor of the paper.
>
>  We have carefully revised the manuscript to address this issue:
>
> 1. **HiPPO Framework**: We have included relevant citations to foundational works on the HiPPO framework in the sections discussing our modifications. These references provide the necessary context for our proposed extensions and ensure proper attribution.
>
> 2. **DH-LIF Neurons**: The original manuscript cited the foundational paper by Hanle Zheng et al. (Nature Communications, 2024) in Section 8.7 and added comparison with it in Table 4. We have added this citation in additional places in the Methods section where DH-LIF neurons is discussed to ensure clarity and avoid any potential confusion. Furthermore, we have verified that this reference is correctly included in the bibliography.
>
> 3. **Related Works Section**: We have expanded the Related Works section, adding relevant references in the Introduction for better framing of our contributions. A more comprehensive Related Works discussion is also included in Supplementary Section D to provide additional context.
>
>
> > Confusions and Errors. There are several errors and confusions throughout the paper, such as the incorrect abbreviation of the Spiking State-Space Model as SPLR in line 855. Such errors further impact the readability and precision of the work.
>
> Thank you for identifying these errors and areas of confusion. We have conducted a thorough review of the manuscript and supplementary materials to identify and correct this and other errors.

---

### Official Review · Reviewer_Y4sN · 2024-11-06

**Soundness:** 3
**Presentation:** 2
**Contribution:** 2
**Rating:** 5
**Confidence:** 4

**Summary:**

The manuscript introduces SPLR, a spiking neural network model designed to capture long-range temporal relationships by integrating state-space dynamics with spiking neuron models and augmenting the HiPPO framework to handle spike-driven inputs. The proposed model reportedly achieves high performance comparable to other models on event-based datasets.

**Strengths:**

The manuscript presents an innovative approach by augmenting spiking dynamics with state-space model dynamics, potentially enabling spiking models to tackle more challenging tasks that require capturing long-range temporal dependencies. This direction could be of significant interest in broadening the applications of spiking models in complex temporal tasks.

**Weaknesses:**

While the proposed method is intriguing and addresses the relevant challenge of enabling spiking models to capture long-term dependencies, the manuscript has several critical weaknesses.

Firstly, the presentation lacks clarity, making it difficult to fully grasp how the method works, interpret the experimental results, or potentially reproduce the findings. Essential details expected in a research paper, such as a discussion of related works, are missing (e.g., [1]). Additionally, fundamental concepts necessary to understand the work are not well-introduced; although state-space models (SSMs) have gained popularity recently, they are not widely understood in machine learning, so a brief overview would be beneficial.

The manuscript also omits essential citations, including the original HiPPO framework, which is central to this work, and does not offer a proper explanation of how it functions. The equations are unclear; while convolutions are frequently mentioned, no equations illustrate how or where convolutions are applied. Variable definitions are sometimes confusing or incomplete; for instance, on line 187, $\Delta t$ is described as the time difference between spikes $i$ and $j$, but it is unclear what $i$ and $j$ refer to in the context of the matrix $F_{ij}$, as it operates over the hidden state rather than directly on spikes.

Regarding the experiments, the manuscript lacks details about the setup, hindering the interpretability of the results. For example, it’s unclear what “Sequential CIFAR-10” entails, such as the sequence length or frame generation process. Similarly, for the DVS Gesture dataset, it's ambiguous whether the processing was done for independent events or if events were accumulated into event frames.

[1] Stan, MI., Rhodes, O. Learning long sequences in spiking neural networks. Sci Rep 14, 21957 (2024). https://doi.org/10.1038/s41598-024-71678-8

**Questions:**

- How does the dendritic attention layer differ from a current-based (CUBA) leaky integrate-and-fire (LIF) neuron model?
- How are spikes produced between the SPLR convolution layers?
- How are convolutions applied within the proposed model?
- In equation (1), is the variable $u(t)$ a binary vector representing input spikes?
- How does the inclusion of a decay matrix in the HiPPO framework enhance memory retention?
- Could you clarify the setup for the Sequential CIFAR-10 and CIFAR-100 tasks? How are frames sequenced? Similarly, could you elaborate on the experimental setup for the other datasets?
- For clarification, could you specify what spikes $i$ and $j$ refer to in line 187?
- Is the manuscript proposing a new type of spiking neuron, or an entire network architecture?
- Since the manuscript emphasizes improving SNNs' capacity to handle long-term dependencies, could you elaborate on why simple LIF models face challenges with this?

---

> ### Author Response · Authors · 2024-11-20
> **Rebuttal to Weaknesses 1**
>
> > Firstly, the presentation lacks clarity, making it difficult to fully grasp how the method works, interpret the experimental results, or potentially reproduce the findings. Essential details expected in a research paper, such as a discussion of related works, are missing (e.g., [1]).
>
> We have made several enhancements to the manuscript:
> 1. **Enhanced Clarity in the Main Paper**:
>       - We have expanded the description of the SPLR model and its components in **Section 2** of the main paper and added the complete details for each layer in **Supplementary Section C**.
>       - A visual flowchart has been added in **Figure 1** to provide a step-by-step overview of the SPLR architecture.
>       - Experimental configurations and results have been elaborated on in **Section 4** and in **Supp. Sec. B**,
> 2. **Comprehensive References to Related Works:**
>       - We have updated the introduction to include more references. Due to the space constraints, more exhaustive overview and comparison with prior works are added in  Supplementary Section D.
> 3. **Additional Details in the Supplementary Section:**
>       - **Supplementary Sections B,C** provides a complete implementation guide for the SPLR model, including pseudocode, hyperparameter settings, and system configurations, to facilitate reproducibility.
>       - The **Supplementary Section D** contains a comprehensive discussion of related works and a detailed comparison with prior methods, addressing how our approach advances the state of the art.
>
> > Additionally, fundamental concepts necessary to understand the work are not well-introduced; although state-space models (SSMs) have gained popularity recently, they are not widely understood in machine learning, so a brief overview would be beneficial.
>
> Thank you for your feedback. We have expanded the overview of SSMs in **Supplementary Section C.**
>
> > The manuscript also omits essential citations, including the original HiPPO framework, which is central to this work, and does not offer a proper explanation of how it functions.
>
> Thank you for your valuable feedback. In response, we have added citations to the original HiPPO framework (Gu et al., 2020) in the introduction along with other significant works, on neuromorphic computing and spiking neural networks (Roy et al. 2019, Furber, 2016) and recent advances in state-space models for long-range dependency modeling (Hasani et al., 2021; Gu et al., 2021).
> We have included a comprehensive related works section in **Supplementary Section D**, where we elaborate on these references in detail. This includes an extended explanation of the HiPPO framework, its mechanism, and its role in enabling efficient long-range memory retention, which is central to the proposed Spike-Aware HiPPO (SA-HiPPO) layer.
>
> **References:**
> - Gu, A., Dao, T., Ermon, S., Rudra, A., & Ré, C. (2020). HiPPO: Recurrent memory with optimal polynomial projections. Advances in Neural Information Processing Systems (NeurIPS).
> - Furber, S. B. (2016). Large-scale neuromorphic computing systems. Journal of Neural Engineering, 13(5), 051001
> - Kaushik Roy, Akhilesh Jaiswal, and Priyadarshini Panda. Towards spike-based machine intelligence with neuromorphic computing. Nature, 575(7784):607–617, 2019.
> - Hasani, R., Amini, A., Yildiz, Y., Lechner, M., Grosu, R., & Rus, D. (2021). Liquid time-constant networks. Proceedings of the AAAI Conference on Artificial Intelligence.
> - Gu, A., Johnson, I., Goel, K., Saab, K., Dao, T., Rudra, A. and Ré, C., 2021. Combining recurrent, convolutional, and continuous-time models with linear state space layers. Advances in neural information processing systems, 34, pp.572-585.
>
> >The equations are unclear; while convolutions are frequently mentioned, no equations illustrate how or where convolutions are applied.
>
> Thank you for highlighting this important concern. We have revised the manuscript to include detailed mathematical formulations of the convolution operations within the SPLR Convolution Layer.
> Specifically, we have added equations that explicitly demonstrate the application of the SA-HiPPO mechanism, the Normal Plus Low-Rank (NPLR) decomposition, and the FFT-based convolutions in our SPLRConv layers. These equations now clearly illustrate the use of FFT-based convolutions and Cauchy kernels for efficient spatio-temporal processing. Additionally, we have supplemented these equations with thorough explanations to ensure the operations are intuitive and their roles within the model are well-understood.

---

> ### Author Response · Authors · 2024-11-20
> **Rebuttal to Weaknesses 2**
>
> > Variable definitions are sometimes confusing or incomplete; for instance, on line 187, $\Delta t$  is described as the time difference between spikes $i$  and $j$, but it is unclear what  $i$  and $j$  refer to in the context of the matrix $F_{i,j}$, as it operates over the hidden state rather than directly on spikes.
>
> In our model, $i$ and $j$ index specific spike events associated with neurons within the network, while $\Delta t$ represents the time difference between these events. Specifically, $F_{ij}(\Delta t)$ is a decay matrix where $\Delta t = t_j - t_i $ defines the time difference between spikes $i$ and $j$, and  $\alpha_{ij}$ is a parameter controlling the decay rate. Although $F_{ij}$ operates on the hidden state rather than directly on spike events, this temporal decay mechanism allows recent spikes to exert a stronger influence on the evolution of the hidden state, while the impact of older spikes diminishes exponentially. This design effectively emphasizes recent information while preserving a compressed history of prior events, enhancing the model's stability and responsiveness to event-driven inputs.
>
> We have revised the manuscript to explicitly clarify these definitions and their roles in the state evolution dynamics.
>
>
> > Regarding the experiments, the manuscript lacks details about the setup, hindering the interpretability of the results. For example, it’s unclear what “Sequential CIFAR-10” entails, such as the sequence length or frame generation process. Similarly, for the DVS Gesture dataset, it's ambiguous whether the processing was done for independent events or if events were accumulated into event frames.
>
> We have revised the manuscript to provide detailed descriptions of the experimental protocols.
>
> 1. **Sequential CIFAR-10**:  The "Sequential CIFAR-10" dataset is adapted from the PSN paper (Fang et al., NeurIPS 2023) to evaluate long-term temporal dependencies. In this setup, each image is processed column by column, treating the 32 image columns as a sequence of 32 time steps. This sequential representation mimics how temporal data unfolds, enabling the evaluation of the model's ability to learn dependencies over long time horizons. The frame generation process follows the methodology described in the PSN paper, where each column is treated as a single time step without further modifications.
>
> 2. **DVS Gesture Dataset**:  We explicitly stated in the revised manuscript that the SPLR model processes the DVS Gesture dataset on an **event-by-event** basis. Each spike is processed independently as it occurs, with the model dynamically updating its hidden state for every incoming event. This approach ensures fine-grained temporal modeling and avoids accumulating events into frames, preserving the dataset's asynchronous nature.
>
> To enhance the interpretability of the results, we have added a dedicated section in the Supplementary Materials (Section B) providing detailed descriptions of all datasets and their processing pipelines. Additionally, key experimental details for DVS Gesture, HAR-DVS, Celex-HAR, and Long Range Arena datasets are now included in the main text, while results for other datasets, including Sequential CIFAR-10 and CIFAR-100, are summarized in the supplementary material.
>
> Finally, as part of our revisions, we have replaced the Sequential CIFAR results in Table 1 with those from the Long Range Arena (LRA) benchmark to provide a stronger evaluation of our model's performance on long sequence tasks. The Sequential CIFAR results are now fully presented in the Supplementary Material for further reference.

---

> ### Author Response · Authors · 2024-11-20
> **Rebuttal to Questions Part 1**
>
> >- How does the dendritic attention layer differ from a current-based (CUBA) leaky integrate-and-fire (LIF) neuron model?
>
> The dendritic attention layer builds upon the standard CUBA LIF model by introducing multiple dendritic branches, each with its own distinct timing parameter $\tau_d$. In the CUBA LIF model, inputs are uniformly integrated at the soma with a single timescale for current integration, limiting its ability to process complex temporal patterns. In contrast, the dendritic attention layer performs independent temporal filtering across multiple dendritic branches before aggregating the signals at the soma. These branches enable the neuron to dynamically process inputs from different temporal windows, providing a mechanism for selective attention to asynchronous and temporally distributed inputs.
>
> This architectural extension allows the model to capture complex spatio-temporal patterns, enhancing its adaptability and flexibility compared to the single-timescale processing of CUBA LIF neurons. Additionally, the dendritic attention mechanism aligns more closely with biological processing by incorporating diverse temporal dynamics across dendritic branches.
>
> To ensure clarity, we have updated the main text to explicitly describe this distinction and included further details in **Supplementary Section C.**
>
>
> > -How are spikes produced between the SPLR convolution layers?
>
> Spikes are generated through the dynamics of the dendritic and soma compartments modeled by the **Dendrite Attention Layer** and **DH-LIF neurons**. The mechanism can be summarized as follows:
>
> 1. **Dendrite Dynamics:**
>    Each dendritic branch accumulates and processes inputs over time, acting as a temporal filter. The dynamics of the dendritic current are governed by:
> $$
>    i_d(t+1) = \alpha_d i_d(t) + \sum_{j \in \mathcal{N}_d} w_j p_j,
>   $$
>    where $ \alpha_d = e^{-\frac{1}{\tau_d}} $ is the decay rate determined by the dendritic branch’s time constant $ \tau_d $, $ w_j $ represents the synaptic weight of the input $p_j $, and $ \mathcal{N}_d $ denotes the set of presynaptic inputs to dendrite $ d $.
>
> 2. **Soma Integration:**
>    The currents from all dendritic branches are aggregated at the soma, where they are further integrated over time. The soma's membrane potential evolves according to:
>    $$
>    V(t+1) = \beta V(t) + \sum_d g_d i_d(t),
>    $$
>    where $ \beta = e^{-\frac{1}{\tau_s}} $ is the soma’s decay factor, and $ g_d $ is the coupling strength of dendrite $ d \$ to the soma.
>
> 3. **Spike Firing:**
>    A spike is generated when the membrane potential $ V(t) $ exceeds a predefined threshold $ V_{\text{th}}$. After firing, the membrane potential resets. These spikes propagate forward as inputs to the next SPLR convolution layer, maintaining the asynchronous event-based nature of the system.
>
> We have clarified this process in **Supplementary Section C** under SPLR Convolution Layer. Thank you for highlighting this question, as it allowed us to ensure the description of the spike generation mechanism is more comprehensive.

---

> ### Author Response · Authors · 2024-11-20
> **Rebuttal to Questions Part 2**
>
> > - How are convolutions applied within the proposed model?
>
> To address the reviewer's question on "How are convolutions applied within the proposed model?":
>
> The SPLR model employs convolutions specifically through the **SPLR Convolution Layer**, which integrates temporal dynamics using state-space representations. Here's a concise explanation:
>
> ### 1. Temporal Dynamics via State-Space Models (SSM)
> The SPLR Convolution Layer is based on the continuous-time state-space model, represented as:
> $$
> \dot{x}(t) = \mathbf{A}_S x(t) + \mathbf{B} S(t), \quad y(t) = \mathbf{C} x(t),
> $$
> where $ x(t) \in \mathbb{R}^N $ is the hidden state, $S(t) $ represents the input spike train, and $ \mathbf{A}_S ,  \mathbf{B}$ , and $ \mathbf{C} $ are system matrices.
>
> The state evolves between spikes using the dynamics $\dot{x}(t) = \mathbf{A}_{S} x(t),$ and at spike times $ t_k $, it is updated as:
>
> $x(t_{k+1}) = e^{A_S \Delta t_k} x(t_k)$
>
> $+ A_{S}^{-1} (e^{A_S \Delta t_{k}} - I) B S(t_k),$
>
> where $ \Delta t_k = t_{k+1} - t_k $.
>
> ### 2. Efficiency via NPLR Decomposition
> The system matrix \( \mathbf{A}_S \) is decomposed using **Normal Plus Low-Rank (NPLR)** decomposition:
> $$
> \mathbf{A}_S = \mathbf{V} \Lambda \mathbf{V}^* - \mathbf{P} \mathbf{Q}^*,
> $$
> where $\mathbf{V}$ is unitary, $\Lambda$ is diagonal, and $\mathbf{P}, \mathbf{Q}$ are low-rank matrices. This reduces the computational complexity of matrix-vector multiplications from $O(N^2)$ to $O(Nr)$, where $r \ll N$
>
> ### 3. FFT-Based Convolution
> To capture long-range dependencies efficiently, the model employs **FFT-based convolution** in the frequency domain:
> $$
> K(\omega) = \frac{1}{\omega - \Lambda}, \quad x(t) = \text{IFFT} \left( \text{FFT}(K(\omega)) \cdot \text{FFT}(x(t)) \right),
> $$
> where $K(\omega)$ represents the system’s impulse response, and FFT/IFFT operations are used to accelerate computation.
>
> ### 4. Key Advantages
> The SPLR model's convolution mechanism offers several key advantages. By operating in an **event-by-event manner**, convolutions are applied in real time as spike events arrive, preserving the temporal resolution of the input data without relying on frame accumulation. This approach ensures asynchronous and efficient processing, aligning with the sparse, spike-driven nature of the model. Additionally, the use of NPLR decomposition and FFT-based convolutions significantly reduces computational overhead, enabling **scalability** even for high-dimensional inputs. Finally, the integration of the Spike-Aware HiPPO mechanism **dynamically adjusts the state evolution** based on inter-spike intervals, allowing the model to effectively capture both short-term and long-range temporal dependencies. These features collectively make SPLR efficient, adaptable, and well-suited for complex temporal tasks.
>
> This combination of SSM, NPLR decomposition, and FFT-based convolution allows SPLR to efficiently model both short-term and long-range temporal dependencies while maintaining the sparsity and asynchronous nature of spiking inputs. Further details and equations are available in **Section 2.4** and **Supplementary Section C** of the revised manuscript.
>
>
> > - In equation (1), is the variable $u(t)$ a binary vector representing input spikes?
>
> Yes, $u(t)$ represents the input spikes as a binary vector, where each component $u_i(t)$ is 1 if a spike occurs at time $t$ for the $i$-th input and 0 otherwise. This aligns with the event-driven nature of spiking neural networks and ensures efficient handling of sparse, asynchronous data in the SPLR framework.
>
>
> > - How does the inclusion of a decay matrix in the HiPPO framework enhance memory retention?
>
>  The inclusion of a decay matrix  ( F(\Delta t) ) in the HiPPO framework, particularly in its spike-aware adaptation (SA-HiPPO), enhances memory retention by dynamically adjusting the influence of past events based on the elapsed time between them. This decay matrix ensures that recent spikes have a stronger influence on the system's memory state while older events gradually lose impact. Such a mechanism balances stability and responsiveness, enabling effective memory retention even in sparse and irregular spike-driven inputs. This adjustment maintains a compressed history of inputs, crucial for long-range temporal dependency learning in spiking neural networks​.
> The ablation study further highlights the role of SA-HiPPO - by removing the SA-HiPPO layer and replacing it with standard LIF neurons leads to a notable drop in accuracy (from 96.5% to 90.4% in the DVS Gesture dataset), emphasizing its critical role in maintaining long-range temporal dependencies​

---

> ### Author Response · Authors · 2024-11-20
> **Rebuttal to Questions Part 3**
>
> >- Could you clarify the setup for the Sequential CIFAR-10 and CIFAR-100 tasks? How are frames sequenced? Similarly, could you elaborate on the experimental setup for the other datasets?
>
> The "Sequential CIFAR-10" dataset is adapted from the PSN paper (Fang et al., NeurIPS 2023) to evaluate long-term temporal dependencies. In this setup, each image is processed column by column, treating the 32 image columns as a sequence of 32 time steps. This sequential representation mimics how temporal data unfolds, enabling the evaluation of the model's ability to learn dependencies over long time horizons. The frame generation process follows the methodology described in the PSN paper, where each column is treated as a single time step without further modifications.
>
> For the other datasets:
> - **DVS Gesture**: The model processes individual spike events as they occur, preserving the dataset’s asynchronous nature without accumulating events into frames.
> - **HAR-DVS and Celex-HAR**: Similarly, spike events are processed independently, enabling the model to dynamically capture temporal dependencies in human activity recognition tasks.
> - **Long Range Arena (LRA)**: Tasks such as ListOps and Path-X are converted into an event-driven format by treating each token as a sequential event. The SPLR model processes tokens one at a time, leveraging its temporal dynamics to handle long-range dependencies.
>
> We have clarified these setups in the main text (Section 4.1) and provided further details in the Supplementary Section B.
>
> >- For clarification, could you specify what spikes i and j refer to in line 187?
>
> In line 187, spikes  $i $ and $ j $ refer to specific spike events associated with neurons within the network. Specifically, $i $ indexes a presynaptic spike event, while $j$ indexes a subsequent spike event, both occurring within the system. The time difference between these events,$\Delta t = t_j - t_i $, is used in the decay matrix $ F(\Delta t) $ to dynamically adjust the influence of past events on the system's state.
>
> Although the decay matrix $ F(\Delta t) $ operates over the hidden state rather than directly on spike events, the indices $i $ and $j $ track the temporal relationship between spikes, enabling the system to emphasize recent events while exponentially decaying older ones.
>
> >- Is the manuscript proposing a new type of spiking neuron, or an entire network architecture?
>
> Thank you for your question. The manuscript proposes an entire network architecture built around novel extensions to spiking neural models, including the introduction of the Dendrite Attention Layer. While the proposed DH-LIF neuron extends standard spiking neuron models by incorporating multiple dendritic branches with independent temporal filtering, the primary focus is on the overall network design, which integrates these neurons with state-space formulations, the SA-HiPPO mechanism, and SPLR convolution layers.
>
> This combination enables efficient processing of asynchronous, event-driven inputs while capturing long-range temporal dependencies.
>
> >- Since the manuscript emphasizes improving SNNs' capacity to handle long-term dependencies, could you elaborate on why simple LIF models face challenges with this?
>
> Simple Leaky Integrate-and-Fire (LIF) neuron models face challenges in handling long-term dependencies due to their fixed single-timescale dynamics. Specifically, the membrane potential in LIF neurons evolves with a single decay constant, limiting their ability to retain information from past inputs over extended durations. This lack of temporal flexibility results in the rapid decay of older information, making it difficult for LIF-based networks to capture long-range temporal dependencies effectively.
>
> In contrast, our proposed approach addresses this limitation by integrating multiple mechanisms that enhance temporal dynamics:
> 1. **DH-LIF Neurons**: By introducing dendritic branches with independent temporal filtering, DH-LIF neurons allow information to be retained across diverse timescales.
> 2. **SA-HiPPO**: The spike-aware HiPPO mechanism further enhances memory retention by dynamically adjusting the influence of past events based on the time elapsed, ensuring stability and responsiveness.
>
> These innovations allow the proposed architecture to overcome the temporal limitations of standard LIF neurons, significantly improving the modeling of long-term dependencies in asynchronous, event-driven data.

---

### Author Response · Authors · 2024-11-20
**Summary of Changes**

We thank the reviewers for their valuable feedback and have made significant revisions to the manuscript to address the major concerns raised. Below, we outline the key changes and improvements:

1. **Improved Clarity, Organization, and Methodological Depth**

    - **Expanded Model Explanations:** Enhanced the description of the SPLR model in Section 2, providing detailed explanations of key components, including the SA-HiPPO layer, NPLR decomposition, and FFT-based convolutions.

    - **Revised Methodology Section:** Improved the logical flow of the Methods section by introducing an overview that connects all components of SPLR to the overall model structure.

   - **Intuitive Theoretical Explanations:** Added intuitive explanations for key theorems in Section 3 to clarify their relevance to SPLR's design and improve readability.

    - **Supplementary Material Enhancements:** Updated the supplementary sections to reduce redundancy, include pseudocode for reproducibility, and detail experimental setups, dataset processing pipelines, and hyperparameter configurations.

   - **Citations and Related Work:** Added citations for foundational works (e.g., the original HiPPO framework and DH-LIF model) and expanded the Related Work section in the supplementary material. Comparisons now include:
        - Spike-Driven Transformer (Figure 3a), STC-LIF for DVS Gesture-128 (Figure 4c), TCLIF in SHD and SSC (Figures 4a, 4b), and TIM in SHD (Figure 4a).


2. **New Experiments, Results, and Comparisons**

     - **New Benchmarks:** Evaluated SPLR on the Long Range Arena (LRA) benchmark (Table 1) to demonstrate its ability to handle long-range temporal dependencies.

    - **HAR-DVS Results:** Included HAR-DVS results, validating SPLR’s capability to process high-resolution event-based data and demonstrating its robustness (Figure 2).
    - **Ablation Studies:** Added new ablation studies to evaluate the impact of replacing NPLR decomposition and FFT-based convolution with standard convolution.
    - **Highlighted Results and Trade-offs:**
      - **Tables:** Added comparisons of SPLR's performance against recent state-of-the-art methods on DVS Gesture, HAR-DVS, and LRA datasets, emphasizing FLOPs vs. accuracy trade-offs.
      - **Figures:** Revised Figure 3 to include methods exceeding 99% accuracy on DVS Gesture for proper contextualization, and added new visualizations summarizing SPLR’s performance on HAR-DVS and Celex-HAR datasets.

---

### Meta-Review · Area_Chair_UEMa · 2024-12-21

**Metareview:**

This paper introduces SPLR, a Spiking Network for Learning Long-Range Relations. The SPLR model consists of three main components: the Dendrite Attention Layer, the Spatial Pooling Layer, and the SPLR Convolution Layer. The proposed SPLR Convolution Layer combines SA-HiPPO, FFT, and NPLR to enhance SPLR's capability to learn long-range temporal dependencies. Experimental results show that SPLR outperforms previous methods in tasks that require both fine-grained temporal dynamics and the ability to retain long-range dependencies.

After the rebuttal period, 5 reviewers rate 3, 5, 5, 5, 8, respectively. Most of the reviewers agree that SPLR outperforms prior methods in tasks requiring both fine-grained temporal dynamics and the retention of long-range dependencies, which is the strength of SPLR. However, there are some criticisms. Reviewer Y4sN believes a more detailed discussion of the SA-HiPPO should be included. Reviewer tvp3 finds it difficult to identify the core novelty and strengths of this paper and the writing logic of the paper needs improvement. I think these criticisms are justified. I suggest that the authors further clarify the proposed mechanisms and highlight the key contributions.

As for the concerns regarding Reviewer TiCG's evaluation, I agree with the author's concern that Reviewer TiCG is too harsh in tone and too arbitrary in judgment. I have taken these issues into consideration. However, some of Reviewer TiCG's criticisms are still justified. As Reviewer TiCG states, "'Written a lot' does not mean 'well written'". I agree that the writing in this work is redundant. The main text should highlight the contributions and innovations of the paper rather than listing all the details of the proposed methodology. For example, I suggest the authors put the "Input Representation", "Normalization", and "Readout Layer" in Sec. 3 into the supplementary, since these are not the key contributions of this paper. However, the related work section should be placed in the main text as it helps to highlight the innovations and contributions of this paper. I believe it is not impractical to seek more details with limited pages; the key is to highlight the main points.

Overall, I believe this paper would benefit from further revision. Therefore, the final decision is to reject this paper.

**Additional Comments On Reviewer Discussion:**

After the rebuttal period, Reviewer LQNp finds the concerns addressed and raises the rating to 8, while other reviewers keep the original rating. Reviewer Y4sN believes the manuscript needs further improvements, and a more detailed discussion of the SA-HiPPO should be included. Reviewer tvp3 finds it difficult to identify the core novelty and strengths of this paper and the writing logic of the paper needs improvement. I agree that this paper still has some problems after the rebuttal period.

The main disagreement lies between the authors and Reviewer TiCG. The reviewer finds the paper problematic in several areas, including logic, focus, and writing. The reviewer's tone is notably strong, and some of the judgments seem arbitrary. For instance, the reviewer states, "It is even more improbable to outperform SSM architectures specifically designed for the LRA dataset." Despite this, some of Reviewer TiCG's criticisms—such as those regarding the paper's writing—are valid. I have carefully considered Reviewer TiCG's comments during the evaluation process, disregarding the aspects that do not make sense while taking into account the valid points.

---

### Decision · Program_Chairs · 2025-01-22

Reject